



# Long-lived contrails and convective cirrus above the tropical tropopause

Ulrich Schumann[1], Christoph Kiemle[1], Hans Schlager[1], Ralf Weigel[2], Stephan Borrmann[2,3], Francesco D'Amato[4], Martina Krämer[5], Renaud Matthey[6], Alain Protat[7], Christiane Voigt[1,2], Michael Volk[8]

[1] Deutsches Zentrum für Luft- und Raumfahrt, Institut für Physik der Atmosphäre, 82234 Oberpfaffenhofen, Germany
[2] Johannes-Gutenberg-University, Institute for Atmospheric Physics, Mainz, Germany
[3] Max-Planck-Institute for Chemistry, Mainz, Germany
[4] Istituto Nazionale di Ottica, CNR, Firenze, Italy
[5] Forschungszentrum Jülich, Institut für Energie und Klimaforschung (IEK-7), Jülich, Germany
[6] Université de Neuchâtel, Laboratoire Temps-Fréquence, Neuchâtel, Switzerland
[7] Australian Bureau of Meteorology, Research and Development Branch, Melbourne, Victoria, Australia
[8] University of Wuppertal, Department of Physics, Wuppertal, Germany

*Correspondence to*: Ulrich Schumann (ulrich.schumann@dlr.de)

**Abstract.** Contrails of the Russian high-flying research aircraft M-55 "Geophysica" are investigated in measurements above the tropical tropopause during the SCOUT-O3 field-experiment near Darwin, Australia, in 2005. The aircraft reached 19 km altitude, far above the tropopause with -87°C temperature at 17 km. In-situ, lidar, and microwave-temperature profiler measurements on board the Geophysica are used. An upward-looking lidar on the German research aircraft "Falcon", the CPOL radar near Darwin, and NOAA-AVHRR satellites provide complementary data. Exhaust emission indices are derived from a self-match experiment of the Geophysica in the polar stratosphere in 2010. Plume positions are estimated based on measured or analyzed wind and parameterized wake vortex descent. One contrail is detectable in a photo, and characterized in-situ during contrail formation downwind of the overshooting convective system "Hector" of 16 November 2005. The upper part of the contrail formed in the tropical lower stratosphere at ~60 % relative humidity over ice at -82°C. The ~1-h lifetime is explained by engine water emissions, slightly enhanced humidity from Hector, low temperature, low turbulence, and possibly nitric-acid hydrate formation. The long persistence suggests large contrail coverage from future high-flying aircraft. Further Geophysica contrail parts are found in the measurements inside the strongly convective Hector clouds on 30 November 2005. Most of the non-volatile aerosol measured over Hector is traceable to aircraft emissions. Cirrus clouds observed by lidar above the anvil occur in coincidence with computed contrail positions. The upper part of the stratospheric anvil can be explained as contrail cirrus in this case. The radar indicates that the cirrus was measured in-situ mostly besides and above overshooting convection, and the maximum ice water content in the overshoots is far higher than measured along the flight path. The evidence suggests that parts of the ice clouds measured are contrails or mixtures of convective and contrail cirrus. The number of ice particles in the contrails is less than 1 % of the number of non-volatile aerosol particles,



possibly because of sublimation losses and undetected very small ice particles. The findings are of relevance with respect to hydration of the lower stratosphere, overshooting convection, and future increases of air traffic in the lower stratosphere.

- Keywords: overshooting convection, ice, contrail, cirrus , stratosphere, lifetime, tropics, exhaust

Key Points:

• Exhaust and contrail properties of the Geophysics research aircraft are determined for flights in the tropical stratosphere.
• A contrail was observed to persist below ice saturation at low temperature and low turbulence in the stratosphere for
nearly one hour.
• The ice particle concentrations found are far lower than the concentration of non-volatile aircraft aerosol in the exhaust.
• The analysis adds insight into overshooting convection and is of relevance with respect to hydration of the lower stratosphere.

**1 Introduction**

Contrails are aircraft-induced cirrus clouds forming at low ambient temperature. Much has been learned about contrails from measurements behind commercial aircraft, typically at altitudes between 8 and 12 km and temperatures between -65 and -48°C; see Schumann and Heymsfield (2016) for a recent review. Here, ice particles form by condensation of water on suitable cloud condensation nuclei (CCN), mainly nonvolatile (soot) particles in the exhaust (Lee et al., 2010; Bond et al.,
2013). The water droplets freeze quickly and grow by uptake of ambient humidity in ice-supersaturated air (Kärcher et al., 1996). Because of high ice number concentrations and sufficient sizes, the relative humidity inside the young contrail approaches ice saturation quickly (Kaufmann et al., 2014). Contrails are generally expected to survive several minutes, essentially the wake vortex phase, only as long as ambient humidity exceeds ice saturation (Paoli and Shariff, 2016; Unterstrasser, 2016), though contrails have also often been observed below ice-saturation (Kübbeler et al., 2011; Jeßberger et
al., 2013). If contrails persist long for ice-subsaturated conditions, the contrail coverage could be larger than expected so far (Schumann et al., 2015; Bock and Burkhardt, 2016; Chen and Gettelman, 2016).

Little is known about contrails at low temperatures and high altitudes. Only a few contrail measurements have been reported for temperatures below -78°C or above 15 km altitude for a research aircraft contrail (Gao et al., 2006; Schumann et al., 2016). At lower temperatures, the amount of water available from ambient air for deposition on ice is lower, implying
smaller ice particles for the same number concentration; and the amount of water from the engine emissions is of higher importance at lower temperatures and lower pressure because of lower absolute humidity in ambient air (Schumann, 2012). At low temperatures and for low soot emissions, also volatile aerosol from aircraft exhaust and ambient air may act as CCN, according to model studies (Kärcher and Yu, 2009). Future aircraft may use other fuels with lower non-volatile (nv) particle emissions and fly at higher levels so that understanding of ice formation at low temperature and low pressure is of increasing
importance (Lee et al., 2010; Moore et al., 2015). At low temperatures, nitric acid trihydrate (NAT) particles form by



condensation of nitric acid with water vapor on preexisting aerosol (Hanson and Mauersberger, 1988). NAT has been shown to be fundamental to explain formation of polar stratospheric clouds and ozone destruction in the polar stratospheres (Crutzen and Arnold, 1986; Toon et al., 1986). NAT forms also in the cold tropical tropopause region (Voigt et al., 2008). Nitric acid is taken up by ice under NAT forming conditions (Gao et al., 2016; Iannarelli and Rossi, 2016) possibly when

ambient humidity exceeds ice saturation (Gao et al., 2004). Nitric acid and water emissions from high-flying aircraft could increase stratospheric cloud formation probability (Peter et al., 1991), and enhance contrail occurrence (Arnold et al., 1992). Long-lived contrails could occur in sub-saturated air at low temperatures if ice sublimation gets retarded. Recent studies show that the atmospheric lifetimes of small ice particles containing $HNO_3$ hydrates at 190 K may be more than 20 times larger than for pure water ice (Iannarelli and Rossi, 2016). Various other microphysical processes have been suggested

which may retard ice sublimation, depending on ice surface properties (Pratte et al., 2006), exhaust pollution (Diehl and Mitra, 1998), and electric charges (Nielsen et al., 2011). Hence, we are looking for observations of contrails at low temperatures and high altitudes.

In 2005, the high-flying Russian M55 Geophysica aircraft, equipped with a large set of instruments, performed flights in cirrus near and above deep tropical convection, in particular the so-called Hector cloud, which forms over the Tiwi Islands

near Darwin, Australia, almost daily in November/December (Vaughan et al., 2008). The measurements were performed during the SCOUT-O3/ACTIVE project, and the instrumentation, meteorology, and measurement strategy are well documented (Brunner et al., 2009; Schiller et al., 2009). During the measurements, cirrus was observed up to 1.4 km above the local tropopause. Concurrent lidar measurements on board the Geophysica indicated that these ice clouds were a result of overshooting convection (Corti et al., 2008). The findings led to new insights and further research on how overshooting

convection affects the water content of the tropical tropopause region, including model studies (Frey et al., 2015). This is particularly important since water vapor, a greenhouse gas, does have a significant impact on the radiative budget of the atmosphere, and, hence, on the climate (Chemel et al., 2009; Dessler et al., 2015). High particle number concentrations of various aerosol types were measured at the same time when measuring cirrus in the lower stratosphere, and the origin of these particles remained unclear (de Reus et al., 2009; Borrmann et al., 2010; Frey et al., 2014). Upward transport of

humidity and other trace species by overshooting convection is of interest with respect to climate and ozone chemistry also at mid-latitudes (Anderson et al., 2012; Homeyer et al., 2014; Huntrieser et al., 2016).

At the low temperatures near the tropical tropopause, the Geophysica induced its own contrail. An example is documented in a photo (Figure 1), taken from the cockpit of the Deutsches Zentrum für Luft- und Raumfahrt (DLR) research aircraft Falcon, which also performed upward-looking lidar observations. The Hector anvil has an outer diameter of about

100 km and the convective activity lasts several hours. Therefore, the Geophysica penetrated the Hector cloud several times, partly along complex flight paths. Here, the question arises as to how to distinguish contrails from cirrus. Corti et al. (2008) discussed unintended contrail sampling during measurements with the Geophysica, and eliminated data from such events based on computed spreading and advection of potential contrails. A few cases of possible exhaust plume encounters without





contrails have been identified (Weigel et al., 2009) from simultaneous particle and $NO_y$ concentration peaks, and were used to estimate the particle number emission indices for the aerosol measured. These findings motivated further investigation of contrails in the Geophysica measurements. As will be shown, measured ice events were not always uniquely identifiable as contrails or convective cirrus, but this investigation also provides new insight on the properties of exhaust aerosol, contrails,

and cirrus at low temperatures, and on the Hector cloud.

## 2 Data and Methods

### 2.1 Instruments and data

For this study, we use data mainly from the SCOUT-O3 flights of 16 and 30 November 2005. In addition, for determination of exhaust emission indices for the Geophysica, see Appendix, we use data from a self-match experiment in the dry polar

stratosphere during RECONCILE on 30 January 2010 (Sumińska-Ebersoldt et al., 2012; von Hobe et al., 2013). The data originate from a set of instruments, as listed in Table 1. The time resolutions listed in this table for Geophysica instruments are taken from von Hobe et al. (2013), who also provide the instrument's accuracy, with changes explained below. The data are available from bases at Norsk Institutt for Luftforskning (NILU; http://scout-tropical.nilu.no/), and the European Commission (https://www.fp7-reconcile.eu/reconciledata).

Some changes compared to the data bases are to be noted. The archived COPAS data are 15-s running averages. For emission index and local analysis, we use 1-s data. The FISH total $H_2O$ data used passed the quality checks described in Krämer et al. (2009). Ice number concentration is provided by the FSSP100 and CIP instruments (de Reus et al., 2009), and also computed as a function of the MAS measured 532-nm backscatter coefficient (Cairo et al., 2011). Ice water content (IWC) is derived from maximum of FSSP100/CIP and FISH-FLASH data when available. Time shifts between UCSE-

temperature and other data are taken into account as determined from correlation analyses by T. Corti (Technical Note, 2007, available in the NILU data archive). The $CO_2$ data from instrument HAGAR (Homan et al., 2010) have been reanalyzed for this study. All data are now quoted on the WMO X2007 scale (Tans et al., 2009). The data are corrected for the time delay of the inlet and for slight pump-induced biases using diagnostic calibrations through the inlet pump. The data are provided with 1-Hz resolution, but the instrument time resolution is 2-3 s for the flight segments relevant to this study. The errors given

with the data are an estimate for the mean precision during the whole flight; for absolute accuracy, one has to add 0.1 µmol $mol^{-1}$. The high-frequency noise (relevant for the detection of small $CO_2$ peaks) is about 0.05 µmol $mol^{-1}$.

We also analyzed the data from other SCOUT-O3 flights in the period 16 to 30 November 2005 in Darwin, and from the Geophysica flights during the Tropical Convection, Cirrus and Nitrogen Oxides Experiment (TROCCINOX) with measurements of deep convective clouds and lightning near Sao Paulo, Brazil, on 4 and 5 February 2005 (Schumann and

Huntrieser, 2007; Corti et al., 2008); the data are available at http://www.pa.op.dlr.de/troccinox/. The results are summarized



in this paper without presenting details. Further, we checked the data from the Geophysica flights in the SCOUT-AMMA project in West Africa in 2006 (Cairo et al., 2010), but found no indications for contrail encounters during these flights.

### 2.2 Meteorological data

Meteorological data are available from the Darwin radiosonde YPDN. Numerical weather prediction reanalyses from the European Center for Medium-Range Forecasts (ECMWF; ERA data) are available with 0.5° resolution, 60 vertical levels (1100 m interval at z=18 km), every 3 h (Dee et al., 2011). Ten-day backward trajectories from the Geophysica flight track for SCOUT-O3, computed for 3-hourly ECMWF operational analyses, are available from the NILU data base (Brunner et al., 2009). The trajectories represent the large-scale history of air masses near Hector.

High-resolution (~1 km) NOAA AVHRR satellite data were provided by the Bureau of Meteorology, Melbourne, Australia, as received locally near Darwin, and processed at DLR. The infrared channels of the AVHRR provide 10.8 and 12-μm brightness temperatures (BT), giving information about the cloud top temperature for optically thick clouds. For optically thin cirrus over warm Earth surfaces, the 10.8-12 μm BT difference (BTD) reflects the different absorption of surface IR radiation by thin ice clouds in particular when these contain small ice particles (Inoue, 1985). Over optically thick clouds, the BTD should be close to zero. For further discussion see Bedka et al. (2010). The BTD is often used to identify thin cirrus and contrails (Betancor Gothe and Graßl, 1993; Luo et al., 2002; Minnis et al., 2013).

The Bureau of Meteorology also provided radar data from the scanning C-band dual-polarization radar CPOL, at 131.04°E, 12.25°S, northeast of Darwin Airport (Keenan et al., 1998). The data are gridded with 2.5 km resolution horizontally, 500 m vertically, and 10 min in time (Kumar et al., 2013). Here we use radar reflectivity Z interpolated to constant altitude and time and or to vertical cross-sections along the Geophysica flight paths for 3-9 UTC 30 November 2005.

Figure 2 shows the vertical temperature profile for 30 November 2005. For comparison with photo, lidar and Microwave Temperature Profiler (MTP) altitudes we refer to geometric altitudes z above mean-sea level. We use pressure altitude $z_p$ for comparison with ECMWF data. Pressure altitude is computed for given pressure according to the International Civil Aviation Organization (ICAO) standard atmosphere. The day was characterized by weak mean winds, with about 10 m s$^{-1}$ mean wind in the stratosphere from the north-east (Brunner et al., 2009). We note the two extreme temperature values indicating strong overshooting convection. The independent TDC and UCSE temperature data, with nominal resolutions of 1 Hz and accuracies of 0.5 K and 2 K (Schiller et al., 2008; Weigel et al., 2014), differ locally by up to 2 K at 1-s resolution, but agree in the occurrence of temperature extreme values. The radiosonde and ERA data are useful outside Hector, but differ significantly from local values, see Figure 2.





### 2.3 Plume detection method

In order to identify possible Geophysica contrail or exhaust plume encounters by in-situ measurements and with lidar observations in the "curtain" below or above the aircraft (from MAL or DIAL), we search for cross-sections of the advected plume path with the flight path. For this purpose, we perform a double loop. The outer loop considers all aircraft positions

(x, y, z) at time $t_2$. The inner loop considers all past aircraft positions at times $t_1 < t_2$, and computes the position of the plume or contrail trajectory $(x_c, y_c, z_c)$ at time $t_2$ representing the potential position of the aircraft exhaust plume/contrail including advection during the plume age $\Delta t = t_2 - t_1$. The cross-sections or the points with minimum distance between $(x,y)$ and $(x_c, y_c)$ at time $t_2$ are recorded together with the height $z_c$ as potential plume encounters.

For comparison with in-situ observations we require that the magnitude of $\Delta z = z_c - z$ remains in prescribed limits, e.g. less

than 100 m. For lidar observations, $z_c$ must be in the range of the lidar beam. For this computation, all positions are expressed in geometrical distances (in meters)

$$x= (\lambda-\lambda_0) R \cos(\varphi_0), \quad y=(\varphi-\varphi_0) R \qquad (1)$$

where $\lambda$ and $\varphi$ are geographical longitude and latitude positions (in radians for this computation) and R is the Earth radius; $\varphi_0, \lambda_0$ are the mean positions during the flight part under consideration. Plume advection is calculated for mean local wind u,

v, w versus plume age $\Delta t$,

$$x_c(t + \Delta t)= x(t) + u \, \Delta t, \quad y_c(t + \Delta t)= y(t) + v \, \Delta t, \quad z_c(t + \Delta t)= z(t) + w \, \Delta t. \qquad (2)$$

An example of results of this method is shown in Figure 3. Here the black curve is the Geophysica flight path between times $t_1$ and $t_2$, and the red curve is the computed position of the plume at time $t_2$ for a plume that started from the aircraft positions in the time interval between $t_1$ and $t_2$.

As one of several criteria for assessment of the likelihood that the computed potential plume encounters are real encounters, we compute the change in wind velocity that would be required to advect the plume exactly to the position of the measurement in the time period $(t_1, t_2)$.

$$\Delta u= \Delta x/\Delta t, \quad \Delta v= \Delta y/\Delta t, \quad \Delta w= \Delta z/\Delta t, \qquad (3)$$

where $\Delta t = t_2 - t_1$, and $\Delta x$, $\Delta y$, $\Delta z$ are the separations between the positions of the plume $(x_c, y_c, z_c)$ and the aircraft $(x,y,z)$ at

time $t_2$. We also compute the potential-temperature difference $\Delta\theta = \theta(t_2) - \theta(t_1)$ at the aircraft positions. Adiabatic wake vortex sinking or lifting does not change potential temperature $\theta$, but $\theta$ may change by $\Delta\theta = -5$ K after sinking wake vortices have mixed with ambient air with $N_{BV}= 0.025$ s$^{-1}$ near $\theta = 400$ K, after, e.g., $\Delta z \cong 200$ m descent.

For horizontal wind we use averaged in-situ measurements because of inherent oscillations in the wind data. Aircraft are known to deviate, even in quiet air, from the straight steady flight path, performing phugoid oscillations and other aircraft





dynamics oscillations with various frequencies (typically in the range 0.01 to 0.06 s$^{-1}$). Frequency details and amplitudes depend on the aircraft speed and mass and on the autopilot properties (Nelson, 1998). Such aircraft oscillations become obvious for the Geophysica when one plots the aircraft altitude and attitude angles as a function of time. The UCSE wind velocity increases for large roll angles. Therefore, we ignore the wind velocity data during maneuvers with roll angles > 5°.

We also ignore wind data when the wind direction turns from 360° to 0° or vice versa, because these data suffer from averaging based on yaw angles as noted by the UCSE team. The wind data show either very strong turbulence at flight levels or other disturbances. Therefore we average all data within altitude intervals of a few hundred meters, as shown in Figure 2d and e, and use u and v interpolated vertically in the mean wind profile. Figure 2 shows the wind results deduced from TDC wind data. When using UCSE wind, the mean values change by ±0.6 m s$^{-1}$. Obviously the true wind may differ from the

interpolated wind velocity by more than 1 m s$^{-1}$. Since the plume positions change linearly with the product of wind velocity and age, the uncertainties in these data matter, in particular for aged plumes, and require careful discussion of the results. In all applications, the analyses were repeated with variations of the wind to test the robustness of the results.

The vertical wind velocity was not measured. Model analyses suggest vertical velocities of about ± 2 m s$^{-1}$ in the stratiform region above the tropopause and far higher velocities of up to 25 m s$^{-1}$ in the convective regions (Chemel et al.,

2009). Regions with strong updrafts were avoided by the pilot as far as foreseeable. Hence, we start our analysis assuming zero vertical wind. For the plume analysis we distinguish between the primary wake and the top of the secondary wake forming above the primary wake (Paoli and Shariff, 2016). For the top of the secondary wake we assume zero descent velocity relative to ambient air. For the primary wake, which descends for some time $t_{wake}$ until final wake vortex decay, we estimate the descent velocity $w_0$ and the time $t_{wake}$ from Holzäpfel (2014). For the Geophysica, with wing span 38.4 m, mass

20 Mg and with, e.g., true airspeed 190 m s$^{-1}$, at 80 hPa air pressure, -83°C temperature, and Brunt-Väisälä frequency $N_{BV}$ = 0.025 s$^{-1}$, one finds $w_0$ = 1.34 m s$^{-1}$ and $t_{wake}$ = 6 $t_0$ ≅ 132 s, with $t_0$ = 22 s as the wake vortex time scale for this aircraft. Other models imply an about 20 % deeper descent (Unterstrasser, 2016). Hence, the primary wake sinks typically by 200 m. Since $t_0 N_{BV}$ = 0.55 < 1, the vortices dissipate before they can ascend in stratified air (Holzäpfel, 2014; Paoli and Shariff, 2016). After wake vortex decay, we assume again zero descent velocity relative to ambient air.

When in-situ measurements show events at a time $t_2$ with peaks of species concentrations that might originate from engine exhaust, one can use the inner-loop calculations to find the time $t_1$ when the species might have been emitted. The method has been applied for events with measured nv aerosol peaks for TROCCINOX (5 February 2005), SCOUT-O3 (25 November 2005) (Weigel et al., 2009), and for RECONCILE (25 and 30 January 2010) (Sumińska-Ebersoldt et al., 2012), see Table S1 in the Supplement. For 8 out of 9 events, the method shows that the measured aerosol peaks can be closely

related to exhaust plumes. The computed plume ages reach up to 2.2 h, the differences in Δθ remain below 2.4 K, and vertical distances Δz below 200 m. One aerosol peak, F1.1 of TROCCINOX, cannot be explained by Geophysica exhaust this way; it may be remainders from overshooting convection somewhere upstream hours or days earlier (Nielsen et al., 2007).





When estimates of plume dilution $N_{dil}$, i.e., the mass ratio of plume air to consumed fuel per distance, are needed, we use the empirical relationship,

$$N_{dil}(\Delta t) \approx 7000 \ (\Delta t/t_{dil})^{0.8}, \text{ with } t_{dil} = 1 \text{ s.} \tag{4}$$

This relationship fits measured dilutions for many aircraft within a factor of 3 (Schumann et al., 1998).

**3 Results**

**3.1 Photographed Geophysica contrail in the lower stratosphere**

On 16 November 2005, during the first SCOUT-O3 flight from Darwin, the Geophysica and the Falcon were on mission to observe Hector in situ and with lidar. The photo shown in Figure 1 was taken from the Falcon cockpit at 8:35 UTC (18:05 LT). At this time, the Falcon had reached the western tip of the Tiwi Islands, at 12.7 km geometric altitude, heading

westbound (~270°). The crew observed clouds, obviously with a complex contrail pattern above other clouds. The photo shows convective clouds illuminated by the sun (azimuth 252.82°, elevation 10.82°) at their western sides.

Only 6 min before the photo was taken, Hector was overflown by NOAA-12. Figure 4 shows the Geophysica flight path above the Tiwi Islands overlaid on the 10.8-µm BT image from NOAA AVHRR data. The Geophysica arrived from the north, passed Hector at a maximum altitude of 19 km, and then descended with several curves. At the time of the photo, the

Geophysica was on its way back to Darwin and below 16 km. The grey line depicts the position of the exhaust plume at the time of the photo computed for the averaged wind profile from the last 1.5 h before the photo was taken. The mean wind velocity was about 15 m s$^{-1}$ and has advected the oldest part by about 80 km in southwesterly direction during this time. We see a circular turn in the far southwest part of Figure 4 which results from the flight at about 7:30 UTC (27000 s), about 1 h before the photo was taken. The descent occurred in the clear outflow of Hector, with about 1 µmol mol$^{-1}$ enhanced $H_2O$

molar mixing ratio and enhanced CO compared to the values measured earlier on the windward side further north-east. Hence, Hector possibly contributed to a local hydration of its stratospheric outflow this day.

A weak, isolated Hector was observed and probed in its late stage this day (Brunner et al., 2009). The BT is larger than 205 K over Hector, indicating that Hector is in a dissipating stage with optically thick cloud parts below the tropopause (Bedka et al., 2010). Stronger convection with BT down to 182 K is still active near Darwin at this time. The convective

clouds in the background of the photo are visible as small disturbances in the south-west part of the satellite picture. The line-shaped cloud in the middle of the photo, slightly above the Falcon level, looks like an aged contrail. We do not know its source. This cloud can be found as narrow line in 300° direction in the 10.8-12 µm BTD, with about 1 K difference above background (not plotted). Geophysica contrails were not visible in BTD images. Fresh convective clouds in the lower right corner are remainders from Hector which otherwise is to the right and in the back of the photo plane.



From the cloud pattern it is obvious that the upper part of the photo shows contrails from the Geophysica. The optically thin contrails are at far higher levels than the Falcon, as can be seen when relating them to the horizon. The afternoon sun supports the visibility of thin cirrus and contrails by forward scattered light in spite of a low optical depth. The contrail pattern reveals the curved flight path. In the photo, we see thicker contrail edges, possibly from the primary vortices with

maximum particle concentrations, but no fall-streaks. So the ice particles are likely small and sublimating.

The camera image can be compared to a "synthetic photo". Figure 5 shows a projection of the computed contrail position to the image plane of the camera. The projection is computed as described in Schumann et al. (2013a). Azimuthal and vertical orientation of the camera (238°, -3°), and a zoom scaling factor of 0.9 are selected to reach best agreement of the synthetic scene with the photo. The result would look much different for a clock-time error >1 min. Figure 4 also shows the

Falcon position at the time of the photo, and the azimuth range covered by the camera. The white and red overlaid parts of the grey exhaust line are the computed positions of the upper and the lower contrail segments detectable in the photo, above and below 18 km geometric altitude. The circular turn after 7:30 UTC is in the far southwest part of Figure 4 and not visible in the photo. Any contrail from this turn would have occurred in the perspective of the photo just below the visible contrail part and above the thick linear cloud (Supplement, Figure S1). Two further photos exist (Figures S2 and S3), taken within a

30-s time interval, showing the scene under slightly different perspectives, confirming this interpretation. However we see no such contrail and, hence, this contrail part had been dried out at the time of the photo. It is referred to as "invisible plume" below. During descent, the Geophysica flew a large oval of 10 to 15 km width and 60 km length which formed the upper contrail, see white line in Figure 4, between this turn and the Falcon position. Later, the lower contrail part, see red line, followed. As indicated by the large differences in the simulated contrail positions in the photo perspective with and without

vortex descent, most important for the accuracy of the computed contrail positions is the wind velocity. We experimented with small changes in the vertical wind shear. Added shear of $0.002$ s$^{-1}$ relative to 17 km altitude causes significant changes in the position details. Changes of wind with time are not known and therefore not taken into account. Such changes likely explain curved contrail parts which apparently resulted from straight flight segments, e.g., the upper contrail part in the photo.

The contrail properties can be related to the in-situ measurements at the time of contrail formation. Both the upper (white) and the lower (red) contrail segments are the results of Geophysica flight segments of about 1000 s duration, corresponding to about 170 to 200 km lengths. The contrails from the upper and lower wake parts got spread horizontally by wind shear to about 1 to 3 km width, as deduced from the photo and the computed dispersion of the upper and lower plume tracer lines. The air sampled by the Geophysica was cloud-free along the descent flight path, as indicated by zero FSSP

counts. Table 2 lists the times and ages derived from the trajectories and the photo, and the properties of the air in which the contrails formed as measured in-situ with the Geophysica instruments, see Table 1 (TDC for temperature, FLASH for $H_2O$, SIOUX for $NO_y$, FOZAN for $O_3$, COLD for CO molar mixing ratios, COPAS for nv and total concentrations of particles with sizes > 10 nm, Brunt-Väisälä frequency $N_{BV}$ and standard deviation z' of isentrope altitudes from MTP).





For later discussion we also include the Schmidt-Appleman criterion (SAC) liquid critical (LC) threshold temperature for contrail formation (Schumann, 1996), the frost-point temperature $T_{ICE}$ of ice for given $H_2O$ partial pressure (Sonntag, 1994), and the existence temperature $T_{NAT}$ of nitric acid trihydrate (NAT) for given $H_2O$ and $HNO_3$ partial pressure (Hanson and Mauersberger, 1988), assuming that 80 % of the $NO_y$ abundance above NO is $HNO_3$. The properties of the invisible

plume part are also listed in Table 2 because they provide a further test for sublimation theories. UCSE temperatures are 1.3 to 1.5 K higher than the better qualified TDC temperatures. As a consequence, RHi values would be 5 % to 30 % lower when using UCSE instead of TDC.

Hence, the lower part of the stratospheric Geophysica contrail formed near the tropopause at -87°C with low humidity (about 2-3 $\mu$mol mol$^{-1}$) but near ice-saturation. The upper contrail part is nearly one hour old and formed at -82°C, at $T>T_{ICE}$

and persisted in ice-subsaturated air. Hence, the contrail originated mainly from engine emissions. Parts of the contrail are cold enough to allow for NAT existence. $T_{NAT}$ is 3.3 to 5.1 K higher then $T_{ICE}$. For further discussion, see Section 4.1.

### 3.2 Contrails and cirrus observed during the SCOUT-O3 flight of 30 November 2005

Overshooting convection with associated cirrus and possibly with contrails was observed over the Tiwi Islands north of Darwin during the morning flights of 30 November 2005 (Geophysica: 3:44 - 8:22 UTC, Falcon: 3:51 - 8:15 UTC), the so-

called "Golden Hector day" (Brunner et al., 2009). Contrails formed along the whole flight above 13 km because of the low air temperature. A sequence of photos shows Hector in its early developing stage from 1:45 to 5 UTC (11:45 to 14:30 LT) (Huntrieser et al., 2009). A contrast-enhanced version of such a figure is shown in the Supplement. The thin cirrus seen in the 5 UTC photo contains the Geophysica contrail as generated during ascent. We have no photo of Hector or contrails at later times. The following sections describe results from observations and calculations of potential contrail encounters. The

results provide indications that the measurements occurred in convective cirrus and partly in Geophysica exhaust plumes and contrails. The discrimination of both is not obvious and will be discussed in Sections 4.2 and 4.3.

#### 3.2.1 In-situ measurements

Much information about Hector, its air composition, and convective cirrus properties is available from the in-situ measurements along the Geophysica flight path. The cold-point tropopause altitude was penetrated during the Geophysica

ascent at 4:47 UTC (17233 s). The tropopause was again reached at 25296 s and 26039 s, and penetrated during final descent at about 7:25 UTC (26700 s). The TDC cold-point temperatures at the first and last times are -87.3°C and -87.1 °C, both at about 87 hPa pressure, 17.1 km pressure altitude, 17.4 km geometric altitude, and 375 K potential temperature. The MTP and ERA reanalysis data locate the cold-point tropopause slightly (100 to 400 m) deeper.

Figure 6 shows flight altitude, water vapor, ice particle and nv aerosol concentration measurements during the

stratospheric flight part. A similar plot with further data can be found in de Reus et al. (2009). We include earlier flight times because contrails formed in this early period. Events E1 to E6 (shaded time periods) between times 20600 - 25000 s (UTC





times in Table 3) have been identified before by de Reus. The six events are periods in which ice crystals were observed for at least 30 s at flight altitude. Event E6 was discussed in detail by Corti et al. (2008). The mean values of several parameters during these events were derived by de Reus et al. (2009) and are repeated in Table 3. The events occurred between 18 and 18.7 km with particle sizes between 12 and 25 μm in effective radius, $r_{eff}$, and 4.6 and 9.3 μm in volume mean radius, $r_{vol}$.

The small ratio $C=r_{vol}/r_{eff}$ indicates that most of the particle volume is controlled by a few large particles (Schumann et al., 2011), here measured by CIP, with maximum diameters exceeding 100 μm. The IWC reached up to about 1 mg m$^{-3}$, which was assessed high compared to a climatology for low-temperature cirrus (Schiller et al., 2008). The relative humidity over ice, RHi varies between 75 and 157 % in these data. Only two of the six events (E2 and E4) exhibit ice supersaturation.

We note three additional shorter periods with ice crystals, E0, E7, and E10, also listed in Table 3. For about 15 s, around

time 18780 s, the FSSP measured ice particle concentrations up to 0.013 cm$^{-3}$. This event E0 was accompanied by a significant increase in total water concentration from FISH, see Figure 6b, though the data for t < 19800 s did not pass outgassing quality checks (Krämer et al., 2009). We also find a slight enhancement in the depolarization signal at 532-nm wavelength of MAS, but no increase in the COPAS aerosol signals. Later, at times shortly after 21633 s and 22120 s, two short-duration ice events (E7 and E10) are found. During event E9 (22008 s), the FSSP100 sampled only a few ice particles,

but radar data (Section 3.2.4) indicate that this is an ice event. In addition, we see several "dry" events, without ice, with enhanced nv aerosols. For most events, at least one "large" ice particle > 25 μm was observed by CIP. CIP counted no ice particles during the ice events E0, E4, E7, E9, and E10.

In order to find indications for whether these ice or dry events could be caused by Geophysica exhaust plumes or not, we search for plume encounters, as illustrated for event E1 in Figure 3, i.e. closest approaches of the advected plume from past

flight parts with the actual flight position. Table 4 lists related analysis results with event name, day, age and time of event, the geometric altitudes z of the Geophysica at the event, with altitude and wind changes Δz, Δu, Δv, Δw required for perfect match, and related potential temperature changes Δθ. For event E6, we found that the measurements may have occurred in a superposition of contrails from four earlier flight segments. However, not all events are contrails. In fact, events with large negative Δθ (-16 K for event E2) would require strong descent of the air if it contains a contrail, which would likely lead to

adiabatic warming and contrail evaporation (Greene, 1986; Unterstrasser and Sölch, 2010). Strong updrafts could enhance contrail ice water content. The symbols in Figure 6a identify the times and altitudes of the events listed in Table 4, with open symbols for potential plume formation (C), and closed symbols for encounters (E).

Figure 7 shows a map with the complex Geophysica flight path. During ascent and final descent the Geophysica flew often in upper tropospheric cirrus with significant IWC > 0.01 mg m$^{-3}$ (grey symbols). Stratospheric cirrus was measured

while flying over the Tiwi Islands. The red symbols with event labels identify positions of computed plume and contrail encounters. Encounters of dry plumes (red outside blue flight parts) occur mainly at the periphery and to the northeast of the convective region, i.e., on the windward side in the stratosphere. This includes the events E7 and E8 with maximum nv





concentrations. Event E0 without Geophysica exhaust occurred on the southern side. Event E2 coincides with the location of minimum temperature (blue star in Figure 2a) and event E5 is close to the position of maximum temperature (red star in Figure 2a), both occurring closely together in the cirrus flight parts over the center of the Tiwi Islands, possibly because of strong updrafts and downdrafts.

The nv aerosol peaks reach concentrations up to 1000 $cm^{-3}$ in the 15-s running average measurements (up to 3300 $cm^{-3}$ in the 1-s data). The magnitude of the aerosol peaks can be explained quantitatively with the nv aerosol emission index derived in the Appendix. Since $NO_x$ data are not available for this flight, we estimate plume dilution from the empirical dilution law, Eq. (4). This dilution has been derived from exhaust measurements at low to moderate-turbulence and assumes penetration of the plume center. The computed concentrations are depicted by black circles in Figure 6b. They reach the

magnitude of the measured concentration peaks. Hence, it is possible that the nv aerosol peaks could be caused by Geophysica engine emissions.

    For exhaust plumes at, e.g., 2000 s age, for dilution from Eq. (4), and for emission indices as given in the Appendix, we would expect peaks of 0.71 $\mu mol\ mol^{-1}$, 0.65 $nmol\ mol^{-1}$, and -0.82 $nmol\ mol^{-1}$ for $CO_2$, CO, and $O_3$ (from NO titration (Zheng et al., 1994; Schulte et al., 1997)), and 122 $cm^{-3}$ for nv particles. Temperature may increase by 0.1 K from

combustion heat. More important are temperature changes after mixing of descending wake vortices with ambient air of up to about $\Delta\theta = -5$ K (see Section 2.3). For overshooting convection, we expect positive $CO_2$ and CO and negative $O_3$ mixing ratio and $\theta$ deviations correlated with positive $H_2O$ mixing ratio and aerosol concentration peaks. Hence, we expect to see the same correlations for aircraft plumes as for overshooting convection, though with smaller magnitude and with narrower plume shapes. IWC and potential temperature are not conserved during mixing with phase changes.

Scatterplots as shown in Figure 8, but without the $CO_2$ data, have been used before to obtain insights into the air mass composition along the flight path and their correlations to tropospheric or stratospheric origins (Chemel et al., 2009; Frey et al., 2015). The scatterplots show compact correlations of CO, $O_3$, and $H_2O$ molar mixing ratios with potential temperature $\theta$. $CO_2$ and CO shows a generally decreasing trend with altitude ($\theta$), while $O_3$ is increasing, as expected. Still the correlation with $\theta$ appears slightly less compact for $CO_2$ than for CO and for $O_3$, possibly reflecting different lifetimes and different air

mass origins. On average, the $H_2O$ mixing ratio was near ice saturation below 400 K and ice-subsaturated above. Lifting would cause clouds to form preferentially in the more humid layer above 390 K potential temperature.

    Between about 375 and 390 K, just above the tropopause, we see a vertical range of low $CO_2$, low CO, and high $O_3$, coinciding with low $H_2O$ and low IWC. The observed $\theta$-CO-$O_3$ correlations could not be explained with overshooting convection alone (Frey et al., 2015). Instead, horizontal advection of a drier and less polluted air mass from the east, with

more humid air above that layer, may explain the observations. In fact, the local wind profile and 10-d trajectories (Brunner et al., 2009) show that the tropospheric and stratospheric air masses come from different directions.



The mass specific particle concentration follows a correlation with θ very similar to that of CO. Hence, the nv aerosol and CO near the tropopause may both stem from a similar source (Allen et al., 2008; Heyes et al., 2009). However, the aerosol peaks in the range $390 < \theta < 430$ K, without similar peaks in CO, $O_3$ or $CO_2$, stand out. They cannot be explained by overshooting convection. Also the high IWC events stay out of the correlations. The observed IWC is higher than in model simulations (Chemel et al., 2009). Similar IWC values are found near $\theta \approx 400$ K and 360 K. Latent heat release from water condensation and freezing during deep convection may cause a temperature change $\Delta T = L\,\Delta q/cp \approx 40$ K (L = latent heat of fusion, $c_p$= specific heat capacity at constant pressure) only when very humid boundary-layer air (with more than 1 % water vapor mass content q) gets transported without dilution from the surface up to 400 K, which is not very likely. Additional water and heat may come from lateral advection and, to minor degree, from aircraft water emissions.

Figure 8e is similar to the one shown by Corti et al. (2008), with high IWC above the tropopause. Most of the high IWC data shown in that study come from measurements during this 30 November morning flight. Based on the number of points within 10 s relative to related computed plume encounters (red symbols), contrails contribute little to the overall IWC results. However, 10-s segments may not be representative for the whole aircraft effect.

For analysis of the origin of the aerosol peaks and ice events, we refer to Figure 9 with in-situ measured signals at sufficient time resolution to identify details for the events E1 to E3, E7 and E8. Similar plots are available for the later events (Figures S5, S6). From these figures one may try to classify the events as convective (C) or as exhaust (E) events or as a mixture of both (C+E), and we have listed our best estimates in Table 4, and these are supported by further evidence discussed below. For each event we evaluate local scatter plots, as shown for E7 in Figure 10, and for E2 in Figure S6. Here red points represent the data during the event duration, the blue points the neighborhood as measured shortly before and after the events, and the grey symbols show all data to characterize the variability and mean changes with altitude.

This information indicates that event E2 is likely caused by a high-reaching convective updraft, which raises air over an altitude range of 16 K in potential temperature, corresponding to about 440 m vertical ascent (see also Figure 2b). The high CO and low $O_3$ indicate air mass transport from slightly above the tropopause. The presence of large ice particles measured with CIP likely come from the cloud elements transported upward. The nv aerosol is slightly enhanced (about 50 cm$^{-3}$) but not very high and, hence, may come from tropospheric pollution, but contributions from Geophysica exhaust cannot be excluded (C+E). $CO_2$ is not enhanced, as one would expect for tropospheric air, but comes from the low-$CO_2$ layer near 380 K.

The other ice events are less clear. Event E1 exhibits high total water, and many large ice particles, slightly enhanced $CO_2$ indicating tropospheric air, but no related CO or $O_3$ peaks. So this may be air transported horizontally from a convective tower, with mixing reducing the signatures. The mixtures may contain aircraft exhaust and contrail ice. Similar findings apply to E5, E6, and E9.





Event E7 appears to be likely a contrail. This follows from the high $CO_2$ and high CO, significantly above instrument precisions, and slightly reduced $O_3$, a small enhancement in gaseous $H_2O$, a large peak in nv aerosol, and at least a few ice particle counts in spite of humidity below ice saturation. The CO peaks reach about 10 nmol mol$^{-1}$ with related $CO_2$ peak of ~0.5 μmol mol$^{-1}$. If these species come solely from engine exhaust, this would imply a CO emission index of 40±10 g kg$^{-1}$, not unrealistic compared to other measurements (Zheng et al., 1994; Fahey et al., 1995; Slemr et al., 2001). E8, E11, E12 and E13 show similar signatures, without ice, and are all potential exhaust events (E).

### 3.2.2 Observations with the downward-looking lidar on the Geophysica

Evidence for the presence of cirrus clouds above Hector during the morning flight of the 30 November was provided earlier by the downward-looking lidar (MAL), mounted on the Geophysica. Figure 11 shows a plot of the MAL backscatter ratio versus time and altitude similar to the one shown by de Reus et al. (2009), and in part also by Corti et al. (2008), but including larger time and altitude ranges and higher temporal and vertical resolution. Here we have added the computed locations of potential contrail positions in the lidar curtain below the aircraft as a function of flight time and altitude. This shows that the lidar might have seen Geophysica contrails. In particular above 17.5 km altitude, between t = 20000 s and 26000 s, at the times of the potential contrail events E1 to E13, we see several narrow clouds with low backscatter ratio at times and altitudes in approximate agreement with computed positions. Some potential plumes between 22000 and 24000 s above 17.5 km, related to the "dry" events in Table 4, show no corresponding backscatter cloud, possibly because of drier air with quickly sublimated contrails. The backscatter clouds are about 200 m deep; similar to a possible contrail depth. This figure shows the plume positions as computed for nominal mean wind. Variable wind and turbulence could cause slightly different plume encounter positions. If we would allow for additional vertical contrail spread by turbulence or by ice particle sedimentation, the simulated pattern might show even closer similarities to the observations.

Below 17.5 km, we see many high-backscatter clouds (backscatter ratio up to 300) at horizontal scales of order 200 s or 40 km and more than 1 km depth. The depolarization signals (not plotted) reach values of 30 to 70 % with the larger values for the lower and thicker clouds. These clouds are cirrus clouds, mainly in the Hector anvil. Cirrus clouds are detectable below 15 km when averaging the data over larger intervals. The lower cloud boundary or the optical thickness of the cloud below the aircraft cannot be derived from these data. Remind that MAL is a small instrument with energy per shot (3-4 μJ) and telescope aperture diameter (5 cm) limiting the signal-to-noise ratio, in particular during day time, as in this flight. The MAL lidar penetrated the cloud layer down to ground only rarely and in regions with thinner clouds near the outer edge of the anvil. Optical depth analysis was possible, however, for night-time observations (Frey et al., 2014).

### 3.2.3 Observations with the upward-looking lidar on the Falcon

A stratospheric anvil, possibly with contrail cirrus, can be detected with DIAL (Poberaj et al., 2002; Wirth et al., 2009) and characterized in terms of occurrence, geometric scales, optical depth, and particle sizes. The lidar was mounted on board the





DLR-Falcon research aircraft in zenith-viewing direction during the SCOUT-O3 measurements near Darwin. Results from the same instrument during TROCCINOX are described in Kiemle et al. (2008). Water vapor profiles could be obtained during this flight only in the lower part of the cirrus clouds and underneath, because of high signal extinction within the cirrus. But, the backscatter channels at 532 and 1064 nm wavelength were most often able to record the top of the cirrus

clouds, as well as layers of thin cirrus above. The backscatter ratio is the ratio of air molecule plus ice particle to solely air molecule backscatter coefficients, for an assumed lidar ratio (the ratio between extinction and backscatter coefficients) of 20 sr. The backscatter signals at 532 nm and 1064 nm wavelengths show similar results but the near-infrared backscatter gives the less noisy result. The Falcon and Geophysica flew simultaneously along different flight paths near Hector; the Falcon flew at altitudes below 11 km so that the Falcon lidar may have observed parts of the aged Geophysica contrail.

Stratospheric cirrus up to 18.2 km altitude was observed only during the last part of the flight, 2 h after convection started, at times when also Geophysica plumes advected into the lidar beam. Figure 12 shows the observed backscatter signal for the flight period 7:35 to 7:41 UTC when the Falcon was below the southern Hector anvil. The thicker cirrus between 12 and 16 km altitude belongs to a tropospheric anvil part. The Falcon was flying in cloud-free air below this anvil. The stratospheric anvil is composed of two layers, with dimensions as listed in Table 5 (width computed for the Falcon ground

velocity of 205 m s$^{-1}$ in this period). The positions of computed plumes are superimposed on the 1064-nm backscatter result. We find that all prominent cirrus layers above 17.4 km, i.e. above the tropopause, occur when also contrails might be present. At least part of the observed cirrus could be contrail-generated. Such stratospheric contrails would have ages of more than 1 h (up to 3 h), which is not unrealistic (Schumann and Heymsfield, 2016). During this time, the contrail advected from the Hector core by 50 to 100 km distance and might have widened through mixing and wind shear. Note that the

contrail orientations are not perpendicular to the lidar picture and, hence, may appear wider than they are. In particular, the upper layer coincides with computed positions for both the lower and upper edges of the contrails (red and blue triangles). Most coincidences occur in the thicker part of the upper layer between 7:37 and 7:40 UTC but some also in the very thin branches a few minutes earlier and later. The age of the plumes is shown in the second panel of Figure 12 and the mean ages and altitudes of the two anvils are listed in Table 5.

Figure 12 includes the color ratio. The color ratio is the ratio between the backscatter coefficients at 532 and 1064 nm wavelengths. It is close to unity in the tropospheric parts of the observed cirrus, suggesting scattering of particles that are larger than the lidar wavelengths (geometric optics regime, observable in the lower and denser parts of the cirrus). The color ratio is clearly above one (near 2) in the optically thinner stratospheric cirrus, suggesting smaller particles in the Rayleigh scattering regime. The color ratio is large in the gaps between the stratospheric cirrus layers likely because of very small

sublimating ice particle in those gaps. The color ratio shows that the cirrus in the stratospheric anvil was composed of ice particles much smaller than in the troposphere.

Figure 12 also includes the optical depth of the cirrus layer, i.e., the vertical integral of the extinction coefficient. Note that multiple scattering by ice crystals, mainly in the forward direction, may cause that the optical depth is underestimated,





by possibly 40 % for large ice particles with effective radii > 100 µm (M. Wirth, pers. comm.). This should be a minor issue for the thin cirrus in the stratosphere with small ice particles. The effective particle sizes for events E1 to E6 (Table 3) are below 26 µm (de Reus et al., 2009). The optical depth of the layer between 7:30 and 7:35 UTC is about 0.03 and therewith just at the edge of visibility by eye. The upper of the two cirrus layers, seen at 7:37 to 7:40 UTC in Figure 12, have

maximum optical depth of about 0.04, the one further below reaches about 0.08.

The total extinction of the stratospheric anvils, listed in Table 5, can be related to contrail formation. The "total extinction" or "extinction-area" EA is a term defined, e.g., in Unterstrasser and Gierens (2010), as the integral of extinction $\varepsilon$ over the cross-section area A of the cloud. Here we compute $EA = \int \tau \, dx$ over horizontal elements $dx$ in flight direction with $\tau = \int \varepsilon \, dz$ integrated over vertical elements $dz$ within the cirrus layer. The EA values have been computed numerically from the

Falcon lidar data separately for the two stratospheric anvil clouds, see Table 5. For an order of magnitude estimate, we compute the number of ice particles $N_{ice}$ in the anvil per cirrus length perpendicular to the lidar plane, and compare it to the number $N_{nv}$ of nv exhaust particles per contrail length that might nucleate contrail ice particles. The value of $N_{ice}$ can be estimated for given EA and particle sizes from $N_{ice} = EA/(Q_{ext} \, \pi \, r_{area}^2)$. This relation comes from the definition of the area radius $r_{area}$ relating the mean ice particle cross-section area $\pi r_{area}^2 = A_{ice}/N_{ice}$ to the total area $A_{ice}$ and to the total number of ice

particles per length $N_{ice}$, see Eq. 7 in Schumann et al. (2011). The total area $A_{ice}$ is related to the mean extinction efficiency $Q_{ext}$ by $EA = Q_{ext} \, A_{ice}$. Assuming $Q_{ext} \approx 2$ and $r_{area} \approx r_{eff} \approx 15$ µm, about as observed for low temperature cirrus (de Reus et al., 2009; Heymsfield et al., 2014), we compute $N_{ice} = 0.45$ to $1.07 \times 10^{12}$ m$^{-1}$. For comparison, the number $N_{nv}$ of nv particles emitted per flight distance is $N_{nv} = f \, m_F \, PEI_{nv}$, where $PEI_{nv}$ is the particle number emission index (see Appendix) and $f$ (1 to 5) the number of contrail parts crossed along the curtain. The fuel consumption per flight distance $m_F$ is about 1.2 g m$^{-1}$ at

cruise (~846 kg h$^{-1}$, Weigel et al. (2009)). The particle emission index was estimated as $PEI_{nv} = 2.6 \times 10^{15}$ kg$^{-1}$. This implies $N_{nv} = 3$ to $16 \times 10^{12}$ m$^{-1}$. The number of nv particles per flight distance is 3 to 35 times larger than the number of ice particles per cirrus length. Hence, the amount of exhaust particles is large enough to possibly explain the lidar-observed cirrus layers, in particular the upper one, by contrail-cirrus.

### 3.2.4 CPOL radar observations

The CPOL radar provides information about the Hector cloud morphology and microphysics in space and time based on radar reflectivity at 5.4-cm wavelength (May and Keenan, 2005). For monodispersed water droplets of diameter d and concentration n, the reflectivity factor is $Z_R = n \, d^6$. For ice particles, the $Z_R$ value is ~0.2 times smaller because of lower refractive index but may be larger because of nonspherical shapes. Often one uses a logarithmic ratio Z of $Z_R$ relative to the reflectivity $Z_0$ of a raindrop with a diameter of 1 mm in decibel (dBZ) (Hagen et al., 2012), so that $Z_R = Z_0 \, 10^{[(Z/dBZ)/10]}$.

Values of Z above 40 dBZ indicate heavy precipitation. The reflectivity in cirrus anvils is often below 5 dBZ. A stratiform anvil near the tropopause and parts of the non-precipitating anvil clouds should be detectable, except for thinner cirrus layers detrained from the main convective system. The signal-to-noise ratio decreases with distance, here up to about 120 km. For





weak reflectivity from large distance, the radar patterns plotted are sensitive to interpolation in the gridded data. In a few cases, the radar measured reflection from the aircraft structure (see Supplement). Contrails cannot be detected by the radar.

Figure 13 (bottom) shows the Geophysica flight path overlaid on the radar signal for the time interval including events E7 to E10. In the vertical curtain along the flight (lower panel) we see that Hector induces Z exceeding 40 dBZ in the

troposphere up to about 14 km above ground. Between 14 and 18 km, Z is lower (<15 dBZ), and a single faint cloud part (5 dBZ) is found at 20 km height, i.e., 2.6 km above the tropopause. The aircraft flew near 18 km height, above cloud parts with high reflectivity, but essentially in air masses with Z near 5 dBZ, so possibly in anvil cirrus. The data show that Hector is a mesoscale convective system with many updrafts and downdrafts (Huntrieser et al., 2009). The stronger updrafts typically have a diameter of 10 to 20 km, as visible in the horizontal cross-section (upper panel).

In further such images at other times and heights (available by ftp, see Supplement) one finds Z values exceeding 30 dBZ above 17 km at the eastern edge of the Tiwi Islands after 4:45 UTC. Event E0 occurred in air with low Z above a strong tropospheric updraft. Event E1 occurred in a forward (upshear-tilted) anvil produced by a strongly overshooting cell (still 30 dBZ at 18 km) located to the north-west of the E1 location. The convection was most active during the times of E1 to E3. During E2 and E3 (Figure 13, top), the Geophysica flew within about 5 km distance of a strong updraft with reflectivity

reaching 30 dBZ. The updrafts were far weaker for E4 to E6. E4 and E5 occur in the remainders of clouds above dissipating convection. Event E9, for which only few ice particles were sampled in situ, is indeed an ice event close to a fresh overshooting cell near 10 dBZ. The Geophysica never passed through the center of an updraft but sampled the uppermost parts of convective clouds.

When comparing to the lidar results, it becomes obvious that clouds with high radar reflectivity occurred below 15 km

altitude, not visible to MAL, as expected. The strong cloud backscatter seen in the MAL data (**Figure 11**) during the dive between 25000 s and 26200 s come from clouds below the flight level during a flight leg towards a turning point outside the Hector cloud. The optically thin anvil clouds seen by the Falcon lidar, which were observed while the Falcon flew south of strong updrafts, are not detected in the CPOL data.

The exponent of the relationship between Z and IWC, IWC $\sim Z_R{}^b$, for the C-band is qualitatively similar to those for

other radar wavelengths, with b in the range 0.55 to 0.65 (Liu and Illingworth, 2000; Hogan et al., 2006; Protat et al., 2016). The CPOL reflectivity for E2 is between 5 and 10 dBZ, which translates into IWC of 0.76-1.5 mg m$^{-3}$ at T = -87.1°C, using the Hogan et al. (2006) temperature-dependent relationship. This is in good agreement with the in-situ measurements (Table 3). Relating $Z_R$ to Z as explained above, one finds that the IWC at 30 dBZ is a factor of 30±10 larger than at 5 dBZ. Hence, the IWC in the overshooting clouds was more than a factor of 20 higher than measured in situ at the same heights in the

neighborhood.



### 3.2.5 NOAA-AVHRR satellite observations above Hector

Satellite data provide information on Hector and its anvil. The AVHRR data have better spatial resolution than the geostationary satellite data discussed in previous SCOUT-O3 papers (Brunner et al., 2009; Chemel et al., 2009; Frey et al., 2014). NOAA-15 and -12 Hector-overpasses occurred at 5:16 and 7:43 UTC. The two satellites observed Hector in its early

active phase (4:47 to 7:00 UTC, 14:17 -16:30 LT) and mature phase (Frey et al., 2014). The image for the first overpass is available in Figure S8. Figure 14 shows the image for the second overpass, nearly simultaneously with the Falcon lidar observation of the southern anvil. The figure includes the flight path of the Geophysica until the satellite-overpass times. The general pattern is consistent with model studies (Frey et al., 2015), and also with the ice event locations measured in situ, see Figure 7, and the radar data. The highest and thickest part of Hector occurs first over the eastern and later over the western

parts of the Tiwi Islands. The low-temperature convective core of Hector extends over about 80 and 140 km in east-west and 50 and 60 km in north-south direction, and the BT minima are 180.1 K and 184.2 K at 12 μm and 180.1 and 185.6 K at 10.8 μm, in the two images. Local overshoots are detectable as small cool spots of a few AVHRR pixels in the BT results when adapting the color scales to a narrow temperature range from 183 to 205 K. They are better seen in the radar signals, where we see cloud tops up to 20 km. The data indicate cloud top temperatures below -93°C and -89°C, 6 and 2 K cooler than the

cold-point tropopause. For an adiabatic lapse rate, which is close to the dry lapse rate $g/c_p$ (g = gravity) at these temperatures, the temperature differences suggest overshooting by about 620 m and 200 m above the tropopause (Bedka et al., 2010), up to 18.0 and 17.6 km altitude, at these times. These altitudes are 600 to 700 m smaller than the cloud tops observed by lidar, see Figure 11 and Figure 12, and possibly 1000 m lower than clouds tops from radar. Hence, there were optically thin cirrus clouds, including contrails, above the optically thicker cloud parts, besides spotty convective towers. The radar signals

indicate about 2 km lower cloud tops at the second overpass time compared to the first one.

We find positive BTD values outside the optically thick Hector cloud on the southern (leeward) side, likely from the anvil cirrus with small ice particles. Figure 14 includes the computed positions of the potential contrail for the time since the aircraft reached the stratosphere (cyan curve). The Geophysica is descending and returning towards Darwin after having surrounded the north-west corner of the Hector core. However, line-shaped cloud structures, possibly from Geophysica

contrails in the anvil cirrus, could not be detected in the BTD signals. They may be too thin optically to be detectable this way. Also, when there are clouds below, they would overwhelm the signal, leaving the contrails undetected.

Hector is not alone. Deep convection over Northern Australia has often been observed in radar data (Hassim et al., 2014), and can also be seen for this event east and south of Hector in the satellite images. An analysis of backward trajectories for the time period between start of convection (assumed at 3 UTC to allow for wind underestimates) and time of

peak aerosol measurements (Figure S8) shows that the convective tower east of Hector did not provide the high aerosol values which were measured with COPAS over Hector.



Interestingly, we also see a band of slightly negative BTD values along the northern edge of the Hector cloud which is not present on the southern side. An explanation of this pattern requires a more detailed study. Such a study should include different absorption by optically thin clouds with different particle sizes (Inoue, 1985), wind-driven or convective uplift possibly causing pileus clouds (Garrett et al., 2006), presence of nitric acid which has an 11-μm absorption band (Chepfer et al., 2007), the satellite viewing angles, and three-dimensional radiation transfer.

### 3.2.6 Microwave Temperature Profiling of Hector from Geophysica

The MTP measures the temperature profile ahead of the aircraft from which one can derive isentropes along the flight path (Denning et al., 1989). The MTP instrument averages along line of sight segments of a few kilometer lengths. Figure 15 shows the tropopause and the isentrope altitudes, measured ±2 km above or below aircraft. The plot versus longitude identifies the flow pattern over the convective system. The mean wind comes from the east in the stratosphere. We see systematic waves on the western downwind side, supporting the reproducibility of the measurements. The variance $z'$ of the isentropes varies from 200 m at 370 K to 90 m at 430 K; far more than for the 16 November flight (Table 2). The variances reflect the convective activity in Hector decreasing with altitude. The isentropes are higher (by about 100 to 50 m, decreasing with height) over the center than at the boundaries of Hector. Hence, we find a systematic flow pattern with stratospheric air coming from the north-east passing over and around the convective core of the Hector cloud as if it forms an obstacle with lee waves downwind. Air from overshooting convection, after cooling by ice sublimation may contribute to this flow. Large vertical motions occur in particular near the tropopause.

## 4 Discussion

### 4.1 Implications of the long-lived contrail photographed in the stratosphere

The observation that a long-lived contrail formed downstream of the dissipating Hector of 16 November at the tropopause with ice saturation is not surprising, but the fact that we see 50 min old remnants of a contrail which survived in dry air (Table 2: 3 μmol mol$^{-1}$ $H_2O$, 65 % RHi) at altitudes up to 18.6 km in the stratosphere is noteworthy. One may ask whether the formation of nitric acid trihydrate (NAT) or other nitric acid hydrates is essential to explain the long lifetime. The ambient temperature must be below $T_{NAT}$ to allow for formation of NAT (Iannarelli and Rossi, 2016) or "Δ-ice", which is a stable $HNO_3/H_2O$ mixture proposed to explain observed humidity in low-temperature contrail and cirrus clouds (Gao et al., 2004; Gao et al., 2016). One may also ask whether the fact that the uppermost plume no longer contains a visible contrail is consistent with our understanding of contrail formation and sublimation. Here, we show that the long lifetimes of both the lower and upper contrails in the quiet stratosphere can be explained by the contribution from engine exhaust water for low turbulent mixing. Formation of NAT is not necessary but provides another approach to explain the long contrail lifetime.





Further we will show that sublimation of the invisible contrail part is consistent with standard mixing concepts both with and without NAT.

### 4.1.1 Classical mixing concept for low turbulence

First we characterize mixing. The observed width W ≈ 3 km of the contrail caused by differential lateral advection of the upper and lower edges of the wake with vertical separation D ≈ 180 m over its age ≈ 3000 s indicates a wind shear of S=W/(age D) ≈ 0.006 s$^{-1}$. Wind shear might excite turbulence if the Richardson number Ri = $N_{BV}^2/S^2$ is low (Turner, 1973). Here, strong stratification ($N_{BV}$ ≈ 0.03 s$^{-1}$), apparently larger than the shear, causes large Ri = 30, and may have prevented turbulence. The level of turbulence in the ambient air is unknown, but the MTP data indicate small variability z' of the isentrope altitudes at horizontal scales > 10 km in this time period (see Table 2), implying small vertical gravity wave velocities w' ≈ z' $N_{BV}$, of 6.5 and 19 cm s$^{-1}$ for the upper and lower contrail altitude ranges, without mixing by breaking waves. The contrail formed downstream of a weak and dissipating Hector cloud (Brunner et al., 2009). Hence, turbulence was weak in the stratosphere in this case, and this could explain why parts of the contrail experience little dilution.

The classical (Schmidt-Appleman) thermodynamic contrail formation process is illustrated in Figure 16, adapted from Schumann (1996) for the stratospheric conditions. The mixing line in this plot represents conditions in the plume varying from engine exit conditions (high temperature and high humidity at the right end of the mixing line) to ambient air conditions (temperature and partial water pressure at point E) with increasing dilution $N_{dil}$, if without phase change. The mixing-line gradient G = $c_p$ p $EI_{H2O}$ ($M_{air}/M_{H2O}$)/[$Q_c$(1-η)] depends on pressure p, water emission index $EI_{H2O}$, molar mass ratio ($M_{air}/M_{H2O}$) ≈ 29/18, fuel combustion heat $Q_C$ = 43.2 MJ/kg, overall propulsion efficiency (η ≈ 0.1 for this descending flight section), and on the specific heat capacity $c_p$ = 1004 J/(kg K). G is smaller for the upper contrail because of lower pressure. The ambient temperature is far below the threshold temperature for liquid contrail formation, LC, with maximum liquid humidity in the plume, as listed in Table 2. There is no doubt that contrails formed during the whole Geophysica flight above about 13 km altitude, even for zero ambient humidity.

The point I2, at the lower-left end of the mixing line, close to ambient conditions E, is defined by the second intersection between the mixing line and the ice saturation curve. I2 is reached for a dilution

$$N_{dil,I2} = Q_C (1-η)/(c_p (T_{I2}-T_E)),\qquad(5)$$

where $T_{I2}$, the temperature at point I2, is computed according to the given definition, and $T_E$ is the given temperature in the environment. For given G and measured RHi, we compute $N_{dil,I2}$ = 1.0×10$^6$ for the upper and 5.2×10$^6$ for the lower contrail. For fully dry air (RHi = 0) the value of $N_{dil,I2}$ is lower: 0.35×10$^6$ and 0.68×10$^6$.

For orientation, we note that the dilution of the Geophysica exhaust plume is first caused by the aircraft-induced turbulence, and reaches a value of about $N_{dil}$ = ρA/$m_F$ ≈ 0.5×10$^6$, when we assume that the exhaust from burning fuel at mass





flow rate $m_F$ per flight distance ($m_F \approx 1$ g m$^{-1}$, an estimate for slow descent) gets uniformly mixed over the plume. Here, density $\rho$ is known, and the cross-section A of the young wake vortex may be estimated from A = 3 s$^2$ (Greene, 1986; Schumann et al., 2013b), where s = 37.5 m is the wing span of the Geophysica. Hence, the contrail survives wake dilution since the dilution at the end of the wake phase is too low to dilute the humidity in the exhaust plume below ice saturation.

For known dilution $N_{dil,I2}$, one may estimate the corresponding contrail age $t_{I2}$ by inverting the dilution law, Eq. (4), $N_{dil,I2} = N_{dil}(t_{I2})$. Of course, the result depends on this dilution law. Moreover, the result depends strongly on the assumed ambient humidity RHi. Here, we find that the exhaust stays ice supersaturated until ages of 140 to 2400 s for RHi from 0 to 0.9, and stays longer only for nearly ice-saturated ambient conditions with RHi > 90 %, see Figure 16.

     This concept explains why the lower contrail persists until an age of about 1680 s for RHi < 87 % as measured in the
lower stratospheric part of this flight. However, it does not explain nearly 1 h age for the upper contrail for the assumed dilution law. It suggests that mixing in the stratosphere is slower than suggested by Eq. (4). In fact, for a factor 0.25 reduced dilution (thin curves), the lifetime can be explained also for the upper contrail with RHi about 64 %.

     Future numerical simulations of the contrail dynamics, including the ice microphysics and turbulent mixing in the aircraft wake, should overcome limitations of this thermodynamic analysis. The mixing concept assumes that the
sublimation time scale $t_{subl} \approx (4\pi D_{H2O} n_{ice} r)^{-1}$ is small compared to the plume age. The $t_{subl}$ depends on water vapor diffusivity $D_{H2O}$ in air, and on the number $n_{ice}$ and size r of the ice particles (Korolev and Mazin, 2003). For constant pressure, the water vapor diffusivity is nearly 50 % smaller at -87°C ($\sim 1.5\times10^{-4}$ m$^2$ s$^{-1}$) than at -50°C (Pruppacher and Klett, 2010), but the particle concentration is high. If each nv particle nucleates an ice particle, the ice number concentration would be $n_{ice} \approx \rho$ PEI$_{nv}$/$N_{dil}(t)$ in the plume at age t, and high because of the large particle emission index PEI$_{nv}$. Therefore, the time
scale of sublimation stays far below the plume age, even for a radius r = 0.1 μm, a limit of visibility (see Supplement). For small ice particles, the dependency of the equilibrium vapor pressure on the ice surface tension (the ''Kelvin effect'') should also be taken into account (Lewellen, 2014). It would contribute to quick sublimation of the smallest ice particles.

**4.1.2 Ice formation concept including the possibility of nitric acid hydrates formation**

**Figure 17** shows the change of mean temperatures in the contrail plume with time. Consistent with the Schmidt-Appleman
mixing concept and dilution definition (Schumann et al., 1998), the plume temperature and the molar water vapor and nitric acid mixing ratios

$$T(t) = T_E + Q_C(1-\eta)/[c_p N_{dil}(t)], \tag{6}$$

$$[H_2O](t) = [H_2O]_E + EI_{H2O} (M_{air}/M_{H2O})/N_{dil}(t), \tag{7}$$

$$[HNO_3](t) = [HNO_3]_E + EI_{HNO3} (M_{air}/M_{HNO3})/N_{dil}(t), \tag{8}$$





exceed ambient conditions (E) because of the combustion emissions. We do not know the $HNO_3$ emission index $EI_{HNO3}$ for the Geophysica. Previous $EI_{HNO3}$ measurements behind commercial aircraft at cruise range from 0.06 to 0.8 g kg$^{-1}$ (Arnold et al., 1992; Schumann et al., 2000). For a lower bound estimate of NAT contributions, we assume $EI_{HNO3}$ = 0.06 g kg$^{-1}$. The frost point and the NAT existence temperatures, $T_{ICE}$ and $T_{NAT}$, are calculated for given mixing ratios and ambient pressure

inverting known relationships (Hanson and Mauersberger, 1988). These temperatures approach the values listed in Table 2 after infinite dilution.

The plots show the results for high ($N_{dil}$ from Eq. 4) and for low (same $N_{dil}$ reduced by factor 0.25) dilution. We see that the plume temperature starts high because of combustion heat. The curves show the early time period of contrail formation, when the plume temperature sinks below the dew point, after about 0.1 to 1 s plume age, and the late time period of

sublimation when the plume temperature exceeds ice equilibrium temperatures at times larger than several minutes. The curves represent the results for the three plume parts as listed in Table 2. Part 0: the invisible plume, part 1: the upper contrail, and part 2: the lower contrail. For part 2, we see that the plume temperature stays below the frost-point, regardless of which of the dilution laws is used. For part 1, we see the plume temperature stays below the frost point only for low dilution; alternatively the plume temperature stays below $T_{NAT}$; hence, the upper contrail can be explained either by low

dilution, or by formation of $HNO_3$ hydrates, or by NAT hindering the sublimation of ice particles. For part 0, we see that the plume temperature stays above $T_{ICE}$ and $T_{NAT}$ for the given age, regardless of the dilution law. Hence, the early sublimation of the contrail that formed in this flight segment, leaving behind the invisible plume, is consistent with the contrail formation concept regardless of the levels of turbulence and NAT formation.

**4.2 Arguments for stratospheric Hector cirrus being caused by contrails**

The observations during the 30 November over the Tiwi Islands show cirrus at altitudes of 17.2 to 18.5 km, clearly above the tropopause, in particular over the Hector cloud system. Here we collect the arguments which indicate that some of these cirrus clouds could be contrail cirrus induced or affected by the Geophysica.

Peak E7, at 6:00 UTC, is likely caused by a contrail. This can be concluded from Figure 10. It is also supported by the radar data. Looking at successive CPOL images, we see no overshooting convective cell in the area of event E7 in the last 30

min. But the event is of short duration (~15 s) and contains only little ice. The six ice events of de Reus et al. (2009), partially with far higher IWC, occur at positions, as identified in Table 4, which could be potential plume encounters, except for E2. Event 6 may be even a superposition from four Geophysica contrails. There is one further, shorter ice event (E10), which is also in the range of a potential plume encounter. This is shown by the plume advection analysis and the scatter plots for these events. A counterargument against contrails could be the fact that large ice particles were found for several events

with CIP. If this is the key argument, then at least events E4 and E10, which are free of CIP counts, could be contrails, and the others could still be a mixture of contrails and natural cirrus.





The MAL lidar observations, Figure 11, shows cloud patterns above 17 km altitude with shape and low optical depth as expected for contrails. Many of these clouds are collocated with computed potential contrail positions. Also the Falcon lidar observations, Figure 12, of cirrus above the tropopause, show cirrus patterns consistent with Geophysica contrails. The width and geometrical depth, the optical depth of the upper anvil cirrus, and the total extinction of the stratospheric anvils could be
explained with contrail properties, see Section 3.2.3.

The properties of the in-situ observed ice clouds (Table 3) are generally consistent with other measurements in contrail cirrus. Figure 18 shows previous contrail data (Schumann and Heymsfield, 2016). This plot includes the SCOUT-O3 results as if they were contrail cirrus. The data extend the available information for long-lived contrails and low temperatures. The Geophysica data are generally within the large range of variability found for other measured and modelled contrails. The
measured IWC for the contrail event E7 is low because of low ambient humidity. IWC values for events E1 and E2, which contain contributions from convection, are high, but when plotted versus temperature, Figure 19, not out of a reasonable range and not far from the result of Gao et al. (2006) for $T \approx 197$ K.

All measured cirrus clouds in these events coexist with high nv aerosol concentrations. The magnitude of the aerosol concentrations is in the range as expected from previously determined dilution for given plume age and for the emission
index of nv particles (see Appendix). No nv aerosol peak is measured before the Geophysica had the chance to penetrate its own plume. Hence, the aerosol may come from the engine exhaust. The only piece of cirrus which must be free of aircraft influence (E0) shows no nv aerosol. This supports the assumption that the nv aerosol comes from the exhaust. Further peaks of nv aerosols occur at places all coinciding with potential plume positions. The others are peaks without ice, in particular on the north-east side of Hector, which could be expected if the air arriving with weak wind from the north-east was too dry to
let contrails persist over the plume age. Also, aerosol peaks measured in other flights (Weigel et al., 2009) can be explained by engine exhaust (Table S1).

The nv particle concentration is highly variable along the Geophysica flight path. This suggests that the aerosol stems from a recent source with ages too small for more uniform mixing. The maximum aerosol concentration in the cirrus is higher than measured anywhere else during that day (Allen et al., 2008; de Reus et al., 2009). The maximum is found on the
windward side in events E7 and E8. The equivalent potential temperature near the surface was below 365 K based on radiosondes. The aerosol was found above 400 K at the windward side. Hence, Hector was not strong enough to bring boundary layer aerosol locally from the Tiwi Islands to the positions with large aerosol concentrations.

Alternative sources may exist but are not obvious. The 10-d backward trajectories show that air masses reaching the flight path above the Tiwi Islands on 30 November 2005 came from the west (troposphere) or the east (stratosphere) after
having passed Northern Australia (Brunner et al., 2009; Heyes et al., 2009), a potential source of dust or biomass burning aerosol (Allen et al., 2008; Heyes et al., 2009). Small filaments or pockets of polluted air may have been transported from the tropopause aerosol layer upwards within overshooting and mixing convection to places where the Geophysica was





measuring. The convective transport may have occurred upstream during recent days, but it would require very low shear and low diffusivities to explain survival of filaments over a day so compact as measured. We have no indications for contributions from volcanic aerosol emissions for Hector this day (Kremser et al., 2016).

The ratio of ice crystals to nv aerosol particle number concentrations is low, but was found to be much higher in the
mature stage of Hector (after 5:30 UTC) in this altitude range than elsewhere (Frey et al., 2014). In fact, the highest cloud to aerosol particle ratio among all stages of cloud evolution was found here for reasons not understood (Chemel et al., 2009; Frey et al., 2014). If the ice particles were formed from exhaust aerosol in contrails, then this could explain relatively large ice particle concentrations in this cloud.

The alternative explanation of the ice events by overshooting convection is not clearly supported for all events by
tracer-humidity or tracer-aerosol correlations. As discussed in Section 3.2.1, in the period of flight above Hector, between 5 and 7 UTC, including events E1 to E6, the measurements show lower $CO_2$ and CO than in the troposphere.

### 4.3 Arguments for Hector cirrus being caused by cloud dynamics

As discussed in the Introduction, several observation and model studies support the view that the observed cirrus was formed by overshooting convection. Here we collect the findings and arguments which indicate that the observed cirrus clouds were
caused by convection and only partially affected by contrails.

Previous radar observations and model studies have shown that Hector clouds reached above 18.5 km altitude on 30 November 2005 (Vaughan et al., 2008; Chemel et al., 2009). The C-POL radar reported frequent overshoots from 4:30 UTC, i.e. at times before the Geophysica reached the region (Frey et al., 2015). The radar observed overshoots reached higher than the Geophysica flight path and contributed to large ice particles and high humidity at that position. Besides vertical mixing
also transient vertical motions and waves in the stably stratified atmosphere contribute to θ being variable along the flight path.

Without mixing or other heat exchange, large changes in potential temperature at constant flight level may be explained by converting kinetic energy into potential energy for large updraft velocities of order w= $\Delta z$ $N_{BV}$ (Turner, 1973). The required vertical motions are possible for maximum updraft velocities of 10 m s$^{-1}$, which appear not unrealistic (Chemel et
al., 2009; Dauhut et al., 2015; Frey et al., 2015). Latent heat (L) contributes little to changes of temperature at these low temperatures because of low water saturation mixing ratio, $q_{sat}$ $L/(c_p$ T) << 1. Cloud cores with strong updrafts are often surrounded by a ring of downward motion explaining strong downdrafts and measured and simulated CO and $O_3$ analysis indicate considerable vertical mixing with downward transport of stratosphere air (Frey et al., 2015). Hence, overshooting convection with vertical mixing has occurred. Still, overshooting convection may also carry contrail ice water upwards and
exhaust particles upwards and downwards.





For the potential contrail event E7, observed at 6:00 UTC with computed lifetime of 50 min, the radar images show that there was some overshooting at 5:15 UTC which may be the source of water vapor causing the long lifetime, and even may have contributed to the ice particles measured in the slightly ice-subsaturated ambience. The high concentration of large ice particles for events E1, E2, E6 (and just one particle for event E3), measured by CIP, with maximum diameters exceeding 100 µm, can hardly be understood from contrails (Jeßberger et al., 2013). Also the IWC measured is far larger than what could be explained by local phase change possibly triggered by contrails, even if the contrails formed in air initially at liquid saturation. The large particles may originate from lower cloud parts and also from the thin tropopause cirrus in the neighborhood of Hector. Subvisible cirrus with large ice particles near the tropical tropopause was observed also elsewhere and simulated in models (Lawson et al., 2008; Davis et al., 2010; Heymsfield et al., 2014; Zhou et al., 2016).

Event E2, with the most negative $\Delta\theta$ value, was likely caused by Hector, because of a strong convective updraft. Further, we observed one short segment (event E0) with cirrus in pristine air (without increases in nv aerosol) at 18.2 km altitude. This event was not impacted by the Geophysica. Hence, this short period of ice cirrus was caused by Hector. It remains open whether this short event is an effect of overshooting convection or of other cirrus formation (such as a pileus clouds).

The geometrical scales of the stratospheric anvil clouds observed by lidar (but not by radar) are at the upper limit of what could be explained by contrail cirrus (Schumann et al., 2016). Similar stratospheric cirrus was observed also elsewhere, without aircraft causes (Iwasaki et al., 2015).

If one looks at the lidar data only, the structure of the Hector cloud may be similar to the overshooting towers in and above the tropical tropopause layer as sketched in Fig. 8 of Vernier et al. (2011). The MAL observations suggest that the Geophysica flew around the deepest convective towers, mostly above the Hector anvil, and occasionally in convective cloud elements. The CPOL data, however, show a mesoscale convective system with many updrafts below the anvil.

### 4.4 Implications for contrail properties

Observations of contrails at low atmospheric temperatures are of interest because of potential impact on high ice-particle concentrations in contrails (Kärcher and Yu, 2009), and because high relative humidity over ice has been observed at low temperatures in nitic acid containing contrails (Gao et al., 2004).

The humidity measured in ice clouds above the tropopause was mostly near or below ice saturation (Table 3 and Figure 8d). Hence, there is no indication for systematic enhancement of relative humidity with respect to ice saturation in the cold cirrus. But the data do not exclude nitric acid hydrate formation in the contrail. Higher nitric acid hydrates may exist at temperatures > 3 K above the frost point temperature, in particular for high $HNO_3$ and $H_2O$ emissions; and nitric acid hydrates in the contrail ice particles could have contributed to reduced sublimation rates (Iannarelli and Rossi, 2016) and hence long lifetimes of the photographed contrail .





Figure 18, discussed with respect to IWC before, shows that the measured ice number concentration is about a factor 0.01 below the median at similar contrail ages. The volume and effective radius values are similar. In fact, large ice particles in contrails mixed with other cirrus were also seen in other measurements (Kübbeler et al., 2011). Table 3 includes mean values of non-volatile aerosol concentration increases above background, $\Delta n_{nv}$. The ice particle concentrations are a factor of

0.003 ($10^{-5}$ to 0.008) smaller than the concentration of nv aerosol. For cirrus, this would imply that the aerosol contained few particles which act as ice nuclei. Taking the $PEI_{nv}$ as reference, we find an apparent emission index for ice $PEI_{ice}$ of order $10^{13}$ kg$^{-1}$, about 100 times less than for other contrail measurements (Schumann et al., 2016).

The magnitude of $PEI_{ice}$ to be expected at the time of measurements can be estimated also from the size of the contrail particles and the visibility of the contrails. Visibility requires sufficient optical depth, which increases with the geometrical

depth of the contrail, the number of ice particles per volume, the mean square of the ice particle sizes and their extinction efficiency. If many ice particles form which share in the given amount of water in the contrail, the particles are small. For sizes smaller than the wavelength of light, their extinction efficiency decreases strongly (van de Hulst, 1957). Hence, for quasi-spherical ice particles and fixed ice water path, the optical depth reaches its maximum for a radius of 0.41 μm (Figure S11). The particle size scales with the third root of the apparent particle emission index $PEI_{ice}$ (Schumann and Heymsfield,

2016). For the upper contrail in Table 2, one computes that $PEI_{ice}$ must be of the order of $10^{12}$ kg$^{-1}$ to reach maximum optical depth (Figures S10 and S11). Hence, consistent with the measurements, the "effective" emission index is far smaller than the emission index for nv particles. Otherwise the particles would be too small to be visible. Effective means, $PEI_{ice}$ at the time of observations. The more crystals form initially, the smaller is the fraction of ice crystals surviving sublimation during the wake vortex phase (Unterstrasser and Sölch, 2010). Even if the number of ice particles generated earlier during contrail

formation is larger than $PEI_{nv}$, many of the smaller ones must have sublimated in ice sub-saturated air leaving their water for the remaining ones during the time until they get observed or measured, otherwise the remaining particles would be too small to be measureable.

Alternatively, the low number of small ice particles measured may be a consequence of the instruments used. The FSSP100 is suited for detection of particles larger than 2.7 μm. Far higher ice particle concentrations, of about 50 cm$^{-3}$ after

20 min age, were measured in the contrail of the NASA-WB-57, above ice saturation at low temperatures at the tropical tropopause over Florida and South of Costa Rica (Flores et al., 2006; Gao et al., 2006). The instruments used on the WB-57 resolved ice particle sizes > 0.5 μm and most contrail ice particles were found smaller than 2 μm. On the other hand, also the MAS backscatter signals, the thin backscatter clouds of contrail shape in the MAL lidar images, and the low optical depth of DIAL observed potential contrail-cirrus suggest low ice number concentrations.





## 5 Conclusions

Properties of long-lived contrails of the Geophysica aircraft were determined for low temperature at high flight altitudes. Geophysica measurements from the SCOUT-O3 field experiment were analyzed near deep convection in the tropics, in particular for Hector over the Tiwi Islands near Darwin. Photos and related in-situ, satellite and profiler data provide

properties of an isolated contrail that formed downwind of Hector in the lower stratosphere at temperatures down to -88°C. Cirrus and contrails above Hector were studied with the in-situ, lidar and profiler data from the Geophysica, together with lidar data from the Falcon aircraft, and NOAA-AVHRR satellite and CPOL radar data to characterize the atmosphere around the in-situ observations. For interpretation of the aerosol data, the emission index of non-volatile particles from the Geophysica was determined using data from the RECONCILE self-match experiment in the polar stratosphere. We found

that the aerosol and cirrus measured during the Hector day in overshooting convection, were mainly a result of engine exhaust aerosol and a mixture of contrails and natural cirrus.

The Geophysica contrail observed in cloud-free and ice-subsaturated air downwind of the dissipating Hector on 16 November 2005 formed from the water emitted with the exhaust and stayed in the lower stratosphere at temperatures between -82 and -88°C with a lifetime of nearly 1 h. The long lifetime can be understood without the need to invoke

previously suggested mechanisms for reduced sublimation rates. Instead it is explained here by low dilution rates, because of low ambient turbulence in the highly stratified and weakly sheared stratosphere, with low wave activity downstream the dissipating Hector. For RHi below but near 100 %, any humidity increase from convection contributes strongly to enhanced lifetimes of contrails (and cirrus). Nitric acid hydrates formation in the contrails at low temperatures near the tropopause, with high water vapor and high nitric acid concentrations from engine emissions, could also contribute to long contrail

lifetimes. An interesting band of low BTD is seen on the windward side of the Hector, possibly because of infrared radiation absorption by ice containing nitric acid or because of pileus cloud formation.

Several arguments pro and contra aircraft impact on the aerosol and cirrus were discussed for the Hector in-situ measurements of 30 November 2005. In particular the high nonvolatile aerosol is best explained with aircraft emissions. Some ice events likely originated from ice and humidity advected by overshooting convection, others were likely of

contrail/exhaust origin, but several events could have been impacted by both, see Table 4. The Geophysica measured mainly above the tropospheric anvil and occasionally near the overshooting convective towers. The large ice particles found in a few events resulted in one case (E2) from a strong updraft, in other cases (e.g., E3) from ice advected from nearby convective towers. The overshooting convection likely has contributed the humidity needed for long lifetimes of contrails. The MTP data show that Hector interacts with ambient flow like an obstacle, with lee waves. The stratospheric anvil formed two layers

of 600 to 1200 m depth and 25-40 km width with optical depth < 0.08. The lidar color ratio shows that the stratospheric anvil was composed of ice particles much smaller than typically in the troposphere. In spite of the large geometrical dimensions, the optical depth and the total number of ice particles can be explained by contrail cirrus, at least in the upper part of the stratospheric anvil. Even if the majority of the cirrus observed this day above the tropopause originated from Hector cloud





dynamics, the cirrus properties for the six ice events listed by de Reus et al. (2009) may also be interpreted as contrail-cirrus properties in the lowermost stratosphere under overshooting convection conditions.

Potential contrail sampling was analyzed with the same methods for all Geophysica flights for SCOUT-O3 and also for the TROCCINOX field experiment in Brazil. Here we summarize the results without reporting details. During SCOUT-O3, several plume encounters were found for the three flights of 29 and 30 November 2005, by far most of them during the Hector morning flight. Some potential plume encounters were computed also for 16, 19 and 25 November 2005, and for the TROCCINOX flights of 4 and 5 February 2005. No contrail encounters were found above 380 K potential temperature during these flights, except for the Hector morning flight. The latter contributed most of the high IWC data above the tropopause at 375 K and up to 425 K, with IWC exceeding 1 mg m$^{-3}$ only below 410 K potential temperature (Corti et al., 2008). However, the radar observations indicate that the IWC in strong convective events was at least a factor of 20 higher than measured in situ along the flight path at the same height. Besides Hector also its neighbors over Northern Australia reach often similar altitudes.

The ice particle concentrations measured in the clouds at low temperatures were always far smaller than the concentrations of nonvolatile aerosol, perhaps because of instrument size limitations and frequent quick sublimation losses of small ice particles. There is no indication that the number of ice particles in low-temperature contrails is higher than what is expected for aircraft with moderately high nonvolatile particle emissions. A higher initial particle formation results in quicker sublimation and larger particle losses, so that the initial PEI$_{ice}$ value loses importance. The measurements show no systematically enhanced relative humidity in cold contrails or cirrus. Often, the cirrus was observed below ice-saturation.

The Hector measurements during the Golden Day of SCOUT-O3 are unique. We do not know of comparable in-situ measurements combined with such a variety of remote sensing data elsewhere. The long-lived contrail observed in one case in a photo and potential contributions from long-lived contrails in the in-situ data suggest that contrails may survive in ice-subsaturated air longer than expected so far, at least for weak turbulence conditions. The data may be useful to constrain ice sublimation models and to assess climate effects of potential future increases of air traffic in the lower stratosphere. Future measurements should include instruments to sample ice particles below 1 μm in size. The analysis shows that the aerosol and cirrus measured with instruments on the Geophysica aircraft in Hector was significantly affected by its own exhaust and contrail contributions. Nevertheless, the combined analyses of in-situ and remote sensing data confirm that deep overshooting convection contributes to hydration of the tropical stratosphere, here up to about 20 km altitude or 2.6 km above the tropopause. The data or future measurements may allow determining hydration by comparing profiles measured upstream and downstream of the cloud systems. Also, a more detailed analysis of the combined multi-aircraft in-situ, lidar and satellite observations of subvisible cirrus below the tropopause in regions affected by outflow from deep convection may extend the present understanding of the origin and importance of such aerosol and cirrus for dehydration of the tropical tropopause region.





*Data availability.* The data are available as explained in Section 2. Revised and added data are available on request from the authors.

**Appendix: Exhaust emission indices from plume encounter during RECONCILE**

Gaseous and particulate emission indices EI and PEI, i.e., the mass of an exhaust species or the number of exhaust particles

per mass of fuel burned, need to be known to estimate any emission contribution to mass-specific concentrations in the plume, $\Delta c = PEI/N_{dil}(age)$, for given plume dilution $N_{dil}(age)$ (Schumann et al., 1998). The emission index $EI_x$ of an exhaust emission x can be determined from

$$EI_x = EI_r \, \Delta c_x / \Delta c_r \qquad\qquad (A1)$$

for known mass-specific concentrations $\Delta c_x$ and $\Delta c_r$ of the species x and a reference species r in the exhaust plume above

ambience (Zheng et al., 1994; Schulte et al., 1997), assuming both are conservative exhaust tracers and mix similarly. Suitable reference species with known emission indices are $CO_2$ and $H_2O$, $EI_{CO2}=3.15$, $EI_{H2O}=1.24$ for the given fuel.

Here, we analyze concentration peaks of several species as measured in a planned self-encounter of the Geophysica plume during the RECONCILE experiment (Sumińska-Ebersoldt et al., 2012) in a flight of 30 January 2010 west of Kiruna/Sweden, see Figure 20. The encounter occurred at 19.0 km pressure altitude, 17.8 km geometric altitude, at T = -

74.5±0.2 °C, p = 64±0.3 hPa, position = (66.9+-0.1°N, 0.7+0.4°E), with the sun 4° below the horizon, and θ=438 ±0.2 K. Ice particles were not detected by the FSSP100, and the $H_2O$ molar mixing ratio was about 4±1 µmol mol$^{-1}$, with relative humidity over ice near 20 %. The temperature was 9 K below the Schmidt-Appleman threshold, so a contrail formed but sublimated in the time period before the encounter. The aircraft flew at constant altitude, implying an encounter of the secondary wake. The encounter occurred when the Geophysica was flying nearly parallel to its own exhaust plume. Figure

21 shows two encounter events of about 0.5 min duration, with one double peak at 30489 to 30522 s (17.2±0.2 min age) with maximum molar mixing ratio $CO_2$ increase of 0.6 µmol mol$^{-1}$ and a single slightly stronger peak at 30594 to 30622 (20.5±0.2 min age) with significant $CO_2$ increase of about 1 µmol mol$^{-1}$ above a slightly variable background level. Simultaneous increases of CO, $NO_y$, and particle concentrations of particles with size > 10 nm, volatile and nonvolatile, were found. The peak increases exceed the expected instrument precisions (von Hobe et al., 2013). For CO, the short term (<2

min) precision is ±1 nmol mol$^{-1}$, the long term (≥ 2 min) precision is ±3 nmol mol$^{-1}$; this is also the limit for the long term accuracy. Concentrations of particles > 6 nm and > 14 nm (not plotted) show similar peaks. The $NO_y$ peaks include mainly emitted $NO_x$ (NO and $NO_2$) (Arnold et al., 1992; Schulte et al., 1997). A weak peak of CO (0.7 nmol mol$^{-1}$) is detectable above a background of 2 to 3 nmol mol$^{-1}$. The background is far lower than in the midlatitude and the tropical stratosphere (Fahey et al., 1995; Viciani et al., 2008). CO concentrations may be low in the lower polar stratosphere for long air

confinement in the polar vortex with large chemical losses during this time and low source rates at low temperatures and low





actinic fluxes (Minschwaner et al., 2010). Temperature, water vapor and other trace gases show no significant peaks above measurement noise.

The plume ages were derived from backward trajectories to the source position for which the measured wind brings the air closest to the encounter position, see Figure 20. An ideal match is found for the two events for given wind directions and for velocity, which differ from the measured velocity of about 15 m s$^{-1}$ by 3±1 m s$^{-1}$. This is either a measure for wind variability along the trajectory from the place of source to the place of exhaust measurements or for wind measurement accuracy.

Figure 21 shows different shapes in the various signals indicating additional uncertainty from different temporal resolutions, response times, inlet positions, and travel times from the inlets to the detectors. Here, the instrument time resolutions in s are estimated as 2, 1, 4, 1, for $CO_2$, $NO_y$, CO, $n_{nv}$, respectively. To maximize the Pearson correlation coefficient $R^2$ of the time series of individual species relative to $CO_2$, a time delay of -9, -8, -4, -1 s was added to the times reported by the instruments for the CO, $NO_x$, and COPAS data ($n_{nv}$ and $n_{10}$).

The measured data were converted to mass-specific concentrations and the emission index was derived from the ratio of average mass concentrations of the individual species relative to that of $CO_2$ with known emission index $EI_{CO2}$. For averaging, we integrate over the time interval with obvious exhaust peak. The start and end points of this interval are selected so that the concentrations at the end points coincide with estimated background concentrations, and we assume a linear trend for the background within the interval. Mean, maximum and minimum EI values were derived from the ensemble of results for both events and for slightly different interval end times, see Table 6.

The measured $CO_2$ molar mixing ratio peak of 0.6 to 1 µmol mol$^{-1}$ fits well to the expected peak values $\Delta CO_2 = (29/44)$ $EI_{CO2}/N_{dil}(t) \approx 1$ to 1.2 µmol mol$^{-1}$ for $N_{dil}(t)$ from Eq. (4). The $CO_2$ peak concentration is in the same range as measured for a slightly younger plume (600 s age) in the NASA ER-2 exhaust (< 0.8 µmol mol$^{-1}$), and the derived $NO_x$ emission index is in the range (3.6-4.3) g kg$^{-1}$ estimated for the ER-2 at similar cruising flight conditions (Fahey et al., 1995), and also in the range (3.6-6.5) g kg$^{-1}$ estimated for the Geophysica by Weigel et al. (2009). About 60 % of the exhaust particles > 10 nm are non-volatile. The particle concentration for sizes > 6 nm is not much larger than that for > 10 nm; volatile particles < 6 nm may be abundant but were not measured. The PEI for nv particles 2.6 (2.0-3.2) ×10$^{15}$ kg$^{-1}$) derived this way is within the broad range of values (0.5 - 10)×10$^{15}$ kg$^{-1}$ for a variety of jet aircraft (Anderson et al., 1999). More modern engines have been found to emit of the order of ( 0.1-1) ×10$^{15}$ kg$^{-1}$ soot particles, depending on engine type and power setting (Lee et al., 2010). For the ER-2, only volatile particles were measured (range of 0.24 to 3×10$^{15}$ kg$^{-1}$, derived from Table 2 in Fahey et al. (1995)). These results confirm that the concentration peaks are likely caused by engine exhaust. The PEI value for nv particles is about ten times smaller than the value (24–44)×10$^{15}$ kg$^{-1}$ derived earlier for the SCOUT-O3 flight of 25 November 2005 (Weigel et al., 2009). The previous analysis did not account for an incidental but unrecognized time shift of about 2 min between the instruments' individual clocks. By considering this time shift, the $NO_y$ peak which chronologically



correlates at the best with the aerosol peak ranges at 3 to 4 times higher mixing ratios, implying correspondingly lower $PEI_{nv}$ values, still larger than the present result. Also the CO emission index has some uncertainty. The CO peak measured does not exceed the given short-term precision, the derived CO emission index is low compared to other measurements (Slemr et al., 2001), and the data for events E7, E8 and E13 explained in Section 3.2.1 suggest larger $EI_{CO}$ values. Still, the correlations

5 between CO and $CO_2$ and the other data support the validity of the data analysis.

*Author contributions.* The first author performed the analyses and drafted the paper; the co-authors contributed text, unpublished data, additional analyses, ideas, and discussed the results.

*Acknowledgements.*

The authors thank all colleagues who were engaged in the field experiments resulting in the data analyzed here. We thank all
10 principal investigators of the aircraft instruments for permission to use their data. The SCOUT-O3, RECONCILE, and TROCCINOX projects were funded by the European Commission under grants GOCE-CT-2004-505390, RECONCILE-226365-FP7-ENV-2008-1, and EVK2-CT-2001-00122, respectively. H. Schlager and R. Weigel received funding by the German BMBF within the joint ROMIC-project SPITFIRE (01LG1205A). Additional data and information were provided by Winfried Beer, Marius Bickel, Dominik Brunner, Luca Bugliaro, Ann Mari Fjæraa, Kaspar Graf, Andy Heymsfield, Peter
15 Hoor, Mareike Kenntner, Boon H. Lim, Peter May, Ralf Meerkötter, Valentin Mitev, Thomas Peter, Markus Rex, Michel Rossi, Anja Schubert, Nikolay Sitnikov, Silvia Viciani, Vasily Volkov, Martin Wirth, and Vladimir Yushkov, and are gratefully acknowledged. Moreover, we thank several further colleagues for valuable comments. In particular, we thank Charmaine Franklin and Heidi Huntrieser for valuable comments on the manuscript. We deeply regret that we can no longer discuss this research with Drs. Michael J. Mahoney, Cornelius Schiller and Genrikh Shur, who passed away in recent years.



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



**Table 1. Instruments (with time resolution Δt) providing data for this study.**

| *Aircraft /* Instrument | Parameter | Name, technique, remarks | Δt / s | Reference or Principle Investigator (PI) |
|---|---|---|---|---|
| *M-55* | | Geophysica, a twin engine reconnaissance jet, designed and operated by Myasishchev Experimental Design Bureau (MDB), Russia | | PI: G. Belyaev Stefanutti et al. (2004) |
| FISH | $H_2O$ (total) | Fast In-situ Stratospheric Hygrometer, Lyman-α. Data are not available for 16 November 2005. | 1 | PI: C. Schiller (Meyer et al., 2015) |
| FLASH | $H_2O$ (gas phase) | Fluorescent Airborne Stratospheric Hygrometer, Lyman-α. | 8 | Sitnikov et al. (2007) |
| COPAS | Condensation nuclei (total and nv) | Condensation Particle Counting (CPC) System, 4 CPCs for diameters > 6, 10 and 14 nm, and nonvolatile (nv) particles > 10 nm. | 1 | Weigel et al. (2009) |
| FSSP100 and CIP | Total ice number, surface area, and volume concentrations | Forward Scattering Spectrometer Probe covering particle size diameters from 2.7 to 31 μm and Cloud Imaging Probe delivering shadow cast particle images for sizes from 25 to 1550 μm. | 1 | de Reus et al. (2009) |
| MAS | Aerosol backscatter ratio and depolarization | Multiwavelength Aerosol Scatterometer, backscatter sonde at 532 and 1064 nm wavelengths. | 10 | Cairo et al. (2011) |
| MTP | temperature profile | Microwave Temperature Profiler on M55, vertical profiler. The MTP data have been calibrated against Darwin radiosondes. | 30 | PI: M. J. Mahoney (Denning et al., 1989) |
| MAL | backscatter profile | Miniature Aerosol Lidar, nadir-pointing backscatter lidar (532 nm wavelength) with depolarization | 30 | PI: V. Mitev (Corti et al., 2008) |
| FOZAN | $O_3$ | Fast-response chemiluminescent airborne ozone analyzer | | Ulanovsky et al. (2001) |
| SIOUX | NO, $NO_y$ | Stratospheric Observation Unit for Nitrogen Oxides, chemiluminescence. Data are not available for 30 November 2005 | 1 | PI: H. Schlager (Voigt et al., 2007) |
| COLD | CO | Cryogenically operated laser diode spectrometer, tunable diode laser | 4 | Viciani et al. (2008) |





| HAGAR | $CO_2$ | High Altitude Gas Analyzer, IR absorption for $CO_2$. The data have been reanalyzed for this study. Data are not available for 16 November 2005. | 3 | PI: M. Volk (Homan et al., 2010) |
|---|---|---|---|---|
| TDC | T, wind | Thermodynamic Complex, Rosemount probe PT-100, and 5-hole probe. | 0.1 | Shur et al. (2007) |
| UCSE | p, T, wind, position | Basic meteorology measurement system | 1 | Sokolov and Lepuchov (1998) |
| *Falcon* | | Falcon-20 E, twin engine jet, operated by DLR (call sign D-CMET) | | Krautstrunk and Giez (2012) |
| DIAL | backscatter and depolari- zation | Water vapor Differential Absorption Lidar; backscatter ratio at 532 and 1064 nm wavelengths, and depolarization | 10 | PI: G. Ehret (Poberaj et al., 2002) |
| D-CMET | position | Falcon basic measurement system | 1 | PI: A. Giez |





**Table 2. Mean properties of uppermost invisible plume, upper ("white"), and lower ("red") contrail, and tropopause properties with ranges or standard deviations for 16 November 2005.**

| Region | Invisible plume | Upper contrail | Lower contrail | Tropopause |
|---|---|---|---|---|
| UTC time/(h:min) | 7:30 | 7:42 | 7:58 | 8:10 |
| Flight time/s | 27000-27751 | 27751-28721 | 28721-29680 | 29400-29700 |
| Age/min | 53 to 65 | 36 to 53 | 20 to 36 | n.a. |
| $z$/km | 19.06±0.1 | 18.6±0.1 | 17.9±0.3 | 17.3±0.04 |
| $p$/hPa | 65.2±1.1 | 71.2±1.2 | 80.8±4 | 89.0±0.8 |
| $T$/°C | -77.5±0.8 | -82.2±0.6 | -84.9±2.0 | -88.6±0.2 |
| $H_2O$/($\mu$mol mol$^{-1}$) | 3.8±0.25 | 3.4±0.23 | 2.5±0.44 | 1.9±0.2 |
| RHi/% | 31±5 | 64±7 | 87±23 | 132±13 |
| NO/(nmol mol$^{-1}$) | 0.58± 0.09 | 0.28±0.03 | 0.22±0.05 | 0.17±.04 |
| $NO_y$/(nmol mol$^{-1}$) | 2.6±0.3 | 1.3±0.1 | 0.78±0.28 | 0.40±0.04 |
| $n_{nv}$/cm$^{-3}$ | 5.3±0.7 | 12±2 | 45±35 | 85±36 |
| $n_{10}$/cm$^{-3}$ | 18±3 | 30±3 | 136±134 | 327±85 |
| $O_3$/(nmol mol$^{-1}$) | 402±46 | 220±26 | 115± 61 | 35±4 |
| CO/(nmol mol$^{-1}$) | 32±10 | 47±6 | 53±4 | 55±2.8 |
| $T_{LC}$/°C | -65.0 | -63.7 | -62.4 | -60.5 |
| $T_{ICE}$/°C | -84.7 | -84.8 | -85.9 | -86.9 |
| $T_{NAT}$/°C | -79.6 | -80.5 | -81.9 | -83.6 |
| $z'$/m | 1.8 | 2.4 | 7.3 | 7.6 |
| $N_{BV}$/s$^{-1}$ | 0.0263 | 0.0269 | 0.026 | 0.03 |





**Table 3. Mean properties[*] of ice events, E1 - E6 from de Reus et al. (2009) and short-term ice events E0, E7 and E10 for 30 November 2005.**

| | unit | E0 | E1 | E2 | E3 | E4 | E5 | E6 | E7 | E10 |
|---|---|---|---|---|---|---|---|---|---|---|
| t | h:min | 5:13 | 5:43 | 5:52 | 5:55 | 6:21 | 6:28 | 6:55 | 6:00 | 6:08 |
| t | s | 18780 | 20600 | 21140 | 21300 | 22863 | 23318 | 24952 | 21633 | 22120 |
| $\Delta t_e$ | s | 15 | 150 | 40 | 30 | 50 | 60 | 100 | 10 | 15 |
| Age | s | n.a. | 1709 | 1642 | 2529 | 2302 | 4433 | 3000-7157 | 3025 | 1531 |
| z | km | 18.3 | 18 | 18 | 18.4 | 18.4 | 18.7 | 18.2 | 18.3 | 18.0 |
| $z_p$ | km | 18.2 | 17.7 | 17.8 | 18.2 | 18.2 | 18.6 | 18 | 18.1 | 17.8 |
| T | °C | -83.6 | -81.7 | -87.1 | -83.9 | -84.1 | -80.9 | -83.2 | -82.3 | -80.9 |
| p | hPa | 73.0 | 78.3 | 77.6 | 72.1 | 72.8 | 68.3 | 74.8 | 74.1 | 77.3 |
| θ | K | 402 | 396 | 386 | 401 | 399 | 414 | 398 | 403 | 401 |
| RHi | % | n.a. | 76 | 157 | 95 | 107 | 75 | 89 | 68 | 75 |
| $n_{ice}$ | $cm^{-3}$ | 0.01 | 0.1 | 0.3 | 0.015 | 0.048 | n.a. | 0.05 | 0.014 | 0.01 |
| IWC | $mg\ m^{-3}$ | 0.1 | 0.85-1.3 | 0.64-0.99 | 0.1-0.16 | 0.077-0.16 | n.a.-0.072 | 0.21-0.4 | 0.01-0.04 | 0.01-0.015 |
| $r_{eff}$ | μm | 5.0 | 25.3 | 17.9 | 23.6 | 11.5 | n.a. | 19.5 | n.a | n.a. |
| $r_{vol}$ | μm | 6 | 9.3 | 4.6 | 7.7 | 5.1 | n.a. | 7.1 | 9 | 7.3 |
| $\Delta n_{nv}$ | $cm^{-3}$ | 0 | 20 | 40 | 100 | 40 | 50 | 50 | 1000 | 100 |
| $n_{ice}/\Delta n_{nv}$ | 1 | - | 0.005 | 0.0075 | 0.00015 | 0.0012 | n.a. | 0.001 | $10^{-5}$ | $10^{-4}$ |

*) t: UTC time of 30 November 2005, $\Delta t_e$: estimated event duration, age: computed plume age, z: altitude above MSL, $z_p$: pressure

5 altitude, T: temperature, p: pressure, θ: potential temperature, IWC: ice water content (lower value from observed ice crystal size distribution and upper from two hygrometers), $r_{eff}$ and $r_{vol}$: effective and volume mean particle radius. $\Delta n_{nv}$: nv particle concentration above background.





**Table 4.** Potential plume self-encounters (for symbols see text), with classification as convective (C) or exhaust (E) event, 30 November 2005.

| Event | Age/s | t/s | t/(h:min) | z/km | $\Delta z$/km | $\Delta u$/( m s$^{-1}$) | $\Delta v$/( m s$^{-1}$) | $\Delta w$/( m s$^{-1}$) | $\Delta\theta$/K | CIP | C or E |
|---|---|---|---|---|---|---|---|---|---|---|---|
| ice E1 | 1708 | 20600 | 5:43 | 17.96 | -0.44 | -0.28 | 0.91 | -0.17 | -10.7 | y | C+E |
| ice E2 | 1648 | 21140 | 5:52 | 18.00 | -0.40 | -0.12 | -3.42 | -0.14 | -15.9 | y | C |
| ice E3 | 2512 | 21300 | 5:55 | 18.37 | 0.01 | -2.30 | 2.58 | 0.03 | -2.0 | y | C+E |
| ice E4 | 2295 | 22863 | 6:21 | 18.37 | 0.39 | 0.14 | -0.38 | 0.18 | 4.3 | no | E |
| ice E5 | 4430 | 23318 | 6:28 | 18.72 | 0.32 | 0.34 | 1.67 | 0.08 | 6.4 | y | C+E |
| ice E6a | 3006 | 24952 | 6:55 | 18.21 | 0.10 | -1.63 | -0.93 | 0.03 | -1.3 | y | C+E |
| ice E6b | 4473 | 24952 | 6:55 | 18.21 | 0.18 | -1.41 | 1.99 | 0.04 | 2.1 | y | C+E |
| ice E6c | 6047 | 24952 | 6:55 | 18.21 | -0.20 | -0.10 | 0.57 | -0.04 | -10.3 | y | C+E |
| ice E6d | 7219 | 24952 | 6:55 | 18.21 | 0.19 | 1.18 | -0.17 | 0.03 | -1.1 | y | C+E |
| ice E7 | 3025 | 21633 | 6:00 | 18.28 | -0.10 | 2.25 | -2.99 | -0.04 | -6.4 | no | E |
| E8 | 1969 | 21763 | 6:02 | 18.33 | 0.00 | 0.01 | 0.13 | 0.01 | 2.4 | no | E |
| ice E9 | 1514 | 22008 | 6:06 | 18.00 | -0.01 | 0.51 | -0.78 | 0.00 | 0.4 | no | E+C |
| ice E10 | 1531 | 22120 | 6:08 | 18.04 | 0.08 | 0.81 | -1.32 | 0.07 | 2.5 | no | E+C |
| E11 | 1758 | 23588 | 6:33 | 18.50 | 0.11 | -1.20 | 2.79 | 0.06 | 3.1 | no | E |
| E12 | 1831 | 23632 | 6:33 | 18.61 | 0.24 | 0.65 | 3.53 | 0.14 | 7.3 | no | E |
| E13 | 6738 | 24397 | 6:46 | 17.96 | -0.44 | -0.04 | 0.11 | 0.07 | 7.0 | no | E |





**Table 5. Scales of the two stratospheric anvil cloud layers as seen from the Falcon in lidar backscatter signals with maximum optical depth $\tau_{max}$ and total extinction EA, 30 November 2005.**

| Layer | Begin time/(h:min) | Begin time/s | End time/s | Altitude min-max/km | Maximum depth/m | Maximum width/km | $\tau_{max}$ | EA/ m | Plume age/h |
|-------|-----------|-----------|-----------|------------|----------|----------|--------------|-------|-------------|
| Upper | 7:34 | 27250 | 27580 | 17.1-18.5 | 900 | 67 | 0.04 | 639 | 1.56±0.81 |
| Lower | 7:36 | 27370 | 27640 | 16.5-17.6 | 1100 | 55 | 0.08 | 1505 | 2.21±0.66 |

**Table 6. Emission indices derived from two self-encounters at 20 (17.7 to 21.7) min plume ages and Pearson correlation coefficients $R^2$ relative to $CO_2$, 30 January 2010.**

| EI or PEI for species | Unit | Mean | Minimum | Maximum | $R^2$ |
|-----------------------|------|------|---------|---------|-------|
| $EI_{CO}$ | $g\ kg^{-1}$ | 1.9 | 1.1 | 2.8 | 0.81 |
| $EI_{NOx}$ | $g\ kg^{-1}$ | 4.1 | 3.2 | 5.0 | 0.75 |
| $PEI_{nv}$ (nv, > 10 nm) | $10^{15}\ kg^{-1}$ | 2.6 | 2.0 | 3.2 | 0.79 |
| $PEI_{volatile}$ (total, > 10 nm) | $10^{15}\ kg^{-1}$ | 4.4 | 3.4 | 5.2 | 0.91 |





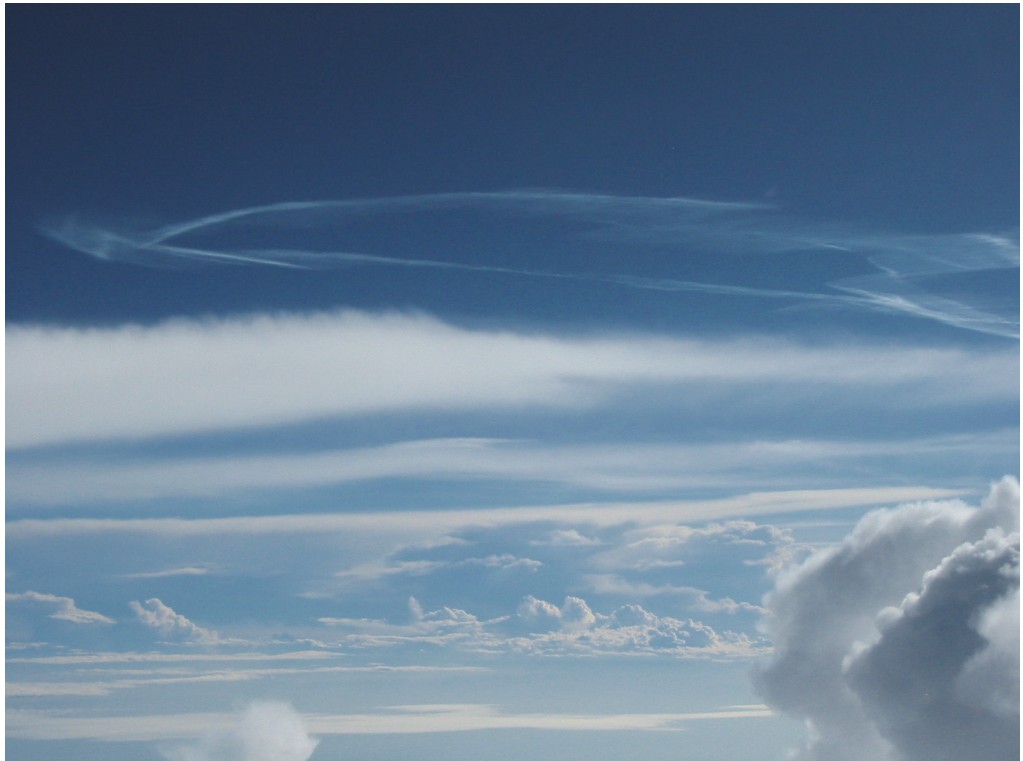

**Figure 1:** Geophysica contrail, photo from the DLR Falcon at 130.0°E, 11.8°S, 8:35 UTC (18:05 LT) 16 November 2005.





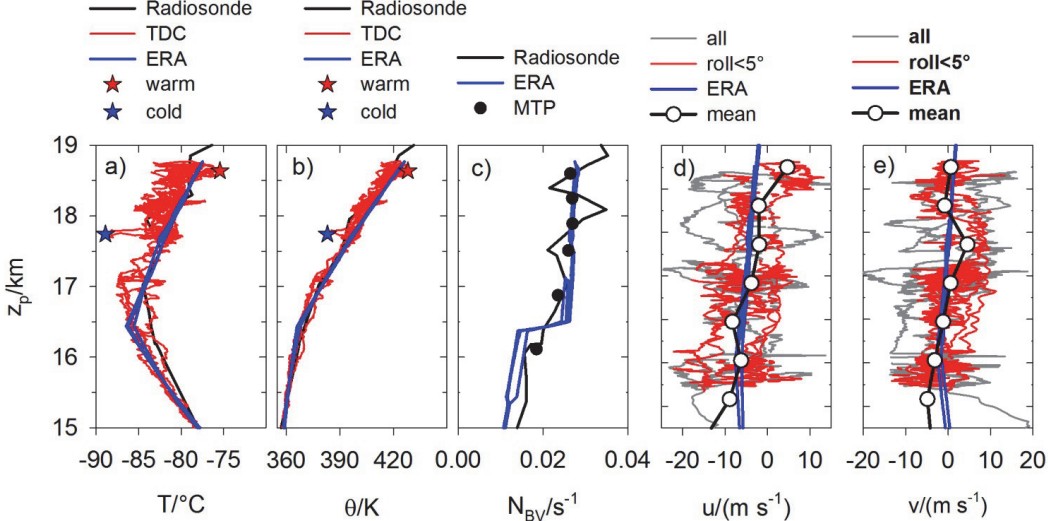

**Figure 2:** Vertical profiles versus pressure-altitude $z_p$ of a) air temperature T, b) potential temperature θ, c) Brunt-Väisälä frequency $N_{BV}$, d) easterly wind component u, e) northerly wind component v, for 30 November 2005. Symbols for a-c: Black line: Darwin radio sounding of 12 UTC. Red line: Geophysica-TDC temperature. Blue line: ERA interim reanalysis interpolated along the Geophysica flight path. Red and blue stars: warmest and coldest points measured in-situ with the Geophysica-TDC in the stratosphere this day (-75.4°C, -88.9°C, from TDC, -74.65°C, -87.44°, from UCSE data). The cold-point tropopause is at about 17 km pressure altitude (θ ≈ 375 K). Black circles in c) denote mean $N_{BV}$ versus pressure altitude over Hector from MTP. Symbols for d) and e): The dark grey line depicts the 30-s running mean values of 1 Hz TDC wind component data. Red: same for roll angle less than 5°. Blue line: ERA interim re-analysis interpolated along the flight path. Black line and full symbols: Mean values for low roll. The geometric altitude z is 0.65 km to 0.2 km higher than $z_p$ between tropopause and maximum flight level.





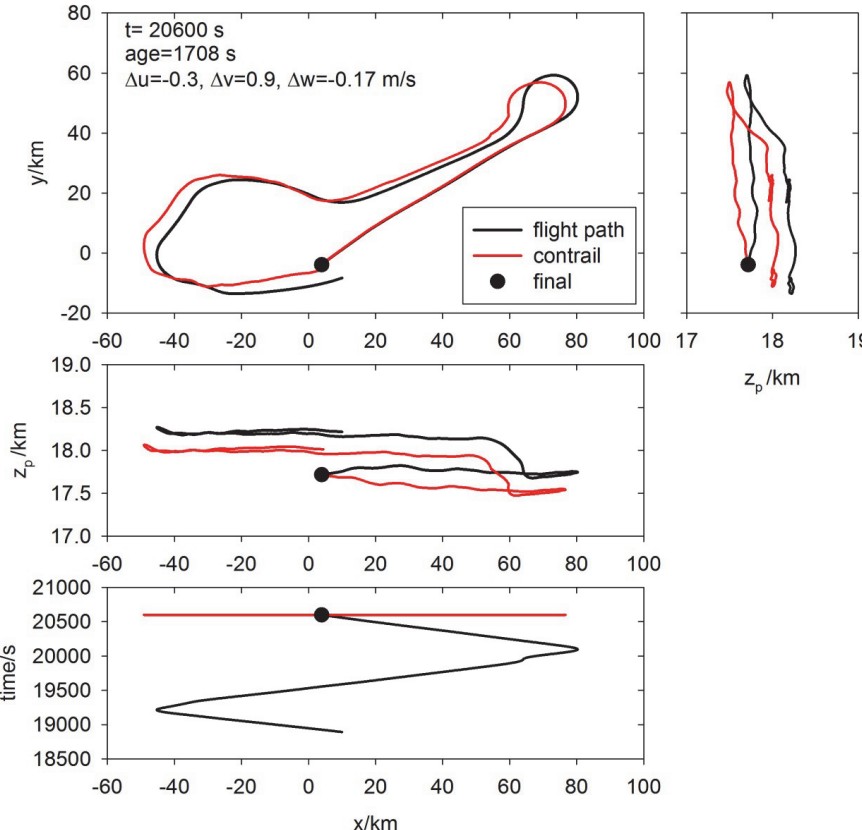

**Figure 3:** Geophysica flight path (black line with final Geophysica position identified by a black circle) and advected plume position (red line, "contrail") in four dimensions, between $t_1 = 18892$ s and $t_2 = 20600$ s (age = 1708 s, event E1) of 30 November 2005. Also given are the velocity component increments $\Delta u$, $\Delta v$, $\Delta w$ necessary for a perfect match of the final position (black symbol) and the position of the oldest plume part (open end of red line).





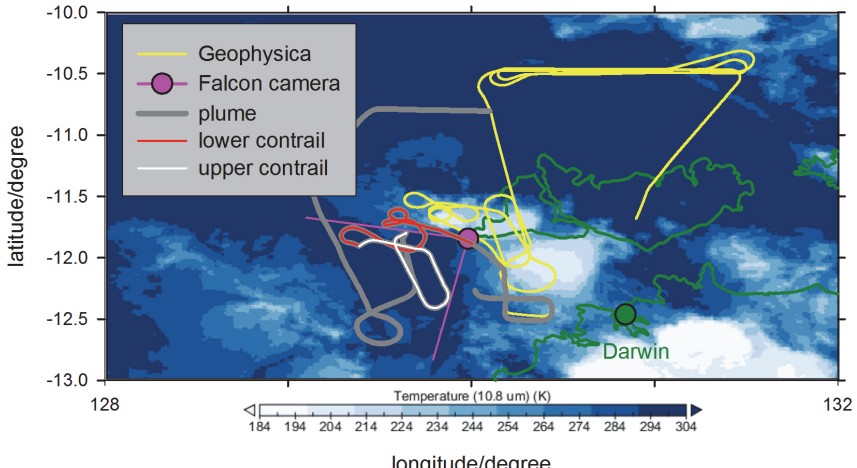

**Figure 4:** Falcon camera view directions above 10.8-μm brightness temperature image from NOAA-12 AVHRR at 08:28:53 UTC 16 November 2005. Yellow curve: Geophysica flight path until 8:35 UTC. Dark grey: computed plume position. Overlaid white and red lines are the positions of the upper and lower contrail parts seen in the photo. Green symbol/line: Darwin and the coast line. The minimum brightness temperature over Hector is 203 K at this time.



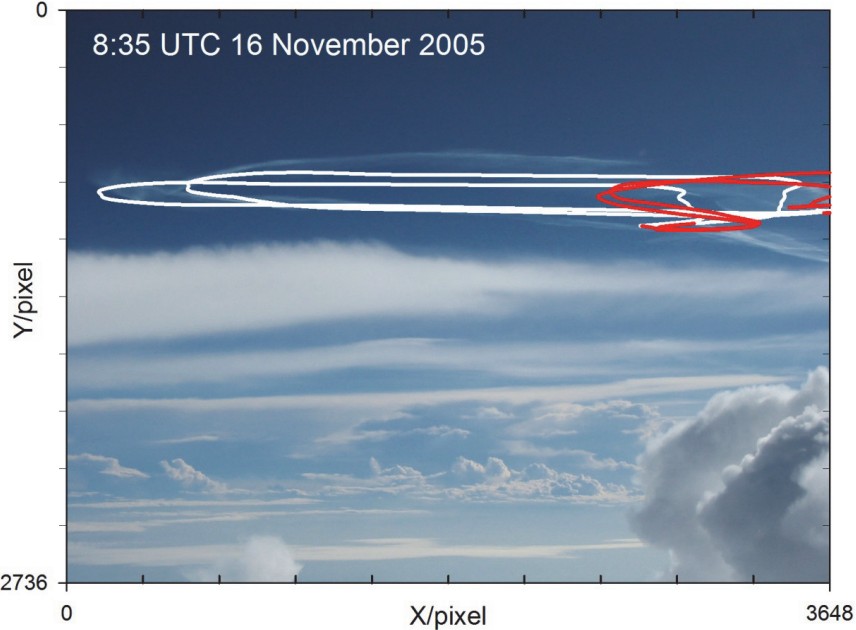

**Figure 5:** Computed contrail positions overlaid over the photo shown in Figure 1 (8:35 UTC, 16 November 2005). The coordinates count the number of photo pixels. The pair of computed contrail lines is derived from plume advection with and without wake descent. The line colors denote an upper (white) and a lower (red) contrail part separated at 18 km altitude.

5     The lower contrail reaches below the tropopause.




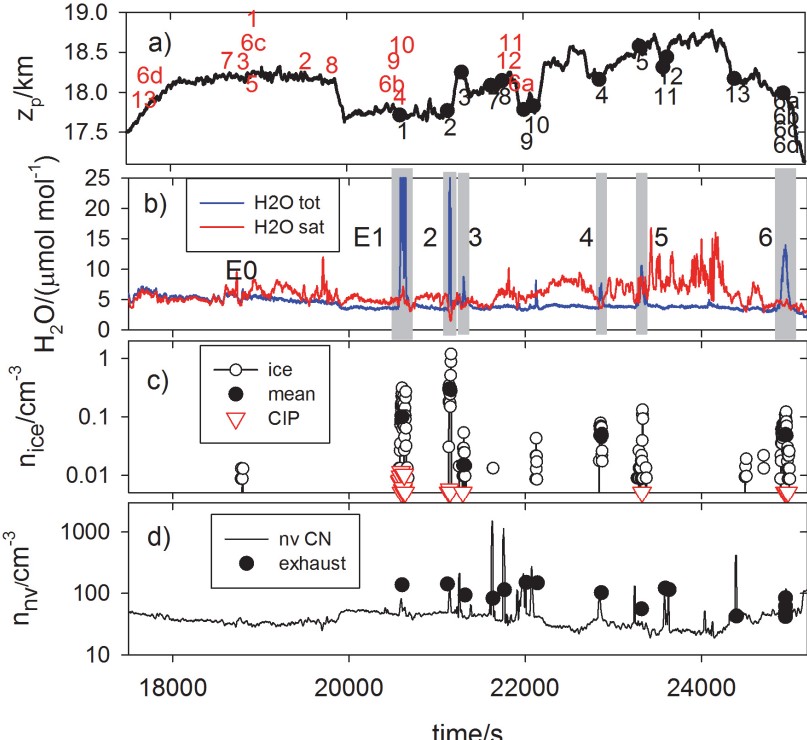

**Figure 6:** In-situ data measured during Geophysica flight in the morning of the Hector Day, 30 November 2005. **a)** Pressure altitude with numbered positions (C, red symbols) where contrails formed that might have caused aerosol and corresponding events E (black symbols) later. There are 16 events, here numbered 1 to 13, with event E6 having 4 source-contrails 6a to 6d of different ages. **b)** Molar water vapor mixing ratio $H_2O$ in µmol mol⁻¹ (blue: total from FISH; data before time 19800 s are shown though classified as possibly being contaminated by initial outgassing in the measurements); red: $H_2O$ ice saturation; grey bars: events E1 to E6 with time periods of high ice particle concentrations (de Reus et al., 2009). **c)** Black line with white circles: ice number concentration $n_{ice}$ from FSSP100; black symbols: mean $n_{ice}$ values for events E1 to E6 from de Reus; red triangles: indicate that concentrations > 0.005 cm⁻³ or at least one CIP particle > 30 µm has been measured. **d)** Number concentration $n_{nv}$ of nonvolatile particles (size >10 nm) from COPAS; black symbols: computed $n_{nv}$ for given plume age, fuel consumption and $PEI_{nv}$..





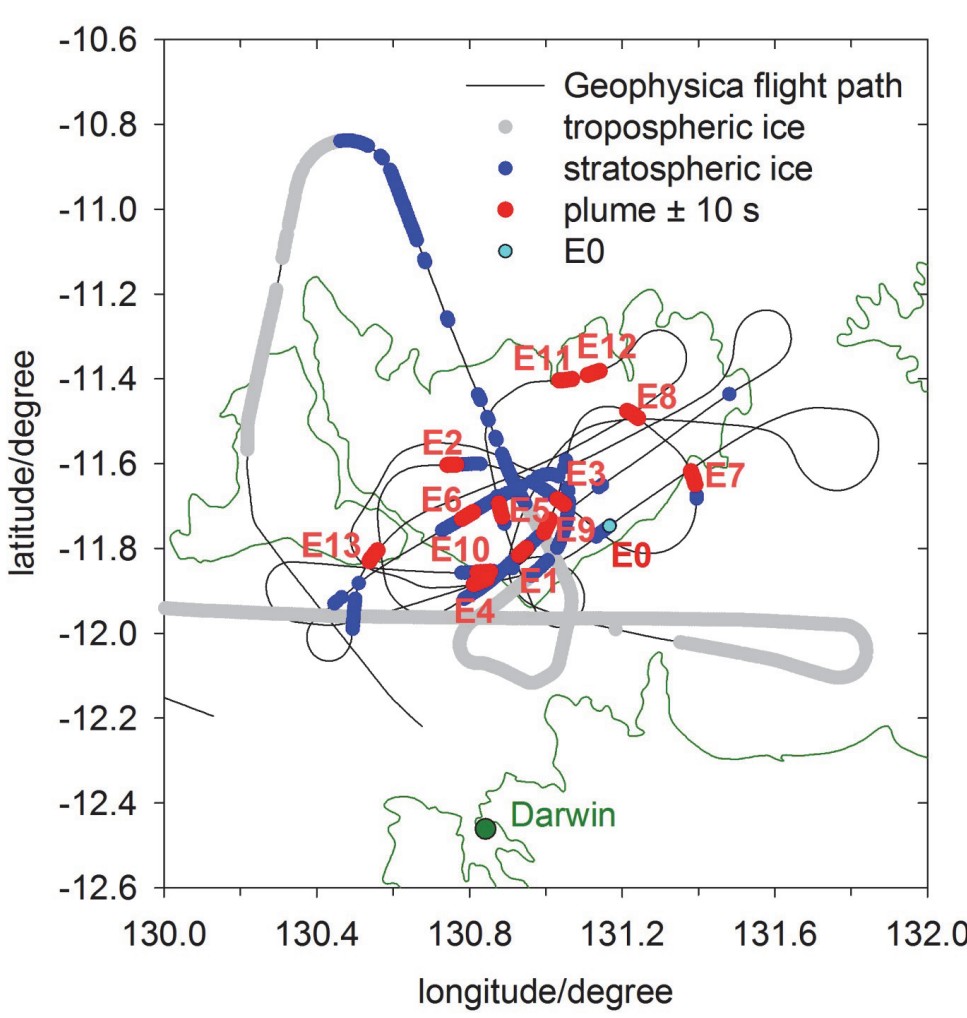

**Figure 7:** Map with Geophysica flight path (black line) of 4:00-7:47 UTC 30 November 2005. Colored symbols on the flight path identify periods with IWC>0.01 g m$^{-3}$ in tropospheric cirrus (grey), stratospheric cirrus (blue), and potential exhaust plume or contrails (red). Event positions are identified with "E" and event number. Darwin and the coast line are indicated in

5    green color.





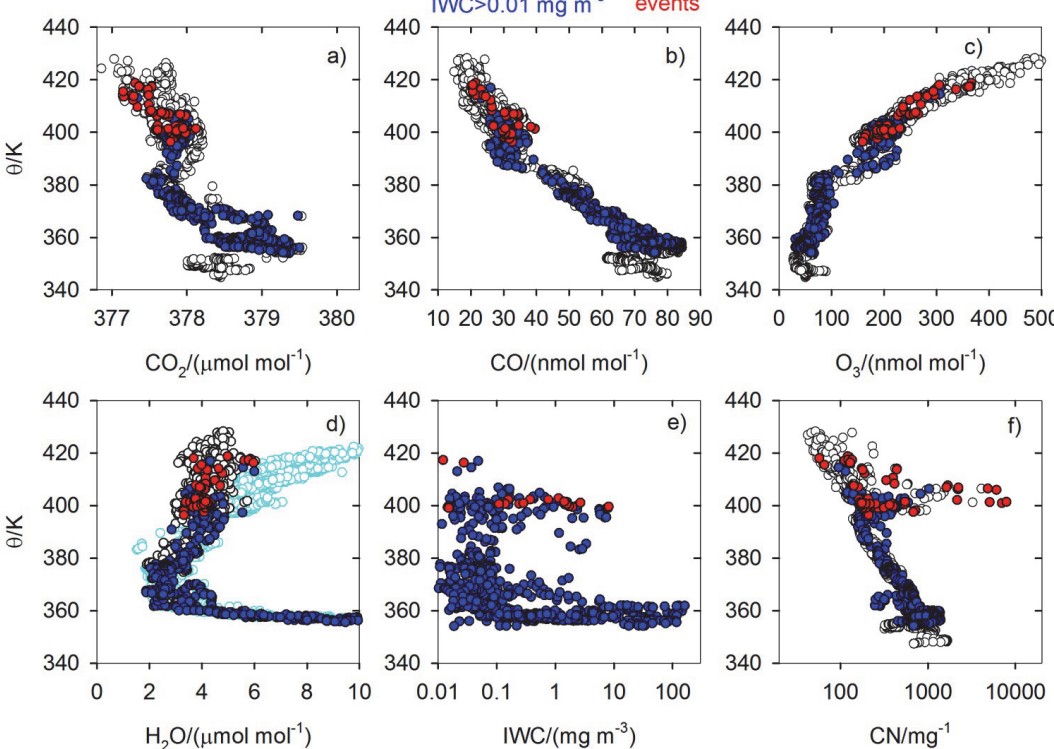

**Figure 8:** Scatter plots of tracers versus potential temperature θ: molar mixing ratios of **a)** $CO_2$, **b)** CO, **c)** $O_3$, **d)** $H_2O$ (from FLASH) and ice saturation mixing ratio (from TDC), **e)** ice water content (maximum from FISH-FLASH and FSSP+CIP), and **f)** mass specific nv particle concentration. White: all data; blue: IWC>0.01 mg m$^{-3}$, red: within 10 s to a potential plume encounter. Cyan in d denotes the $H_2O$ molar mixing ratio for ice saturation. Data interpolated to the same time with 0.2 Hz resolution, for 30 November 2005.


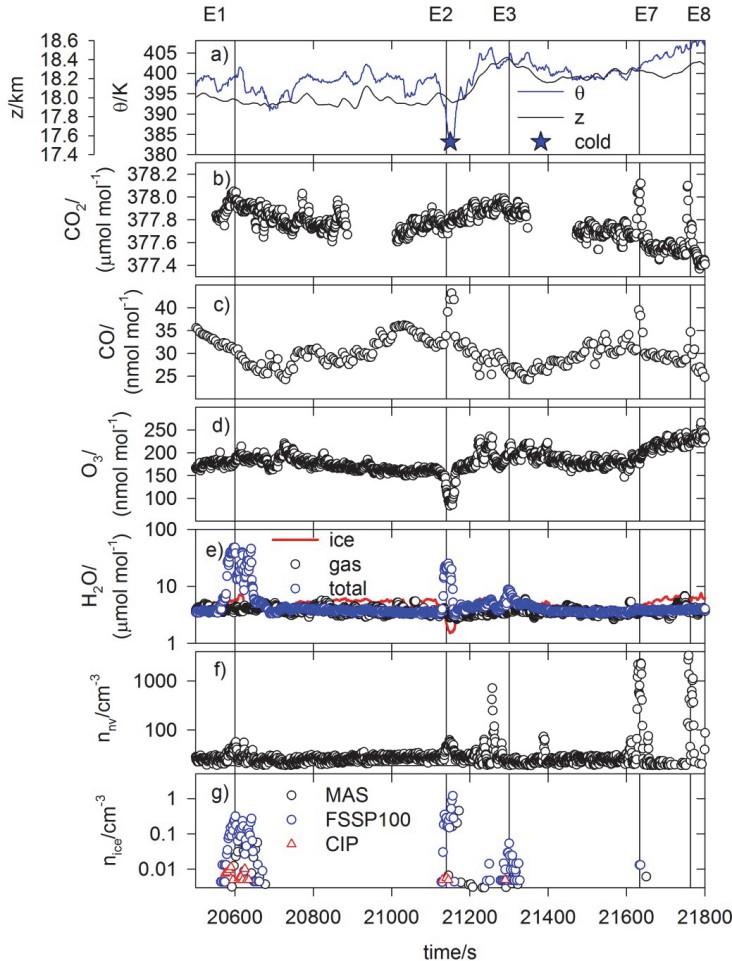

**Figure 9:** Measurement results versus time for the time period containing events E1, E2, E3, E7 and E8 (with event times indicated by vertical lines). a) 1-Hz data for geometric altitude z and potential temperature θ, with identification of the minimum temperature ("cold"). b) Molar mixing ratios of carbon dioxide $CO_2$, c) carbon monoxide CO, d) ozone $O_3$, e) water vapor $H_2O$, for ice saturation (red line), gas phase (black circle), total water (blue circle). f) Nonvolatile particle concentration and g) ice particle concentrations as derived from MAS backscatter (black), FSSP100 (blue) and CIP (red).



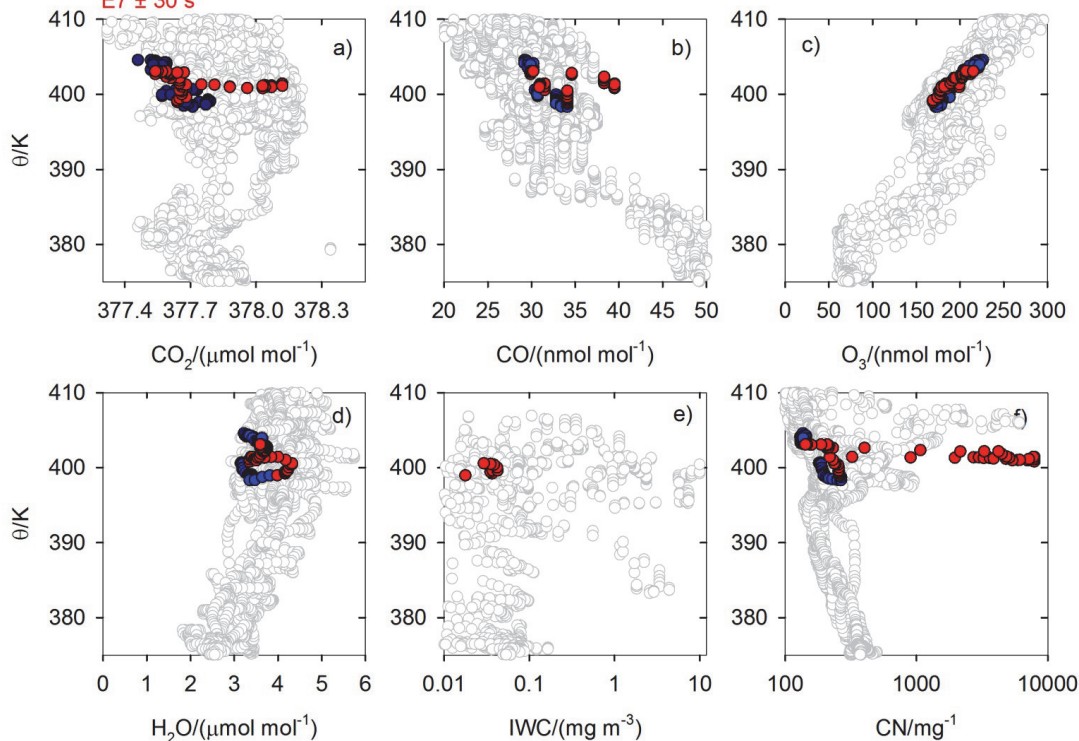

**Figure 10:** Case E7, best example for a contrail-induced ice event. Same data as in Figure 8 but at 1 Hz resolution. Grey: all data. Red: data within ± 30 s before and after t=21633, blue: neighbors in the next 30-s intervals before and after this interval.



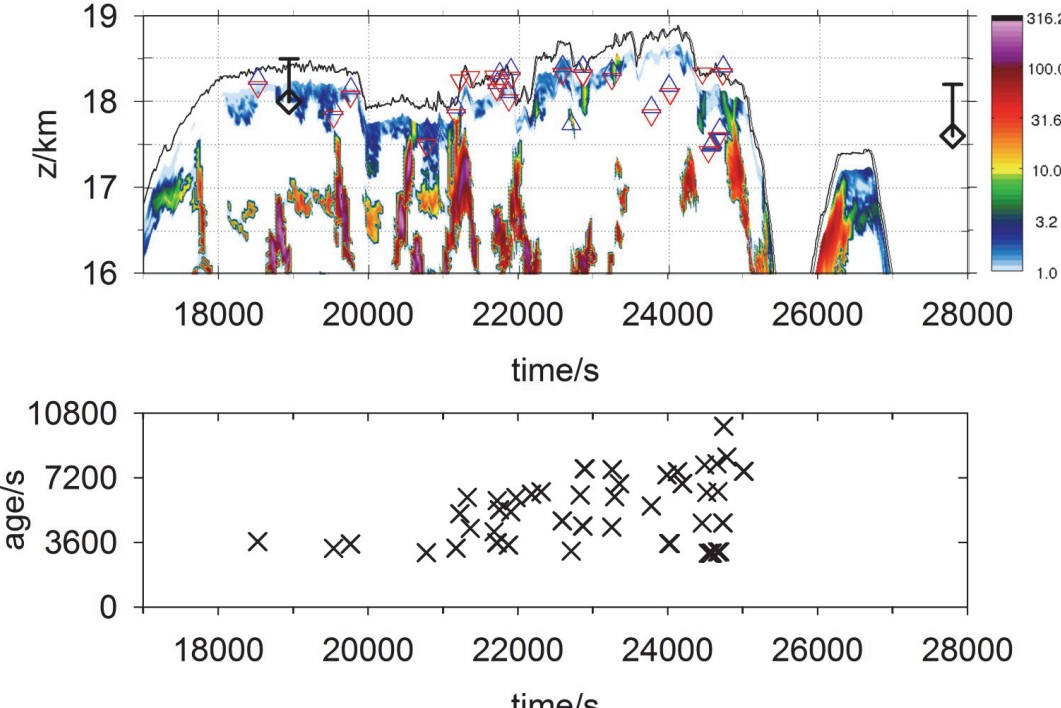

**Figure 11:** MAL lidar backscatter "curtain" below the Geophysica flight path of 30 November 2005 with computed plume or contrail positions and their ages. **Top panels:** 532-nm backscatter ratio (color scale), averaged over 30 s and 42 m in time and vertical directions, versus time. Triangles identify altitudes/times of plumes in the curtain, with red or blue symbols for primary or secondary wakes. The diamonds with one-sided error bars represent the cloud-top altitude derived from NOAA-AVHRR data for optically thick cloud tops, with an estimated altitude range of optically thinner clouds above. **Bottom:** Ages of penetrated plumes versus time.





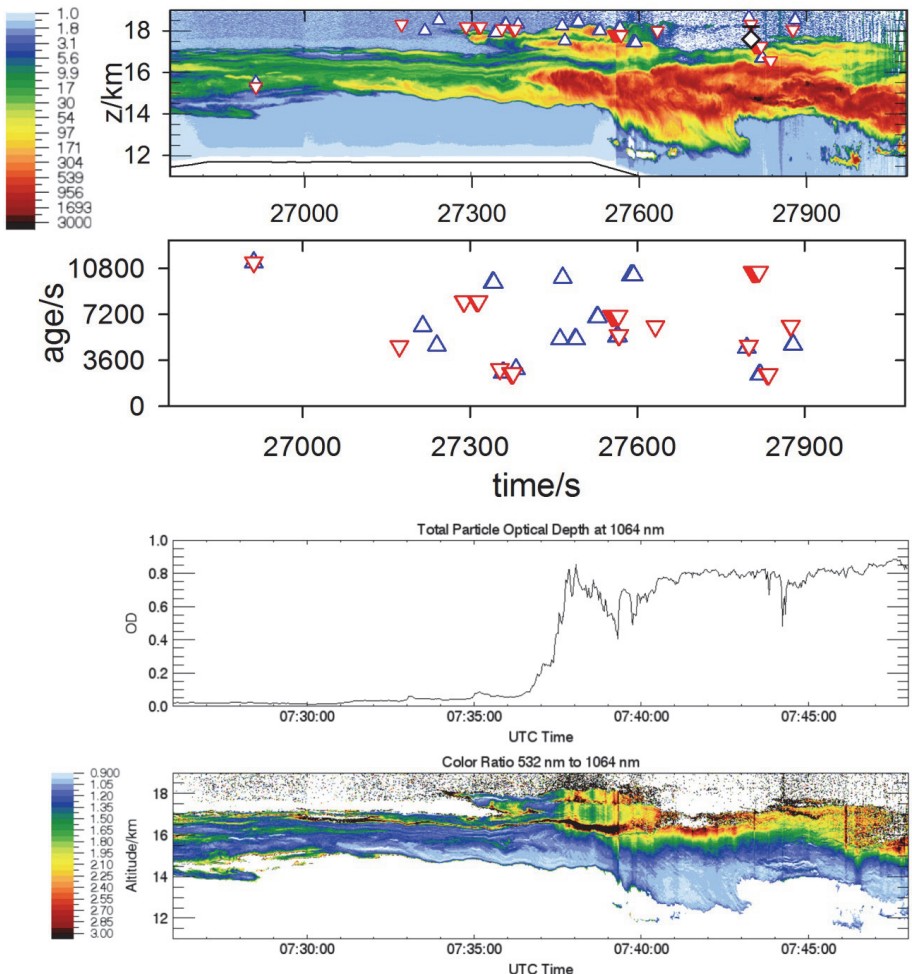

**Figure 12:** Falcon lidar signals in the vertical plane ("curtain") above the flight level for a 23-min time segment, total 254 km, 7:26 to 7:49 UTC, 30 November 2005. **Top panel:** 1064-nm backscatter ratio with overlay of potential contrail positions (red: primary; blue: secondary wakes). At t = 27800 s, 7:43:20 UTC, the black diamond with error bar indicates the cloud top height derived from AVHRR data. **2nd:** Plume ages. **3rd panel:** Total optical depth at 532 nm, **bottom panel:** Color ratio (ratio of extinction at 532 to 1064 nm).





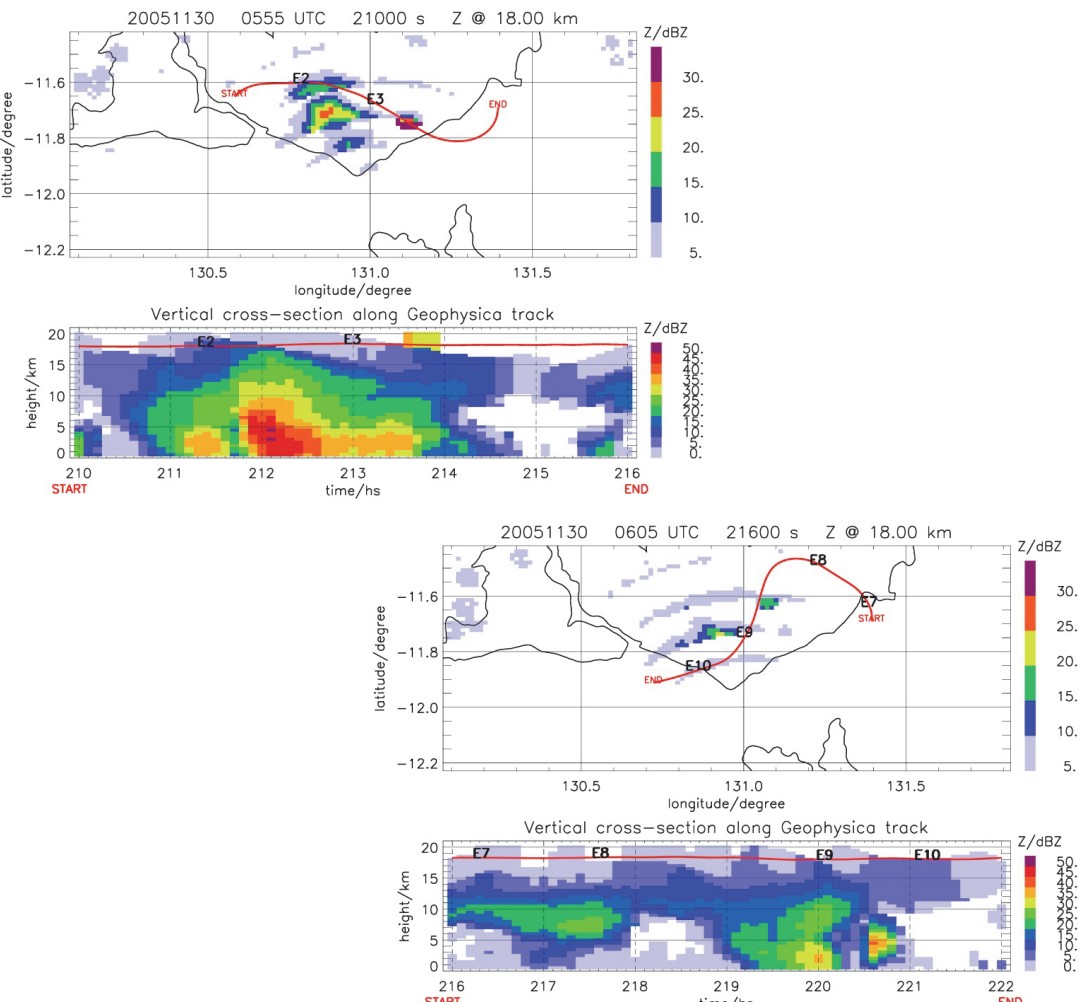

**Figure 13:** Radar reflectivity Z from CPOL (color scales in decibel dBZ) with the Geophysica flight path (red line) during a 600-s time period centered at **top**: 5:55 UTC (21000 s, events E2 and E3) and **bottom**: 6:05 UTC (20600 s, events E7, E8, E9 and E10), 30 November 2005. Top panels: Interpolated Z in a horizontal plane versus longitude and latitude at 18 km altitude. Event positions and "START" and "END" of begin and end of the flight segment are identified. Bottom panel: Interpolated Z versus altitude and time. The circular pattern in horizontal slices for low Z values and large distances reveals the radar observation pattern. The Z peak after E3 at 21340-21400 s is caused by the aircraft.


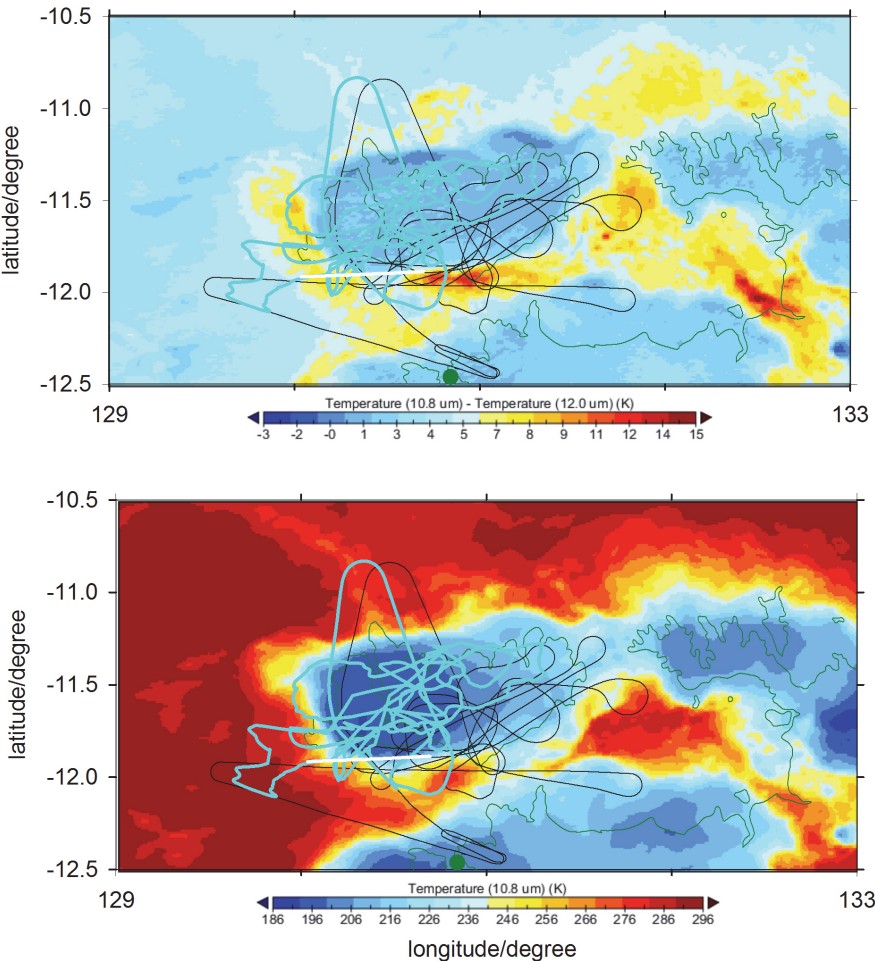

**Figure 14**: Brightness temperature (BT, bottom, at 10.8-μm wavelength) and BT difference (BTD, top; difference between BT at 10.8 and 12 μm) from 1-km resolution NOAA-12 AVHRR at 7:43:20 UTC 30 November 2005. The black curve is the Geophysica flight path. A white line depicts the Falcon flight path during Hector anvil lidar observations between 7:35 and
5   7:41 UTC. The cyan curve is the computed position of the contrail for ages < 2.7 h at the time of the overpass. Green symbol/line: Darwin and the coast line.





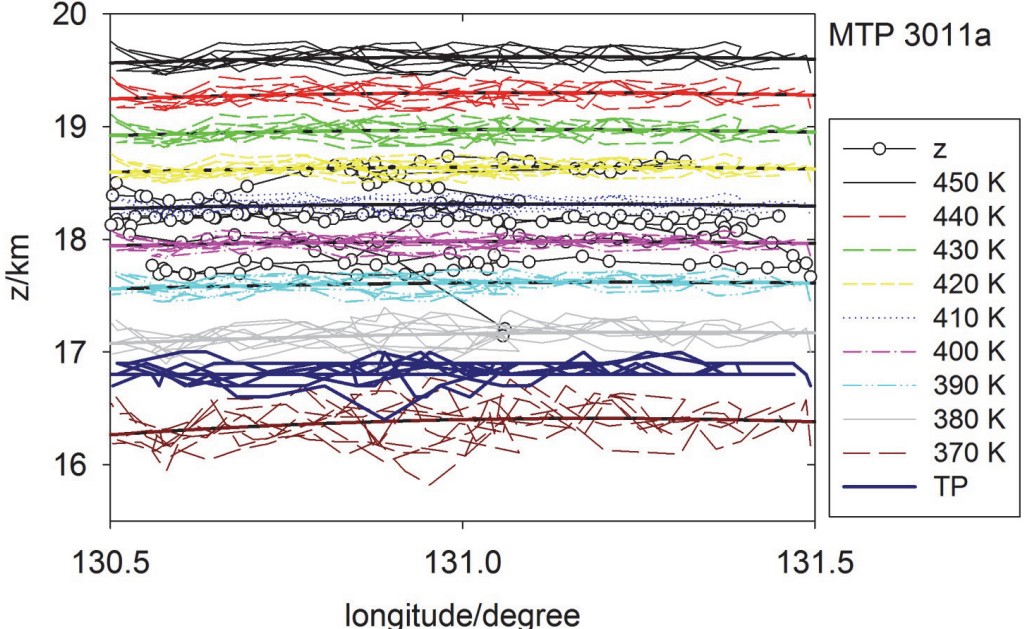

**Figure 15:** Tropopause (TP) and isentrope altitudes for potential temperature between 370 and 450 K, from MTP versus longitude (with thick parabolic regression curves) including the stratospheric flight part over Hector, 18000 s < t < 25200 s, 130.5° < longitude < 131.5°E, 30 November 2005. The parabolas indicate lifting over Hector by ~100 m at 370 K and 50 m

5 at 450 K. The black line with open circles depicts the Geophysica flight altitude above MSL,





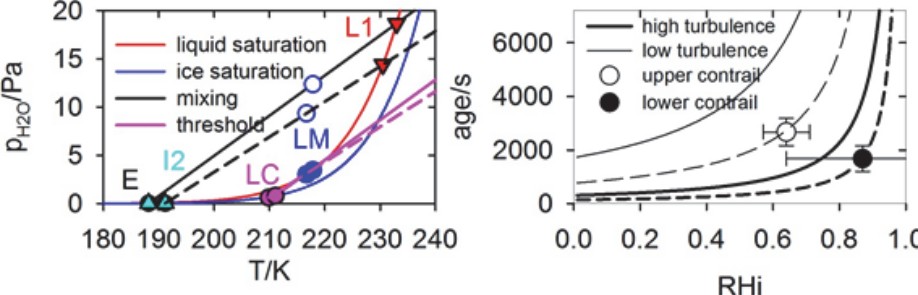

**Figure 16: Left:** Schmidt-Appleman criterion (SAC) for stratospheric conditions with saturation pressures for ice and liquid water (blue and red curves) and mixing lines (black, full for lower contrail, dashed for upper contrail) between exhaust and ambient conditions. Letters at symbols identify LM (maximum potential liquid saturation at saturation and on the mixing line), L1 (first liquid saturation or dew point), I2 (last ice saturation), and LC (liquid threshold conditions). With growing dilution without phase changes, the plume state would approach E from above along the mixing line. The conditions in the environment E are $T_E$=-85°C, p=85 hPa, RHi= 0.87 for the lower contrail and $T_E$=-82°C, p=71 hPa, RHi= 0.64 for the upper contrail, η=0.1. **Right:** Contrail age computed for conditions at I2 for variable ambient relative humidity relative to ice saturation, RHi. The age is derived such that the dilution at I2 equals the dilution at this age from the assumed dilution law ("high turbulence", Eq. 4). The thin curves show the result for a factor 0.25 reduced dilution ("low turbulence"). The symbols with error bars show observed values of RHi and ages for the upper and lower contrail in the photo of 16 November 2005. The dashed lines and the open symbol are for the upper contrail, and the solid lines and the full symbol are for the lower contrail.





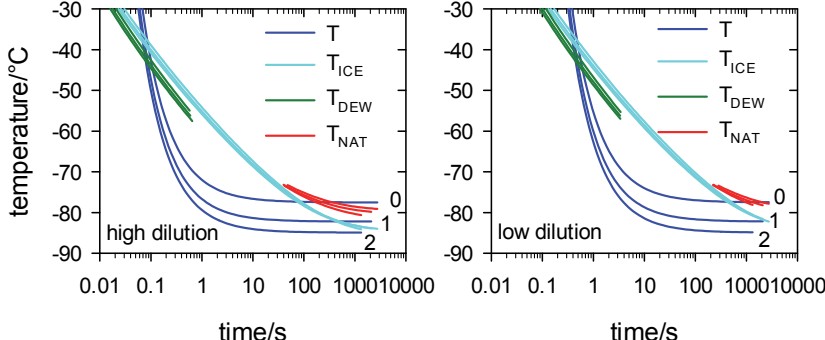

**Figure 17:** Air temperature T, frost point $T_{ICE}$, dew point temperature $T_{DEW}$, and nitric acid trihydrate existence temperature $T_{NAT}$ in the exhaust plumes (0: invisible plume, 1: upper contrail, 2: lower contrail) for high (left) and low (right) dilution. $T_{DEW}$ is shown up to the point of maximum liquid saturation (LM in Figure 16), $T_{NAT}$ is shown for T<200 K, i.e. in the validity range of the relation given by Hanson and Mauersberger (1988).





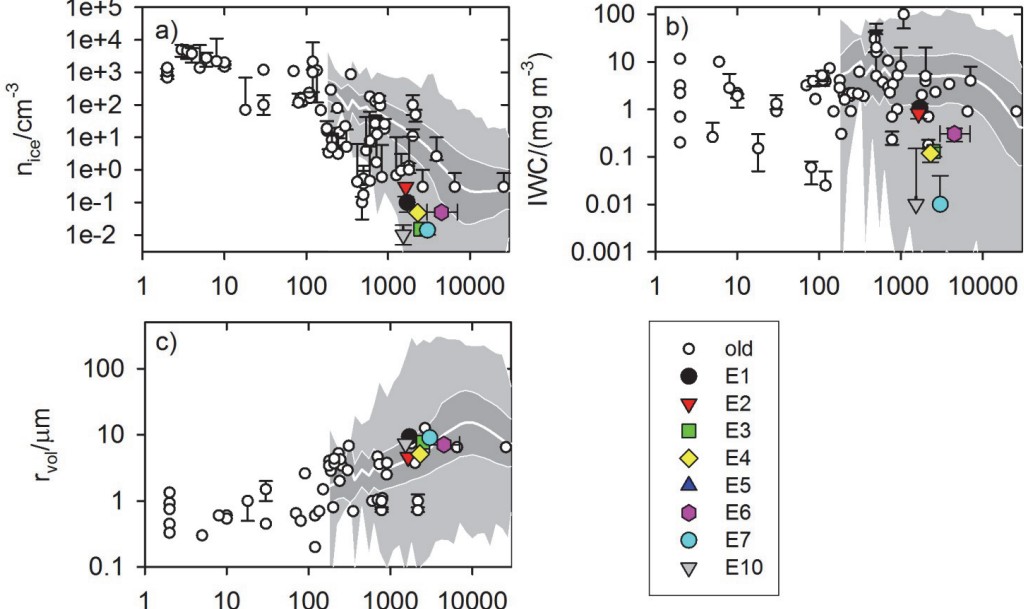

**Figure 18:** Contrail properties as function of contrail age. The grey ranges denote model results from the Contrail Cirrus Prediction model (CoCiP) coupled to the climate atmosphere and aerosol model CAM3-IMPACT (Schumann et al., 2015) with minimum and maximum values and white lines representing 10, 50 and 90 % percentiles versus age, and the black symbols with error bars are the results from previous in-situ and remote sensing measurements (Schumann and Heymsfield, 2016). The colored symbols are the data from Table 3 for events E1 to E10, 30 November 2005.





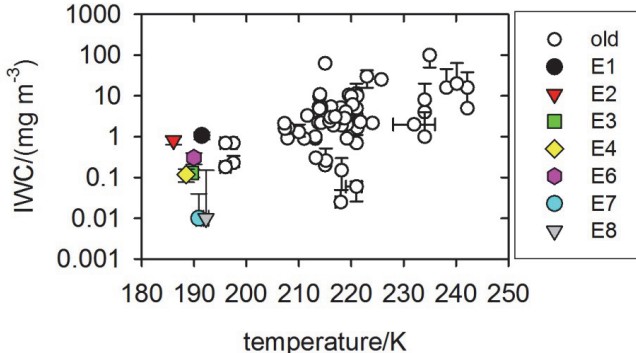

**Figure 19:** Ice water content (IWC) of contrails versus ambient temperature for various observations. Symbols as in Figure 18.

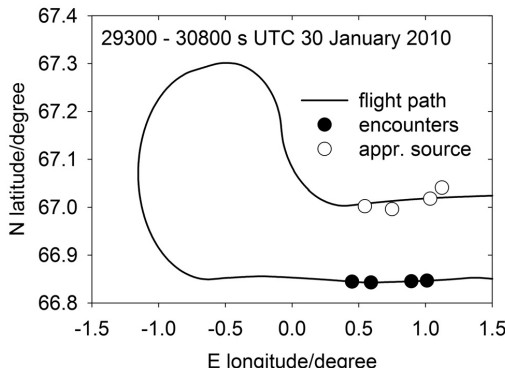

5    **Figure 20:** Geophysica flight path during RECONCILE with exhaust plume encounters in two periods between the pairs of black circles and the corresponding approximate source positions (white circles) computed for measured wind and best fitting plume ages.



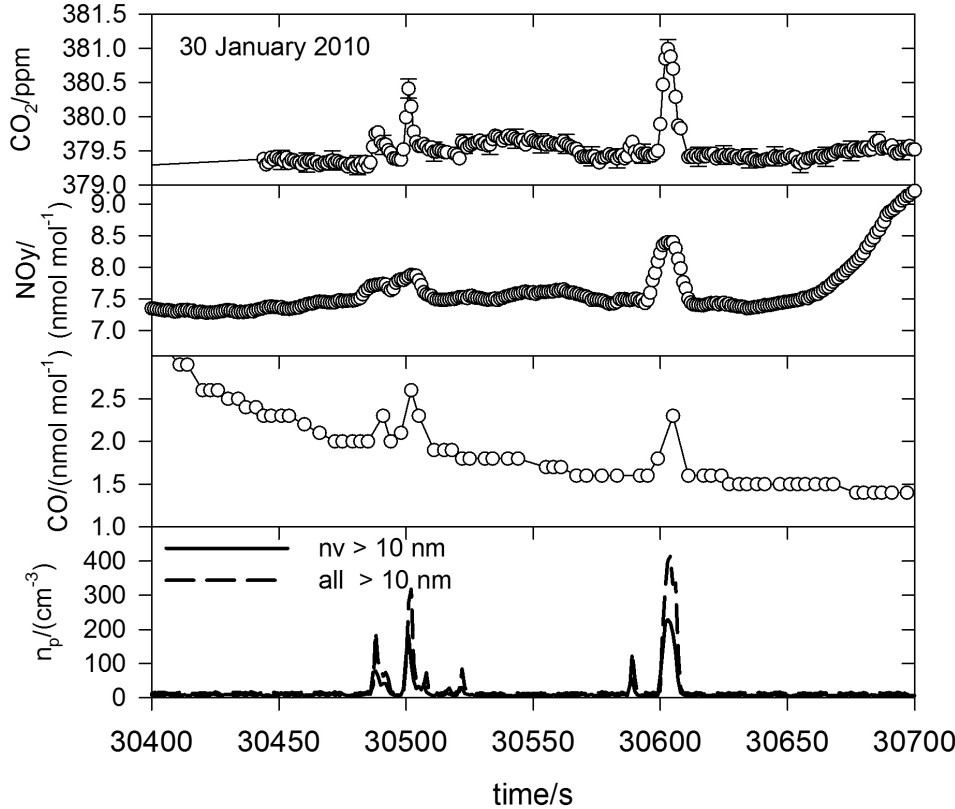

**Figure 21:** Time series (including time shifts as given in the text) of carbon dioxide ($CO_2$, with precision error bar for every 10th value), nitrogen oxides ($NO_y$), and carbon monoxide (CO) molar mixing ratios and concentrations $n_p$ of total and nonvolatile particles with diameters > 10 nm for the RECONCILE plume self-encounter of 30 January 2010.

