# Peer review of "Long-lived contrails and convective cirrus above the tropical tropopause"

_Atmospheric Chemistry and Physics, 2016_

## Referee Comment (RC1) · Anonymous Referee #1 · 23 Nov 2016

This paper presents a convincing case for observations of contrails in the stratosphere from the Geophysica flights over Hector on 16 and 30 November 2005. It covers work already published in previous studies but has enough new material to warrant publication. I have only minor comments on the manuscript.

The paper is very long, very detailed and difficult to follow because of the complex nature of the dataset (this is not a criticism of the author but it does set a challenge). Peripheral sections should therefore be omitted or moved to the supplementary material. For example, Sections 3.2.5 and 3.2.6 (together with their associated diagrams) add little to the overall argument and could safely be omitted.

p.7 l.6 - gives the impression that there is always strong turbulence or some similar disturbance. I'm sure that's not the intention so please re-draft the sentence.

[Figure]

p.7 l.33 'a remnant' rather than 'remainders'

p.8 l.27 'This cloud can be found as a narrow line in the 300° direction'

p.8 l.29 'the remains' rather than 'remainders'

p.9 l.18 'between this turn and the Falcon position.' But the Geophysica was always east of the Falcon camera position so how could the white line be between the Geophysica and the Falcon?

p.12 l.23 show not shows. Also, the tightness of the CO2 scatter plot will be affected by the precision of the measurement. The paper gives a high-frequency noise of 0.05 ppmv which is very small but are there lower-frequency contributions to the random errors?

p.14 l.25 '. . ...data; MAL is. . .. . .'

p.15 l. 33 cause the optical depth to be underestimated

p.16 l.4 has not have; delete 'reaches' from next line

p.22 l.20 either 'during 30 November' or 'during the 30th November'

p.24 l.5 'in this altitude range' - which altitude range?

p.26 l.17 'Effective means,' should read 'By 'effective'; we mean. . .'

---

## Referee Comment (RC2) · D. Baumgardner (Referee) · 10 Dec 2016

The material presented in this manuscript represents a very detailed case study of the cirrus clouds above the Hector convective system. Technically all the "T"s are crossed and the "I"s dotted. A very clever methodology is employed to ascertain the location and movement of the contrail(s) that were created by the Geophysica aircraft.

The manuscript is rather lengthy and perhaps more of the technical details could be moved to the Appendix or supplementary material than is already there. If, however, the authors feel that these details need to remain in the main text, that is their prerogative. I only make this suggestion as it took me, as a reviewer a number of readings to make it through the main text and glean what I think are meant to be the primary results.

This brings me to my primary concern and suggestion. From the abstract and introduction, it seems that the primary objectives of this study are to: 1) present a methodology that will be used to extract contrail evidence from the obscuring natural cirrus, 2) demonstrate this technique with a case study of contrails near or mixed with cirrus, 3) corroborate the results with in situ and remote sensing measurements. From my perspective as the reviewer, these three phases are not delineated clearly enough from the beginning.

Hence, my strong recommendation is that the final section of the introduction should be amended to include a road map that clearly describes the objectives of the study and a step by step elucidation of how the authors plan to achieve those objectives.

My second suggestion, perhaps less strong, is that the sections that discuss the lidar, radar and satellite measurements with respect to the contrails should be shortened as none of the remote sensing results make a strong case for the presence of the contrails. The text and figures in Sections 3.2.2-3.2.5 occupy a fairly large fraction of the paper without seeming to be tied that strongly to the contrails themselves other than demonstrating that there were a lot of cirrus accompanying the contrails.

My final point is that in the section on in situ measurement methodology, more needs to be said about the potential for contamination from ice crystal shattering on the FSSP and CIP and the uncertainty due to the very small sample areas of the two instruments. A great deal has been discussed in the references that are cited, but at least a paragraph is needed to explain why it is likely that shattering is not an issue here and that the low concentrations measured by the FSSP are at the measurement threshold of this instrument. For example, A concentration of 0.01 cm-3 shown from the FSSP represents a single particle in the one second measurement interval, assuming that the aircraft is flying at 150 ms-1. The uncertainties and limitations do not change the results or conclusions but they do underscore the difficulty of extracting information from measurements like these.

I have attached a copy of the manuscript that I have annotated is addition comments

that the authors can address and recommendations that they can implement should they so choose.

Please also note the supplement to this comment:
http://www.atmos-chem-phys-discuss.net/acp-2016-940/acp-2016-940-RC2-supplement.pdf

**Supplement:**

[revised manuscript text omitted]

**Figure 7:** Projections of the flight path (G, black line) of the Geophysica for 30 November 2005 with potential contrail and "dry" (without ice) plume encounter positions and drift lines in space and time in four Cartesian projections relative to the positions of coldest (blue star) and warmest (red star) air masses as identified from the vertical soundings (see Figure 2). Black closed circles denote positons of computed plume or contrail encounters (E) and open circles denote the places from where the plumes or contrails originate (C). The dash-dotted lines are drift paths (trajectories) from C to E points. Ice forming drift paths are indicated in cyan color. The first short ice encounter E0 is identified by a red circle. Events E7 (a potential contrail event with some ice) and E8 (dry) on the windward side (towards north-east) are the plumes with maximum nv particle concentrations.

[revised manuscript text omitted]

---

## Referee Comment (RC3) · Anonymous Referee #3 · 18 Dec 2016

General comments:

This manuscript presented analysis of measurements made during the SCOUT-O3 field experiment concerning long-lived contrails and convection-induced cirrus above the tropical tropopause. While contrails in the upper troposphere occur quite frequently and have been studied extensively in the past, those occurring in the lower stratosphere at very cold temperatures and low turbulence environment are considered rare cases. The very long lifetime ($\sim$1h) of the stratospheric contrail in a sub-saturated environment is especially remarkable. An effective procedure was used by the authors to separate encounters of contrails from other naturally-occurring stratospheric cirrus, which I find very interesting.

The paper also studies stratospheric cirrus that was likely produced by overshooting

deep convection (Hector) near Darwin Australia. A suite of synergistic measurements were utilized to bear on the stratospheric cirrus case, including in situ data (from Geophysica), downward-looking lidar (from Geophysica), upward-looking lidar (from DLR Falcon), ground-based CPOL radar, and satellite IR images. The authors compared the in situ measurements of stratospheric cirrus microphysics with that inferred from lidar and radar data.

The most rewarding part for me to read the manuscript as a reviewer is the detailed discussion written by the authors. The discussions placed the observations in a broader context. I started to better understand the significance and mechanism of the long lifetime of the stratospheric contrail. The authors also weighed evidences in an effort to explain the origin of the stratospheric clouds observed, i.e., whether they are produced by the exhaust of Geophysica or by the Hector cloud. I can clearly see that the senior author's many years of experience in this field gives him a unique vintage point to deliver these nice discussions.

There is one improvement I'd like to suggest: the body part of the paper, namely, Section 3 (Results), is not very well organized. After reading it, I felt as if I've been walking in the woods, seeing many trees, but not sure where I was led to. Each observation and discussion in Section 3 seem interesting by themselves, but it's just how they are connected to the punch line or key points of the paper that is easily lost to me. I had to read the section several times to piece together the whole story. It would be easier if the authors can present a clear road map in the introduction, and perhaps give us a preview of the key results and main findings of each subsection.

Overall, I find this paper interesting and believe it should contribute to the literature. I'd suggest minor but mandatory revision.

Specific comments:

(Page 5, Lines 25-26) Some more discussion is needed to elaborate how the two extreme temperature readings are related to overshooting convection. Which extreme?

[Figure]

Warm or cold extreme?

(Page 12, Line 9) Shouldn't Figure 6b be Figure 6d? I don't see any "black circles" on Fig. 6b.

**[ACPD](ACPD)**

---

## Author Comment (AC1) · 20 Jan 2017

We thank the Referee for his comments. The comments help us to strengthen the paper.

We repeat the comments after "C:" and add replies after R:

C. This paper presents a convincing case for observations of contrails in the stratosphere from the Geophysica flights over Hector on 16 and 30 November 2005. It covers work already published in previous studies but has enough new material to warrant publication. I have only minor comments on the manuscript.

R. Obviously, we did not make clear enough that this paper deals with two topics. As the title says "Long-lived Contrails AND convective cirrus". We now changed the abstract

and added a roadmap at the end of the Introduction to make these points clearer.

C. The paper is very long, very detailed and difficult to follow because of the complex nature of the dataset (this is not a criticism of the author but it does set a challenge). Peripheral sections should therefore be omitted or moved to the supplementary material. For example, Sections 3.2.5 and 3.2.6 (together with their associated diagrams) add little to the overall argument and could safely be omitted.

R: The paper is long because, because it is complex to distinguish between contrail and convective cirrus. Each of the various observations provides different perspective on the complex situation. The situation would be interpreted differently without having seen all material. The lidars, the AVHRR, radar, and MTP data in combination with previously unavailable and new analyses of the in-situ data (for instance CO2) provide information on the nature of the Hector anvil cloud not available before. Since the cirrus above Hector is at least partially affected by Geophysica exhaust and contrails, the question arises of how valid previous conclusions are about the contribution of deep convection to hydration of the lower stratosphere. Therefore, we do not eliminate sections as suggested, but now stress their importance in the Introduction.

The other comments are of minor nature and are fully taken into account in the revised version as follows:

C: p.7 l.6 - gives the impression that there is always strong turbulence or some similar disturbance. I'm sure that's not the intention so please re-draft the sentence.

R: we omit "strong!

p.7 C: l.33 'a remnant' rather than 'remainders'

R: changed as recommended.

p.8 C: l.27 'This cloud can be found as a narrow line in the 300åŮę direction'

R: we add word "the".

C: l.29 'the remains' rather than 'remainders'

R: changed as recommended.

C: l.18 'between this turn and the Falcon position.' But the Geophysica was always east of the Falcon camera position so how could the white line be between the Geophysica and the Falcon?

R: The answer is: because of drift with wind

C: p.12 l.23 show not shows.

R: The sentence was "The scatterplots show compact correlations. " This needs no change.

C: Also, the tightness of the CO2 scatter plot will be affected by the precision of the measurement. The paper gives a high-frequency noise of 0.05 ppmv which is very small but are there lower-frequency contributions to the random errors?

R. Indeed, the total random error for each of the SCOUT flights used in this study is estimated as 0.18 ppm for CO2 molar mixing ratio (including lower-frequency contributions). We changed the text to make it more precise: The errors given with the data are an estimate for the mean precision during the whole flight, including a calibration bias that is constant for a given flight (but may differ between flights); for absolute accuracy, one has to add 0.1umol mol-1. The total random error within each of the SCOUT-O3 flights is estimated as 0.18 umol mol-1, while the high-frequency noise (relevant for the detection of small CO2 peaks) is about 0.05 umol mol-1.

C. p.14 l.25 '. . ...data; MAL is. .

R: changed as recommended.

p.15 C: l. 33 cause the optical depth to be underestimated

R: changed as recommended.

p.16 C: l.4 has not have; delete 'reaches' from next line

R: changed as recommended.

C: p.22 l.20 either 'during 30 November' or 'during the 30th November' -

R: changed according as suggested in the second version.

C: p.24 l.5 'in this altitude range' - which altitude range?

R: much higher than in the troposphere

C: p.26 l.17 'Effective means,' should read 'By 'effective'; we mean. . .'

R: changed as recommended.

NEW Abstract

Abstract. This study has two objectives; 1) it characterizes contrails at very low temperatures and 2) it discusses convective cirrus in which the contrails occurred. 1) Long-lived contrails and cirrus from overshooting convection are investigated above the tropical tropopause at low temperatures down to -88°C from measurements with the Russian high-altitude research aircraft M-55 "Geophysica" and related observations during the SCOUT-O3 field-experiment near Darwin, Australia, in 2005. A contrail was observed to persist below ice saturation at low temperatures and low turbulence in the stratosphere for nearly one hour. The contrail occurred downwind of the decaying convective system "Hector" of 16 November 2005. The upper part of the contrail formed at 19 km altitude in the tropical lower stratosphere at ïA¿60 % relative humidity over ice at -82°C. The ïA¿1-h lifetime is explained by engine water emissions, slightly enhanced humidity from Hector, low temperature, low turbulence, and possibly nitric-acid hydrate formation. The long persistence suggests large contrail coverage in case of a potential future increase of air traffic in the lower stratosphere. 2) Cirrus observed above the strongly convective Hector cloud on 30 November 2005 was previously interpreted as cirrus from overshooting convection. Here we show that parts of the cirrus were

caused by contrails or are mixtures of convective and contrail cirrus. The in situ data together with data from an upward-looking lidar on the German research aircraft "Falcon", the CPOL radar near Darwin, and NOAA-AVHRR satellites provide a sufficiently complete picture to distinguish between contrail and convective cirrus parts. Plume positions are estimated based on measured or analyzed wind and parameterized wake vortex descent. Most of the non-volatile aerosol measured over Hector is traceable to aircraft emissions. Exhaust emission indices are derived from a self-match experiment of the Geophysica in the polar stratosphere in 2010. The number of ice particles in the contrails is less than 1 % of the number of non-volatile aerosol particles, possibly because of sublimation losses and undetected very small ice particles. The radar data show that the ice water content in convective overshoots is far higher than measured along the flight path. These findings add insight into overshooting convection and are of relevance with respect to hydration of the lower stratosphere.

Road-Map, to be introduced at the end of the Introduction:

This study started with the objective to characterize contrails at very low temperatures based on previous airborne measurements above the tropical tropopause. For this purpose we developed a method to identify encounters of exhaust plumes or contrails along the flight track of the aircraft with a trajectory analysis and subsequent discussion of the measured plume properties in respect to exhaust and contrail signatures. Since some contrails were found mixed with convective cirrus, we had to extend this study considerably to characterize also the convective clouds. Section 2 describes the measurements and the data available for analysis. It also describes a method to identify contrails based on plume trajectories. The analysis uses emission indices of the Geophysica as determined in the appendix. Section 3 describes the measurement and analysis results. Section 3.1 analyses the properties of the contrail seen in Figure 1 in the aged outflow of the decaying Hector cloud of 16 November 2005. Section 3.2 describes the measurements inside Hector under strongly convective conditions during 30 November 2005. The results provide indications for potential contrail penetrations

and insight into convective and anvil cirrus. The results are discussed in Section 4. Section 4.1 tries to explain the long live time of the contrail observed in the photo using various simplified ice mixing and sublimation models. Section 4.2 and 4.3 discuss the results of Section 3.2 and show that the measured cirrus samples were partially caused by contrails. Section 4.4 discusses the number of ice particles in contrails at low temperatures. Section 5 provides the conclusions.
* * *

---

## Author Comment (AC2) · 20 Jan 2017

Response to Referee #2 (Darrel Baumgardner)

We thank the Referee for his comments. The comments help us to strengthen the paper.

We repeat the comments after "C:" and add replies after R:

C: The material presented in this manuscript represents a very detailed case study of the cirrus clouds above the Hector convective system. Technically all the "T"s are crossed and the "I"s dotted. A very clever methodology is employed to ascertain the location and movement of the contrail(s) that were created by the Geophysica aircraft. The manuscript is rather lengthy and perhaps more of the technical details could be
moved to the Appendix or supplementary material than is already there. If, however, the authors feel that these details need to remain in the main text, that is their prerogative.

R: We considered shortening and moving parts of the material into the supplement. However, we prefer to keep the basic structure for reasons given below.

C: I only make this suggestion as it took me, as a reviewer a number of readings to make it through the main text and glean what I think are meant to be the primary results.

R: We agree that the material is demanding and requires careful reading. We now describe the strategy of the paper's roadmap in the Introduction. This paper separates the description of the results from the discussion. The individual results can be understood adequately only after the complete (and still limited) set of facts has been described. Our picture on the convective cirrus and contrail situation is the result of a quasi-forensic investigation of many details, and it seems unavoidable that the total picture can be fully grasped only after a second or even third read.

C: This brings me to my primary concern and suggestion. From the abstract and introduction it seems that the primary objectives of this study are to: 1) present a methodology that will be used to extract contrail evidence from the obscuring natural cirrus, 2) demonstrate this technique with a case study of contrails near or mixed with cirrus, 3) corroborate the results with in situ and remote sensing measurements. From my perspective as the reviewer, these three phases are not delineated clearly enough from the beginning.

R: Obviously, we did not make clear enough that this paper deals two topics. As the title says "Long-lived Contrails AND convective cirrus". We now try to make this clearer in the changed abstract and added a roadmap at the end of the Introduction. See reply to Referee 1.

C: Hence, my strong recommendation is that the final section of the introduction should be amended to include a road map that clearly describes the objectives of the study

and a step by step elucidation of how the authors plan to achieve those objectives.

R: Thank you. We follow this suggestion. See reply to Referee 1

C: My second suggestion, perhaps less strong, is that the sections that discuss the lidar, radar and satellite measurements with respect to the contrails should be shortened as none of the remote sensing results make a strong case for the presence of the contrails. The text and figures in Sections 3.2.2-3.2.5 occupy a fairly large fraction of the paper without seeming to be tied that strongly to the contrails themselves other than demonstrating that there were a lot of cirrus accompanying the contrails.

R: This should now be clearer. We are not only discussing contrail properties; we are also discussing convective cirrus (see Title).

C: My final point is that in the section on in situ measurement methodology, more needs to be said about the potential for contamination from ice crystal shattering on the FSSP and CIP and the uncertainty due to the very small sample areas of the two instruments. A great deal has been discussed in the references that are cited, but at least a paragraph is needed to explain why it is likely that shattering is not an issue here and that the low concentrations measured by the FSSP are at the measurement threshold of this instrument. For example, A concentration of 0.01 cm-3 shown from the FSSP represents a single particle in the one second measurement interval, assuming that the aircraft is flying at 150 ms-1. The uncertainties and limitations do not change the results or conclusions but they do underscore the difficulty of extracting information from measurements like these.

R: We agree: We cite Frey et al. (2011, ACP, doi:10.5194/acp-11-5569-2011, 2011, reference now added to the paper): A "widely discussed problem for in-situ ice particle measurements is the shattering of ice crystals on the probe's arm tips and shrouds or inlets (e.g. Field et al., 2006; Lawson et al., 2008; Jensen et al., 2009; Korolev et al., 2011; Lawson, 2011)."

Shattering is a minor issue in these measurements above the tropopause because of the relatively small particle sizes and the low IWC at the low temperatures. The data show no indication for an overestimate of particle concentrations.

We refer to de Reus et al. (2009), Figures 4 and 11, and the extensive discussion in Frey et al. (2011), including its supplement. The AMMA and Hector clouds are similar in respect to ice microphysics. The data sets have been carefully screened to identify and filter out potential shattering events. The comparison between in-situ and MAS backscatter signals in the paper by Cairo et al. shows that shattering cannot have played a major role, because shattering should have become obvious from differences in MAS and FSSP data. These conclusion for the previous studies was that shattering effects for cloud IWC smaller than 0.1 mg m-3 should be small (de Reus et al., 2009; Cairo et al., 2011).

We agree with Darrel Baumgardner, that the small sample areas of the two instruments affect the resolution and accuracy of the ice particle measurements. The uncertainty of the small sample areas; has been considered in the careful error analyses of the corresponding publications and the underlying Ph.D. theses. The reviewer knows these studies as reviewer or co-author. As we mention in our paper, also we see problems in comparing IWC from local measurements with small sample cross-sections with the far more integral results from satellite footprints, radar returns and model grid points results.

Hence, we now write:

The sampling volume of the FSSP100 limits the detectability of ice particles with sizes > 2.7 ïA■m to concentrations > 0.003 cm-3 for 1 Hz data (de Reus et al., 2009). Shattering aspects for the FSSP/CIP instruments are of lower importance for this study because of low temperatures, low ice water content, and low fraction of large ice particles (de Reus et al., 2009; Frey et al., 2011; Cairo, et al., 2011).

C: I have attached a copy of the manuscript that I have annotated is addition comments

that the authors can address and recommendations that they can implement should they so choose.

Essential comments in the text

This Abstract seems overly descriptive and more like a summary and conclusions. Recommend shortening substantially with only a brief description of the objectives, the methodology used to address the objectives and the significant results.

R: The abstract has been restructured to make the objectives clearer. See reply to Referee 1.

C: High-flying aircraft : There is nothing in principle incorrect about this term, but in the aircraft research community I think it is more common to use the term "High altitude". Just a suggestion.

R: "High-flying" was used in previous papers, e.g., by Peter et al (GRL, 1991). Nevertheless, we follow your suggestions and changed the text accordingly.

C: A one or two sentence explanation of what a self-match experiment is would be helpful. I have never heard this term used and I am fairly knowledgeable of airborne research.

R: We add an explanation: self-match experiment, characterizing the change in composition of an air mass between two measurements.

C: It would be helpful to label with a number or letter where the various contrails are as I see two maybe three possible contrail signatures.

R: This is now done. We add labels U1, U2 and L1, L2 for upper and lower contrail parts computed without and with wake descent.
* * *

---

## Author Comment (AC3) · 20 Jan 2017

We highly appreciate the general comments by the reviewer.

The comments are repeated (after "C:") here for reference. Our replies follows after "R:":

C: This manuscript presented analysis of measurements made during the SCOUT-O3 field experiment concerning long-lived contrails and convection-induced cirrus above the tropical tropopause. While contrails in the upper troposphere occur quite frequently and have been studied extensively in the past, those occurring in the lower stratosphere at very cold temperatures and low turbulence environment are considered rare cases.

The very long lifetime ($\_1$h) of the stratospheric contrail in a sub-saturated environment

is especially remarkable. An effective procedure was used by the authors to separate encounters of contrails from other naturally-occurring stratospheric cirrus, which I find very interesting.

The paper also studies stratospheric cirrus that was likely produced by overshooting deep convection (Hector) near Darwin Australia. A suite of synergistic measurements were utilized to bear on the stratospheric cirrus case, including in situ data (from Geophysica), downward-looking lidar (from Geophysica), upward-looking lidar (from DLR Falcon), ground-based CPOL radar, and satellite IR images. The authors compared the in situ measurements of stratospheric cirrus microphysics with that inferred from lidar and radar data.

The most rewarding part for me to read the manuscript as a reviewer is the detailed discussion written by the authors. The discussions placed the observations in a broader context. I started to better understand the significance and mechanism of the long lifetime of the stratospheric contrail. The authors also weighed evidences in an effort to explain the origin of the stratospheric clouds observed, i.e., whether they are produced by the exhaust of Geophysica or by the Hector cloud. I can clearly see that the senior author's many years of experience in this field gives him a unique vintage point to deliver these nice discussions.

Then the referee provides suggestions for improvements.

R: We thank the referee for these suggestions.

C: There is one improvement I'd like to suggest: the body part of the paper, namely, Section 3 (Results), is not very well organized. After reading it, I felt as if I've been walking in the woods, seeing many trees, but not sure where I was led to. Each observation and discussion in Section 3 seem interesting by themselves, but it's just how they are connected to the punch line or key points of the paper that is easily lost to me. I had to read the section several times to piece together the whole story. It would be easier if the authors can present a clear road map in the introduction, and perhaps give

us a preview of the key results and main findings of each subsection.

R: We now provide a road map at the end of the Introduction. See reply to referee 1.

C: Overall, I find this paper interesting and believe it should contribute to the literature. I'd suggest minor but mandatory revision. Specific comments: (Page 5, Lines 25-26) Some more discussion is needed to elaborate how the two extreme temperature readings are related to overshooting convection. Which extreme? Warm or cold extreme?

R: We mean both the warm and the cold extreme values marked with red/blue stars. We now mention these symbols in the text.

C: (Page 12, Line 9) Shouldn't Figure 6b be Figure 6d? I don't see any "black circles" on Fig. 6b.

R: Correct, it is 6d.

---

## Author Response (AR1)

**Response**

We thank the reviewers for their helpful comments.

5 The comments have been taken into account as explained in the replies to the reviewers.

1

The following text identifies the changes in color.

10

**Long-lived contrails and convective cirrus above the tropical tropopause**

Ulrich Schumann1, Christoph Kiemle1, Hans Schlager1, Ralf Weigel2, Stephan Borrmann2,3, Francesco D'Amato4, Martina Krämer5, Renaud Matthey6, Alain Protat7, Christiane Voigt1,2, C.Michael Volk8

[revised manuscript text omitted]

$$\mathbf{x}_{c}(t + \Delta t) = \mathbf{x}(t) + \mathbf{u} \Delta t, \ \mathbf{y}_{c}(t + \Delta t) = \mathbf{y}(t) + \mathbf{v} \Delta t, \ \mathbf{z}_{c}(t + \Delta t) = \mathbf{z}(t) + \mathbf{w} \Delta t.$$
(2)

An example of results of this method is shown in Figure 3. Here the black curve is the Geophysica flight path between times  $t_1$  and  $t_2$ , and the red curve is the computed position of the plume at time  $t_2$  for a plume that started from the aircraft positions in the time interval between  $t_1$  and  $t_2$ . Fo

(3)

As one of several criteria for assessment of the likelihood that the computed potential plume encounters are real encounters, we compute the change in wind velocity that would be required to advect the plume exactly to the position of the measurement in the time period  $(t_1, t_2)$ .

$$\Delta u = \Delta x / \Delta t$$
,  $\Delta v = \Delta y / \Delta t$ ,  $\Delta w = \Delta z / \Delta t$ ,

5

10

15

where  $\Delta t = t_2 - t_1$ , and  $\Delta x$ ,  $\Delta y$ ,  $\Delta z$  are the separations between the positions of the plume ( $x_c$ ,  $y_c$ ,  $z_c$ ) and the aircraft (x,y,z) at time  $t_2$ . We also compute the potential-temperature difference  $\Delta \theta = \theta(t_2) - \theta(t_1)$  at the aircraft positions. Adiabatic wake vortex sinking or lifting does not change potential temperature  $\theta$ , but  $\theta$  may change by  $\Delta \theta = -5$  K after sinking wake vortices have mixed with ambient air with  $N_{BV} = 0.025$  s-1 near  $\theta = 400$  K, after, e.g.,  $\Delta z \approx 200$  m descent.

For horizontal wind we use averaged in-situ measurements because of inherent oscillations in the wind data. Aircraft are known to deviate, even in quiet air, from the straight steady flight path, performing phugoid oscillations and other aircraft dynamics oscillations with various frequencies (typically in the range 0.01 to 0.06 s-1). Frequency details and amplitudes depend on the aircraft speed and mass and on the autopilot properties (Nelson, 1998). Such aircraft oscillations become 20 obvious for the Geophysica when one plots the aircraft altitude and attitude angles as a function of time. The UCSE wind velocity increases for large roll angles. Therefore, we ignore the wind velocity data during maneuvers with roll angles  $> 5^{\circ}$ . We also ignore wind data when the wind direction turns from 360° to 0° or vice versa, because these data suffer from averaging based on yaw angles as noted by the UCSE team. The wind data show either very strong turbulence at flight levels or other disturbances. Therefore we average all data within altitude intervals of a few hundred meters, as shown in Figure 2Figure 2d and e, and use u and v interpolated vertically in the mean wind profile. Figure 2 shows the wind results 25 deduced from TDC wind data. When using UCSE wind, the mean values change by  $\pm 0.6$  m s-1. Obviously the true wind may differ from the interpolated wind velocity by more than 1 m s-1. Since the plume positions change linearly with the product of wind velocity and age, the uncertainties in these data matter, in particular for aged plumes, and require careful discussion of the results. In all applications, the analyses were repeated with variations of the wind to test the robustness of the results.

30 The vertical wind velocity was not measured. Model analyses suggest vertical velocities of about  $\pm 2 \text{ m s}^{-1}$  in the stratiform region above the tropopause and far higher velocities of up to 25 m s-1 in the convective regions (Chemel et al.,

2009). Regions with strong updrafts were avoided by the pilot as far as foreseeable. Hence, we start our analysis assuming zero vertical wind. For the plume analysis we distinguish between the primary wake and the top of the secondary wake forming above the primary wake (Paoli and Shariff, 2016). For the top of the secondary wake we assume zero descent velocity relative to ambient air. For the primary wake, which descends for some time  $t_{wake}$  until final wake vortex decay, we

- 5 estimate the descent velocity  $w_0$  and the time  $t_{wake}$  from Holzäpfel (2014). For the Geophysica, with wing span 38.4 m, mass 20 Mg and with, e.g., true airspeed 190 m s-1, at 80 hPa air pressure, -83°C temperature, and Brunt-Väisälä frequency  $N_{BV} = 0.025 \text{ s}^{-1}$ , one finds  $w_0 = 1.34 \text{ m s}^{-1}$  and  $t_{wake} = 6 t_0 \approx 132 \text{ s}$ , with  $t_0 = 22 \text{ 
[revised manuscript text omitted]

| A :        | Demandation              | Name technique generales                                                    |   | Defenence              |
|------------|--------------------------|-----------------------------------------------------------------------------|----------|------------------------|
| Aircraft / | Parameter                | Name, technique, remarks                                                    | Δt       | Reference or           |
| Instrume   |                          |                                                                             | / s      | Principle              |
| nt         |                          |                                                                             |          | Investigator (PI)      |
| M-55       |                          | Geophysica, a twin engine reconnaissance jet, designed and operated by      |          | PI: G. Belyaev         |
|            |                          | Myasishchev Experimental Design Bureau (MDB), Russia                        |          | Stefanutti et al.      |
|            |                          |                                                                             |          | (2004)                 |
| FISH       | H 2 O (total) | Fast In-situ Stratospheric Hygrometer, Lyman-a. Data are not available for  | 1        | PI: C. Schiller        |
|            |                          | 16 November 2005.                                                           |          | (Meyer et al.,         |
|            |                          |                                                                             |          | 2015)                  |
| FLASH      | H 2 O (gas    | Fluorescent Airborne Stratospheric Hygrometer, Lyman-α.                     | 8        | Sitnikov et al. (2007) |
|            | phase)                   |                                                                             |          |                        |
| COPAS      | Condensation             | Condensation Particle Counting (CPC) System, 4 CPCs for diameters > 6, 10   | 1        | Weigel et al. (2009)   |
|            | nuclei (total            | and 14 nm, and nonvolatile (nv) particles > 10 nm.                          |          |                        |
|            | and nv)                  |                                                                             |          |                        |
| FSSP100    | Total ice                | Forward Scattering Spectrometer Probe covering particle size diameters from | 1        | de Reus et al. (2009)  |
| and CIP    | number,                  | 2.7 to 31 µm and Cloud Imaging Probe delivering shadow cast particle        |          |                        |
|            | surface area,            | images for sizes from 25 to 1550 µm.                                        |          |                        |
|            | and volume               |                                                                             |          |                        |
|            | concen-                  |                                                                             |          |                        |
|            | trations                 |                                                                             |          |                        |
| MAS        | Aerosol                  | Multiwavelength Aerosol Scatterometer, backscatter sonde at 532 and 1064    | 10       | Cairo et al. (2011)    |
|            | backscatter              | nm wavelengths.                                                             |          |                        |
|            | ratio and                |                                                                             |          |                        |
|            | depolari-                |                                                                             |          |                        |
|            | zation                   |                                                                             |          |                        |
| МТР        | temperature              | Microwaya Temperature Profiler on M55 vertical profiler. The MTP data       | 30       | PI: M. I. Mahoney      |
| IVIII      | profile                  | have been colibrated against Darwin radiocondes                             | 50       | (Denning et al         |
|            | prome                    | have been canorated against Darwin radiosondes.                             |          | (Denning et al.,       |
|            | 1 1                      |                                                                             | 20       | 1989)                  |
| MAL        | backscatter              | Miniature Aerosol Lidar, nadir-pointing backscatter lidar (532 nm           | 30       | PI: V. Mitev (Corti    |
|            | profile                  | wavelength) with depolarization                                             |          | et al., 2008)          |
| FOZAN      | O 3           | Fast-response chemiluminescent airborne ozone analyzer                      |          | Ulanovsky et al.       |
|            |                          |                                                                             |          | (2001)                 |
| SIOUX      | NO, NO y      | Stratospheric Observation Unit for Nitrogen Oxides, chemiluminescence.      | 1        | PI: H. Schlager        |
|            |                          | Data are not available for 30 November 2005                                 |          | (Voigt et al., 2007)   |
| COLD       | СО                       | Cryogenically operated laser diode spectrometer, tunable diode laser        | 4        | Viciani et al. (2008)  |

| HAGAR  | CO 2 | High Altitude Gas Analyzer, IR absorption for CO2. The data have been        | 3   | PI: M. Volk (Homan    |
|--------|-----------------|------------------------------------------------------------------------------|-----|-----------------------|
|        |                 | reanalyzed for this study. Data are not available for 16 November 2005.      |     | et al., 2010)         |
| TDC    | T, wind         | Thermodynamic Complex, Rosemount probe PT-100, and 5-hole probe.             | 0.1 | Shur et al. (2007)    |
| UCSE   | p, T, wind,     | Basic meteorology measurement system                                         | 1   | Sokolov and           |
|        | position        |                                                                              |     | Lepuchov (1998)       |
| Falcon |                 | Falcon-20 E, twin engine jet, operated by DLR (call sign D-CMET)             |     | Krautstrunk and       |
|        |                 |                                                                              |     | Giez (2012)           |
| DIAL   | backscatter     | Water vapor Differential Absorption Lidar; backscatter ratio at 532 and 1064 | 10  | PI: G. Ehret (Poberaj |
|        | and             | nm wavelengths, and depolarization                                           |     | et al., 2002)         |
|        | depolari-       |                                                                              |     |                       |
|        | zation          |                                                                              |     |                       |
| D-CMET | position        | Falcon basic measurement system                                              | 1   | PI: A. Giez           |

| Region                        | Invisible plume   | Upper contrail | Lower contrail | Tropopause     |
|-------------------------------|-------------------|----------------|----------------|----------------|
| UTC time/(h:min)              | 7:30              | 7:42           | 7:58           | 8:10           |
| Flight time/s                 | 27000-27751       | 27751-28721    | 28721-29680    | 29400-29700    |
| Age/min                       | 53 to 65          | 36 to 53       | 20 to 36       | n.a.           |
| z/km                          | 19.06±0.1         | 18.6±0.1       | 17.9±0.3       | 17.3±0.04      |
| p/hPa                         | 65.2±1.1          | 71.2±1.2       | 80.8±4         | 89.0±0.8       |
| T/°C                          | -77.5±0.8         | -82.2±0.6      | -84.9±2.0      | -88.6±0.2      |
| $H_2O/(\mu mol\ mol^{-1})$    | 3.8±0.25          | 3.4±0.23       | 2.5±0.44       | 1.9±0.2        |
| RHi/%                         | 31±5              | 64±7           | 87±23          | 132±13         |
| NO/(nmol mol -1 )  | $0.58{\pm}\ 0.09$ | 0.28±0.03      | 0.22±0.05      | $0.17 \pm .04$ |
| NOy/(nmol mol -1 ) | 2.6±0.3           | 1.3±0.1        | 0.78±0.28      | $0.40\pm0.04$  |
| $n_{nv}/cm^{-3}$              | 5.3±0.7           | 12±2           | 45±35          | 85±36          |
| $n_{10}/cm^{-3}$              | 18±3              | 30±3           | 136±134        | 327±85         |
| $O_3/(nmol mol^{-1})$         | 402±46            | 220±26         | $115\pm 61$    | 35±4           |
| CO/(nmol mol -1 )  | 32±10             | 47±6           | 53±4           | 55±2.8         |
| $T_{LC}/^{\circ}C$            | -65.0             | -63.7          | -62.4          | -60.5          |
| T ICE /°C          | -84.7             | -84.8          | -85.9          | -86.9          |
| $T_{NAT}/^{\circ}C$           | -79.6             | -80.5          | -81.9          | -83.6          |
| z'/m                          | 1.8               | 2.4            | 7.3            | 7.6            |
| $N_{\rm BV}/s^{\text{-}1}$    | 0.0263            | 0.0269         | 0.026          | 0.03           |

Table 2. Mean properties of uppermost invisible plume, upper ("white"), and lower ("red") contrail, and tropopauseproperties with ranges or standard deviations for 16 November 2005.

|                           | unit               | E0    | E1       | E2        | E3       | E4         | E5       | E6        | E7               | E10              |
|---------------------------|--------------------|-------|----------|-----------|----------|------------|----------|-----------|------------------|------------------|
| t                         | h:min              | 5:13  | 5:43     | 5:52      | 5:55     | 6:21       | 6:28     | 6:55      | 6:00             | 6:08             |
| t                         | S                  | 18780 | 20600    | 21140     | 21300    | 22863      | 23318    | 24952     | 21633            | 22120            |
| $\Delta t_{e}$            | S                  | 15    | 150      | 40        | 30       | 50         | 60       | 100       | 10               | 15               |
| Age                       | S                  | n.a.  | 1709     | 1642      | 2529     | 2302       | 4433     | 3000-7157 | 3025             | 1531             |
| Z                         | km                 | 18.3  | 18       | 18        | 18.4     | 18.4       | 18.7     | 18.2      | 18.3             | 18.0             |
| $\mathbf{z}_{\mathbf{p}}$ | km                 | 18.2  | 17.7     | 17.8      | 18.2     | 18.2       | 18.6     | 18        | 18.1             | 17.8             |
| Т                         | °C                 | -83.6 | -81.7    | -87.1     | -83.9    | -84.1      | -80.9    | -83.2     | -82.3            | -80.9            |
| р                         | hPa                | 73.0  | 78.3     | 77.6      | 72.1     | 72.8       | 68.3     | 74.8      | 74.1             | 77.3             |
| θ                         | Κ                  | 402   | 396      | 386       | 401      | 399        | 414      | 398       | 403              | 401              |
| RHi                       | %                  | n.a.  | 76       | 157       | 95       | 107        | 75       | 89        | 68               | 75               |
| n ice          | cm -3   | 0.01  | 0.1      | 0.3       | 0.015    | 0.048      | n.a.     | 0.05      | 0.014            | 0.01             |
| IWC                       | mg m -3 | 0.1   | 0.85-1.3 | 0.64-0.99 | 0.1-0.16 | 0.077-0.16 | n.a0.072 | 0.21-0.4  | 0.01-0.04        | 0.01-0.015       |
| r eff          | μm                 | 5.0   | 25.3     | 17.9      | 23.6     | 11.5       | n.a.     | 19.5      | n.a              | n.a.             |
| r vol          | μm                 | 6     | 9.3      | 4.6       | 7.7      | 5.1        | n.a.     | 7.1       | 9                | 7.3              |
| $\Delta n_{\rm nv}$       | cm -3   | 0     | 20       | 40        | 100      | 40         | 50       | 50        | 1000             | 100              |
| $n_{ice}/\Delta n_{nv}$   | 1                  | -     | 0.005    | 0.0075    | 0.00015  | 0.0012     | n.a.     | 0.001     | 10 -5 | 10 -4 |

Table 3. Mean properties\* of ice events, E1 - E6 from de Reus et al. (2009) and short-term ice events E0, E7 and E10 for 30 November 2005.

\*) t: UTC time of 30 November 2005,  $\Delta t_e$ : estimated event duration, age: computed plume age, z: altitude above MSL,  $z_p$ : pressure 5 altitude, T: temperature, p: pressure,  $\theta$ : potential temperature, IWC: ice water content (lower value from observed ice crystal size distribution and upper from two hygrometers),  $r_{eff}$  and  $r_{vol}$ : effective and volume mean particle radius.  $\Delta n_{nv}$ : nv particle concentration above background.

Table 4. Potential plume self-encounters (for symbols see text), with classification as convective (C) or exhaust (E) event, 30 November 2005.

| Event   | Age/s | t/s   | t/(h:min) | z/km  | $\Delta z/km$ | $\Delta u/(m s^{-1})$ | $\Delta v/(m s^{-1})$ | $\Delta w/(m s^{-1})$ | $\Delta \theta/K$ | CIP | C or E |
|---------|-------|-------|-----------|-------|---------------|-----------------------|-----------------------|-----------------------|-------------------|-----|--------|
| ice E1  | 1708  | 20600 | 5:43      | 17.96 | -0.44         | -0.28                 | 0.91                  | -0.17                 | -10.7             | у   | C+E    |
| ice E2  | 1648  | 21140 | 5:52      | 18.00 | -0.40         | -0.12                 | -3.42                 | -0.14                 | -15.9             | у   | С      |
| ice E3  | 2512  | 21300 | 5:55      | 18.37 | 0.01          | -2.30                 | 2.58                  | 0.03                  | -2.0              | у   | C+E    |
| ice E4  | 2295  | 22863 | 6:21      | 18.37 | 0.39          | 0.14                  | -0.38                 | 0.18                  | 4.3               | no  | Е      |
| ice E5  | 4430  | 23318 | 6:28      | 18.72 | 0.32          | 0.34                  | 1.67                  | 0.08                  | 6.4               | у   | C+E    |
| ice E6a | 3006  | 24952 | 6:55      | 18.21 | 0.10          | -1.63                 | -0.93                 | 0.03                  | -1.3              | у   | C+E    |
| ice E6b | 4473  | 24952 | 6:55      | 18.21 | 0.18          | -1.41                 | 1.99                  | 0.04                  | 2.1               | у   | C+E    |
| ice E6c | 6047  | 24952 | 6:55      | 18.21 | -0.20         | -0.10                 | 0.57                  | -0.04                 | -10.3             | у   | C+E    |
| ice E6d | 7219  | 24952 | 6:55      | 18.21 | 0.19          | 1.18                  | -0.17                 | 0.03                  | -1.1              | у   | C+E    |
| ice E7  | 3025  | 21633 | 6:00      | 18.28 | -0.10         | 2.25                  | -2.99                 | -0.04                 | -6.4              | no  | Е      |
| E8      | 1969  | 21763 | 6:02      | 18.33 | 0.00          | 0.01                  | 0.13                  | 0.01                  | 2.4               | no  | Е      |
| ice E9  | 1514  | 22008 | 6:06      | 18.00 | -0.01         | 0.51                  | -0.78                 | 0.00                  | 0.4               | no  | E+C    |
| ice E10 | 1531  | 22120 | 6:08      | 18.04 | 0.08          | 0.81                  | -1.32                 | 0.07                  | 2.5               | no  | E+C    |
| E11     | 1758  | 23588 | 6:33      | 18.50 | 0.11          | -1.20                 | 2.79                  | 0.06                  | 3.1               | no  | Е      |
| E12     | 1831  | 23632 | 6:33      | 18.61 | 0.24          | 0.65                  | 3.53                  | 0.14                  | 7.3               | no  | Е      |
| E13     | 6738  | 24397 | 6:46      | 17.96 | -0.44         | -0.04                 | 0.11                  | 0.07                  | 7.0               | no  | Е      |

Table 5. Scales of the two stratospheric anvil cloud layers as seen from the Falcon in lidar backscatter signals with maximum optical depth  $\tau_{max}$  and total extinction EA, 30 November 2005.

| Layer | Begin        | Begin  | End    | Altitude       | Maximum | Maximum  | $\tau_{max}$ | EA/  | Plume     |
|-------|--------------|--------|--------|----------------|---------|----------|--------------|------|-----------|
|       | time/(h:min) | time/s | time/s | min-
max/km | depth/m | width/km |              | m    | age/h     |
| Upper | 7:34         | 27250  | 27580  | 17.1-18.5      | 900     | 67       | 0.04         | 639  | 1.56±0.81 |
| Lower | 7:36         | 27370  | 27640  | 16.5-17.6      | 1100    | 55       | 0.08         | 1505 | 2.21±0.66 |

Table 6. Emission indices derived from two self-encounters at 20 (17.7 to 21.7) min plume ages and Pearson correlation coefficients  $R^2$  relative to  $CO_2$ , 30 January 2010.

| EI or PEI for species             | Unit                              | Mean | Minimum | Maximum | R 2 |
|-----------------------------------|-----------------------------------|------|---------|---------|----------------|
| EI CO                  | g kg -1                | 1.9  | 1.1     | 2.8     | 0.81           |
| EI NOx                 | g kg -1                | 4.1  | 3.2     | 5.0     | 0.75           |
| $PEI_{nv}$ (nv, > 10 nm)          | 10 15 kg -1 | 2.6  | 2.0     | 3.2     | 0.79           |
| $PEI_{volatile}$ (total, > 10 nm) | 10 15 kg -1 | 4.4  | 3.4     | 5.2     | 0.91           |

---

## Author Response (AR2)

Dear editors,

We are pleased to hear that our paper got accepted without further changes.

Herewith, I upload the final manuscript.

**Long-lived contrails and convective cirrus above the tropical tropopause**

Except for minor text and figure editing, the paper is unchanged as accepted.

For figures 4, 11, 12, 14, I submit two file versions: pdf and png. Please use the better one.

For Figure 1, I only have the photo (jpg)

For Figure 13, I only have two jpg files (a and b). Can you please put them together?

For all other figures, I submit pdf files. I could also provide eps-files if that is better.

Best regards,

on behalf of all co-authors,

Ulrich Schumann, 24.1.2017